# Neurovascular coupling and oxygenation are decreased in hippocampus compared to neocortex because of microvascular differences

K. Shaw [1], L. Bell[1,2], K. Boyd[1,2], D. M. Grijseels [1,2], D. Clarke[1], O. Bonnar [1], H. S. Crombag [1] & C. N. Hall [1✉]

The hippocampus is essential for spatial and episodic memory but is damaged early in Alzheimer's disease and is very sensitive to hypoxia. Understanding how it regulates its oxygen supply is therefore key for designing interventions to preserve its function. However, studies of neurovascular function in the hippocampus in vivo have been limited by its relative inaccessibility. Here we compared hippocampal and visual cortical neurovascular function in awake mice, using two photon imaging of individual neurons and vessels and measures of regional blood flow and haemoglobin oxygenation. We show that blood flow, blood oxygenation and neurovascular coupling were decreased in the hippocampus compared to neocortex, because of differences in both the vascular network and pericyte and endothelial cell function. Modelling oxygen diffusion indicates that these features of the hippocampal vasculature may restrict oxygen availability and could explain its sensitivity to damage during neurological conditions, including Alzheimer's disease, where the brain's energy supply is decreased.

[1] School of Psychology and Sussex Neuroscience, University of Sussex, Falmer, Brighton, United Kingdom. [2] These authors contributed equally: L. Bell, K. Boyd, D. M. Grijseels. ✉email: catherine.hall@sussex.ac.uk

Brain processes that depend on the hippocampus (HC) become dysfunctional early in several disease states, including Alzheimer's disease and vascular dementia, as well as during aging and after lifestyle changes like the adoption of a high fat diet[1–3]. Changes in vascular structure and function are often early features of these conditions[2]. In Alzheimer's disease, blood–brain barrier dysfunction and inadequate cerebral perfusion promote Aβ accumulation and neurofibrillary tangles[4,5], and patients with mild cognitive impairment show lower local blood volume in the HC[6]. Indeed, reduced hippocampal perfusion in the absence of other pathology is itself associated with impaired cognition[7], and CA1 pyramidal cells are particularly susceptible to death after a hypoxic insult[8]. Thus, maintenance of an adequate oxygen supply may be particularly vital for the HC.

Physiologically, the brain finely regulates its oxygen supply by a process called neurovascular coupling whereby active neurons signal to dilate local blood vessels, increasing blood flow and the supply of oxygen and glucose to these active brain regions. In the neocortex, neurovascular coupling produces an increase in blood flow that more than compensates for the increase in oxygen consumed by active neurons, generating an overall increase in blood oxygenation in the active brain region. The associated decrease in levels of deoxygenated haemoglobin underlies the positive blood oxygen level-dependent (BOLD) signal. In the neocortex, this increase in BOLD correlates with neuronal activity, enabling it to be used as a surrogate measure of brain activation in functional magnetic resonance imaging (fMRI) studies. In the HC, however, the BOLD signal does not seem to reliably correlate with neuronal activity[9], as local field potential changes occur without measurable BOLD signals[10]. This not only suggests that fMRI studies measuring hippocampal BOLD are less-sensitive than cortex to increases in neuronal activity, but also that the HC may be physiologically less able to increase its oxygen supply, and this may underlie its sensitivity to damage at the onset of pathophysiological conditions.

However, because the HC lies beneath the neocortex, and is, therefore, less accessible, previous studies have not been able to directly measure neurovascular coupling in this region in vivo, and therefore no direct evidence exists whether the HC is indeed less able to regulate its energy supply. By removing the overlying cortex[11], we could implant a cranial window over the HC and record neuronal and vascular activity using two-photon imaging, laser doppler flowmetry, and haemoglobin spectroscopy to compare neurovascular coupling in dorsal CA1 to that in the visual cortex (V1). This cranial window allowed us to record from different layers across both regions, which in the V1 comprised of layers I–IV (Fig. 1a), and in CA1 of stratum oriens, stratum pyramidale, stratum radiatum and stratum lacunosum–moleculare (Fig. 1b). In V1, neuronal cell bodies are more dispersed throughout the layers, although slightly more concentrated in layer IV, whereas in CA1 the cell bodies are densely packed in stratum pyramidale and send their long, apical dendrites into stratum radiatum. We studied both resting haemodynamics and neurovascular coupling in HC and cortex, as both factors are likely to be important for maintaining neuronal and cognitive health. As discussed above, decreases in overall perfusion have been associated with brain damage[7], whereas decreases in neuronal activity-driven vessel dilations correlated with reduced oxygenation and preceded cell death in neocortex[12].

We found that the HC had lower resting blood flow and blood oxygenation compared with V1, which was due both to a lower capillary density and reduced RBC velocity and flux in individual capillaries than in V1. We then studied how well blood vessels could respond to increases in local excitatory neuronal activity and found that not only did blood vessels in the HC dilate less

frequently to local increases in activity than in V1, but when responses did occur, the dilations were smaller. Pericyte morphology and vascular expression of proteins that mediate dilation were significantly different in the hippocampal capillary bed compared with cortex, suggesting that the hippocampal vasculature is less able to dilate to increases in neuronal activity. To understand how these vascular differences impact neuronal oxygen availability, we modelled oxygen diffusion from vessels. Our results indicated that oxygen becomes limiting for ATP synthesis in tissue furthest from blood vessels in the HC, but not the V1. We propose that this decreased neurovascular function in the HC contributes to its vulnerability to damage in disease states.

## Results

We measured neurovascular coupling in the dorsal CA1 region of the HC and in V1 of head-fixed mice expressing GCaMP6f in excitatory (glutamatergic) neurons[13]. Mice could run on a running cylinder or remain stationary, whereas visual stimuli (drifting gratings or a virtual reality environment) or a black screen were presented (Methods, Fig. 1c–f). Throughout, resting baseline measurements refer to those taken when the animal is immobile and in the dark. In some experiments, combined laser doppler flowmetry/haemoglobin spectroscopy was used to record levels of deoxy- (Hbr) and oxyhaemoglobin (HbO) and cerebral blood flow (CBF) (oxy-CBF probe; Fig. 1g), allowing us to calculate regional blood oxygen saturation ($SO_2$) and the cerebral metabolic rate of oxygen consumption ($CMRO_2$; Fig. 1g, h). In other experiments, individual neurons and blood vessels were imaged using two-photon microscopy.

**Blood flow and oxygenation are lower at rest in HC than V1.** We first compared haemodynamics in HC and V1 in the absence of visual stimulation when the mouse was stationary. $CMRO_2$, reflecting energy use by summed neuronal activity, was not different between regions (Fig. 2a). However, despite similar energy demands, the resting CBF and $SO_2$ were significantly lower in HC (Fig. 2b, c). In part, these differences in net blood flow and oxygenation arise from a lower capillary density in HC than cortex (Fig. 2d, e[14,15]). However, when we measured red blood cell (RBC) flux, haematocrit and RBC velocity in individual blood vessels, after loading them with fluorescent dextran (Fig. 2g), we found that, despite the sampled vessels themselves being of equal size (Fig. 2f), RBC velocity, flux and haematocrit were also significantly lower in the HC than V1 (Fig. 2h–j). This combination of a lower capillary density and RBC velocity and flux in the HC explains both the observed lower net flow and, because fractionally more oxygen is extracted from capillaries with lower flow rates[16], the lower blood oxygenation we measured in HC.

**Blood vessels in CA1 dilate less to local neuronal activity than those in V1.** Because fMRI studies suggest neuronal activity might be less well-matched to blood flow in HC than cortex, we investigated the capacity of blood vessels to respond to local excitatory neuronal calcium events in vivo by capturing movies of neurons and nearby blood vessels (including arterioles, precapillary arterioles and capillaries) in HC and V1 using two photon microscopy (Fig. 3). Calcium events occurred across environmental conditions (i.e., in the dark or when stimuli were presented on the screens, and when the mouse was at rest or running). In both regions, vessels dilated shortly after neuronal calcium events (Fig. 3d, e, g, h), however, the frequency and size of dilations were significantly greater in V1 than in HC (Fig. 3c, e, h), whereas the average size of calcium peaks was larger in HC (Fig. 3d, g). This suggests that, relative to the amount of neuronal activity, hippocampal vessels dilate less than those in V1,

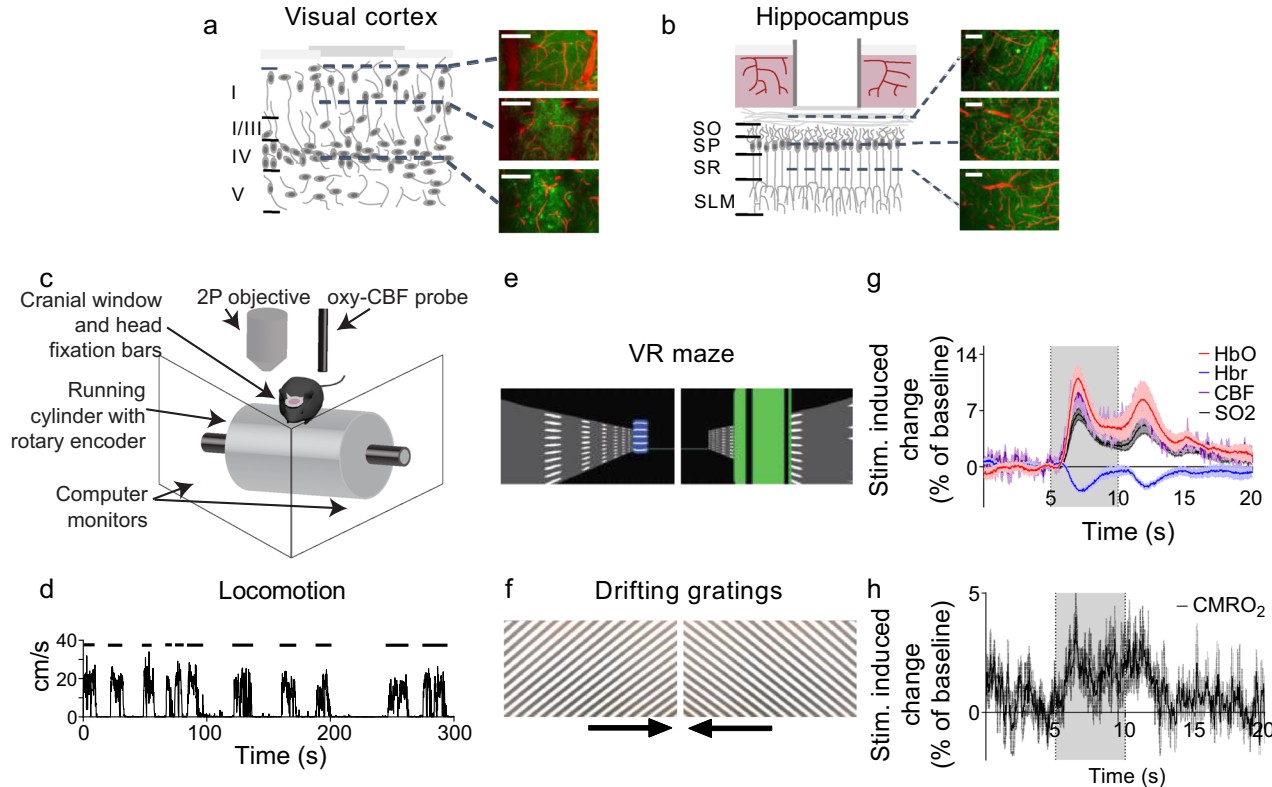

**Fig. 1 Experimental set-up and example haemodynamic data.** Representative schematic showing the GCaMP6f-positive pyramidal neurons (green) and blood vessels (red) accessible for two-photon imaging after **a** visual cortical or **b** hippocampal surgery, with example maximum-projected images across each layer. Scale bars represent 100 μm, and similar z-stack images across layers were taken for nine animals in HC and 11 animals in V1. **c** Schematic of the imaging set-up. Either the two-photon objective or oxy-CBF probe was used to collect data while the mouse was head-fixed but awake and able to run on the cylinder. **d** Representative locomotion recorded by the rotary encoder during one imaging session (centimetres per second). Distinct periods of running are indicated by the black bars. A virtual reality maze **e** or drifting gratings **f** were presented on the screens in **c**. Locomotion advanced the mice through the virtual reality maze. The arrows beneath the drifting gratings display show the direction the gratings travelled. **g** Example, haemodynamic recordings from visual cortex using the oxy-CBF probe during visual stimulation (grey bar represents stimulation, N = 4 animals, 10 sessions, 202 trials). **h** The cerebral metabolic rate of oxygen consumption ($CMRO_2$) is calculated from the haemodynamic parameters collected using the oxy-CBF probe for the data in **g** (see Methods). All data traces are unsmoothed averages, and error bands represent mean ± SEM.

supplying less oxygen. Most of our dilations are related to calcium events and not due to random vasomotion because when traces were shuffled so that vessel dilation data were no longer aligned to calcium events, dilations occurred less frequently (V1: 5.2%, HC: 5.7%; Fig. 3f, i). Even when vessels did dilate in HC, responses were smaller than in V1, despite larger triggering calcium events (Fig. 3j–m). Therefore, smaller dilations across all vessels in HC were due to both a decrease in responsiveness of vessels, and to dilations being smaller in HC when they did occur.

We next looked at vessels with diameters <7 μm to explore the effect of neuronal calcium activity specifically on capillaries. The smaller dilations observed in HC corresponded to smaller increases in RBC velocity in HC capillaries following local calcium activation, as assessed using fast line scanning of capillaries and nearby neuronal soma (Supplementary Figure 1), though RBC velocity was equally likely to increase in the two regions. Whilst these same HC capillaries captured by fast line scanning were less likely to dilate than V1 capillaries, if they did dilate, their responses were the same size in both regions. When we split our xy movie data in Fig. 3 into vessels smaller and larger than 7 μm, we confirmed that both groups of vessels were less likely to dilate in HC than V1. When they did dilate, vessels larger than 7 μm also had smaller dilations in HC, whilst dilations in smaller vessels were the same size in both regions (Supplementary Figure 2). Dilations of HC vessels larger than 7 μm were 51% of those in V1, however, so similarly scaled responses in smaller HC

vessels would be undetectable (and classed as non-responders) as they would be <60 nm, or only 0.5 SDs above baseline.

To characterise the relationship between each individual calcium event and corresponding vessel response, we calculated a neurovascular coupling index ($NVC_{index}$) by dividing responding vessel diameter peaks by the corresponding neuronal calcium peak, so a large $NVC_{index}$ represents a large dilation in response to a given change in calcium. $NVC_{index}$ was significantly higher in V1 than HC (Fig. 3n), suggesting that the hippocampal vasculature increases blood flow less in response to increased oxygen use compared with cortex. The reduced matching of blood supply to changing energetic demands in the HC could be due to a decrease in the instruction from the neurons to tell blood vessels to dilate, or a decreased ability of HC blood vessels to respond to this signal.

**HC neurons do not express lower synthetic enzymes for vasodilatory signalling pathways.** HC neurons could be less able to instruct the vasculature to dilate if HC neurons and astrocytes are less able to produce vasodilatory messengers than similar cells in V1. We tested for differences in expression levels of vasodilatory second messenger pathways in neurons and astrocytes between regions, using an open access single-cell RNA-Seq data set from neocortex (S1) and HC (CA1)[17]. There were no differences in levels of mRNA transcripts in individual pyramidal cells,

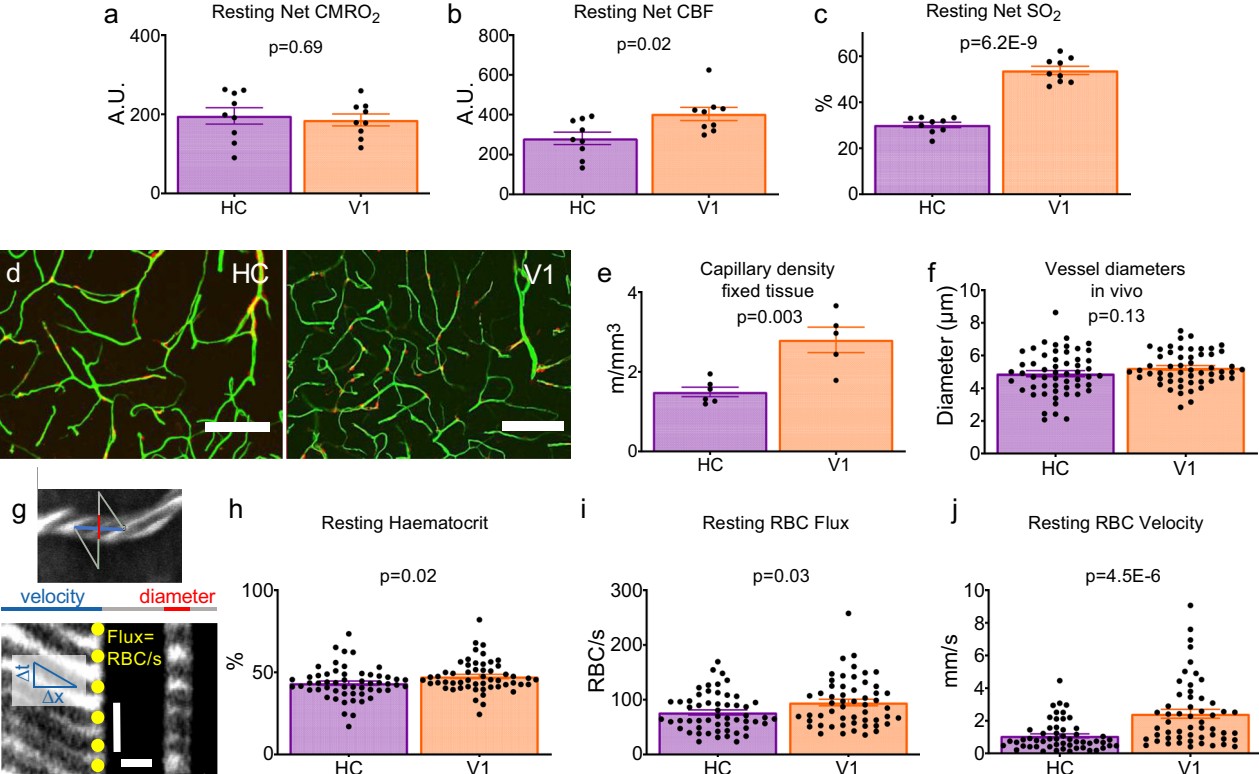

**Fig. 2 Baseline haemodynamics in HC and V1.** Estimated $CMRO_2$ **a**, CBF **b** and oxygen saturation ($SO_2$) **c** in HC (purple) and V1 (orange) in stationary mice in the dark (HC: 9 animals across 21 sessions; V1: 9 animals across 19 sessions, 1–5 sessions per mouse; data points are averages for each animal). **d** FITC-gelatin filled vessels (green) from the HC and neocortex of one NG2-DsRed mouse (pericytes are red). Scale bars represent 100 μm. Example images are projections of the 200 μm Z-stacks compared in **e**. **e** Capillary density in these slices was significantly lower in HC ($N = 6$ slices, five mice) compared with V1 ($N = 5$ slices, four mice). **f** Single capillaries were imaged in vivo using two-photon microscopy (HC: 55 vessels from 11 mice, V1: 54 vessels from 14 mice), and the average diameter of the vessels scanned did not differ between HC (mean: 4.9 μm) and V1 (mean: 5.2 μm, $t$ test $p = 0.13$). **g** Example line scan trajectory from one in vivo two-photon recording to represent the 99 vessels scanned in f (top; as input into the acquisition software—the actual trajectory will differ from that shown owing to mirror inertia). The scan path of the laser goes along (blue) and across (red) the capillary (labelled with i.v. Texas Red dextran, white. Dark stripes are RBCs). Each row in the corresponding line scan image (bottom) represents a single time point. As RBCs move, their shadow shifts along the vessel, so the angle of the stripes (left, under blue line) shows how fast they are moving. The vessel diameter can be measured from the intensity profile of the Texas Red-labelled lumen (right, under red line). The number of red blood cells per second can be calculated by counting the number of dark stripes (marked with yellow circles) within this time period. The vertical scale bar represents 5 ms, and the horizontal scale bar 5 μm. Despite the same vessel types being scanned in **f**, the average resting **h** haematocrit, **i** RBC flux and **j** RBC velocity were different between regions (statistical comparisons were made on individual vessels, $N$ specified in **f**). Data in bar charts represent mean ± SEM, dots are individual vessels or mice, as indicated. $P$ values are from independent sample $t$ tests, or Mann–Whitney $U$ tests (see Statistics Report Table SR1). Source data are provided as a Source Data file.

interneurons and astrocytes of the synthetic enzymes for prostaglandins, epoxyeicosatrienoic acids (EETs) and nitric oxide (Supplementary Tables 1–3), suggesting that cells in HC are as capable of producing these vasoactive molecules as those in cortex. In fact, there were significantly higher levels of Nos1 (Fig. 4a) and Ptges3 (Fig. 4b) expression in HC pyramidal neurons, and of Ptges3 (Fig. 4c) in HC astrocytes, which might predict that, if anything, the production of vasodilatory molecules (prostaglandin and nitric oxide) would be greater in HC than V1.

**HC neurovascular coupling is not lower because neuronal firing is less synchronous.** Alternatively, HC neurons could be less effective at signalling to blood vessels if their firing is less synchronous than in V1, so that levels of dilatory second messengers (e.g., nitric oxide or prostaglandin) summate less, meaning their concentration and potential effect on the vasculature is reduced. Indeed, coding in HC is sparser and distributed, whereas that in V1 is retinotopic, so it might be predicted that neuronal firing in HC would, indeed, be less synchronised. We tested this by

imaging across a wide field-of-view to capture large numbers of excitatory cells (Fig. 5a). We tested for synchronous firing in two ways: (1) by investigating whether the cells fired at the same time, using cross-correlation of the activity trace from each cell with that of each other cell (Fig. 5c), and (2) by measuring whether peaks in the average population calcium trace across all cells (i.e., bursts of synchronous activity) were larger in V1 than in HC (Fig. 5b, d). There were no significant differences in either the correlation of firing, or the average size of population calcium peaks between the regions.

An alternative measure of net neuronal activity is provided by the $CMRO_2$ signal. We detected peaks in $CMRO_2$, alongside corresponding regional increases in cerebral blood volume due to vascular dilation (reflected in the total blood volume, Hbt; Fig. 5e). The sizes of the peaks in the $CMRO_2$ signal (Fig. 5f, h) and the associated cerebral blood volume changes (Fig. 5g, i) were both smaller in HC than V1, but in HC, these blood volume increases were smaller relative to the change in $CMRO_2$ (Fig. 5j; $NVC_{index} = CMRO_2/Hbt$). Thus, measurements of single vessels

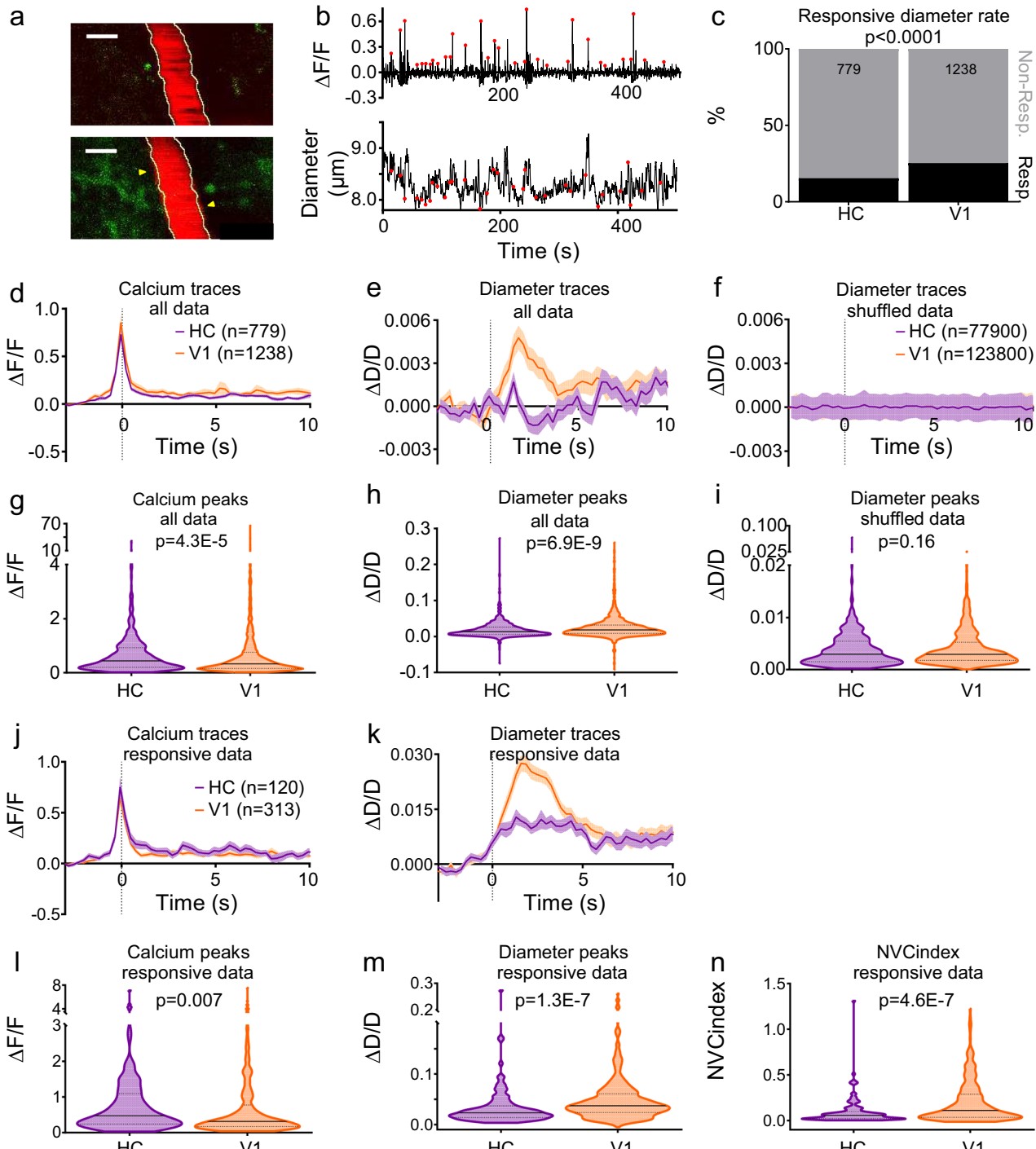

**Fig. 3 Vessel responses to local neuronal calcium events. a** Texas Red dextran-filled vessel (red) and GCaMP6f-positive pyramidal neurons (green) from one recording in V1 before (top) and during (bottom) increases in neuronal calcium (to represent the 87 vessels with local neuronal calcium imaged). Yellow outlines show vessel before calcium event. Arrows indicate the largest dilation. Scale bars represent 5 μm. **b** Neuronal calcium averaged over all cells in the field-of-view of one imaging session and the corresponding vessel diameter. Red dots mark calcium events. **c** Vessel responses to preceding calcium events were more frequent in V1 than HC (Chi-square test). **d** Average calcium response and **e** diameter change in HC (purple, $N = 779$ trials, 46 vessels, 6 animals) and V1 (orange, $N = 1238$ trials, 41 vessels, 7 animals). **f** Diameters when vessel traces were shuffled 100 times so no longer aligned to calcium events (HC $N = 77900$ events, V1 $N = 123800$). **g** The calcium events were larger in HC compared with V1. **h** Vessel dilations were larger in V1 than HC when aligned to calcium events, **i** but not when shuffled (statistical comparisons were made on individual calcium/dilation/shuffled dilation events, N specified in d/e & f). **j** Calcium events that led to dilations in HC ($N = 120$ events, six mice) and V1 ($N = 313$ events, seven mice). **k** Corresponding diameter changes. **l** Calcium peaks leading to dilations were higher in HC ($N = 120$, V1 $N = 313$). **m** Diameter peaks were significantly larger in V1 than HC. **n** A neurovascular coupling index (NVC$_{index}$) was calculated by dividing each dilation peak by its corresponding calcium peak. NVC$_{index}$ was lower in HC than V1. *P* values are from Mann–Whitney *U* tests, unless stated (see Statistics Report Tables SR2a–b). Averages and shaded error bands show mean ± SEM. Horizontal lines on violin plots show median and interquartile range. Source data are provided as a Source Data file.

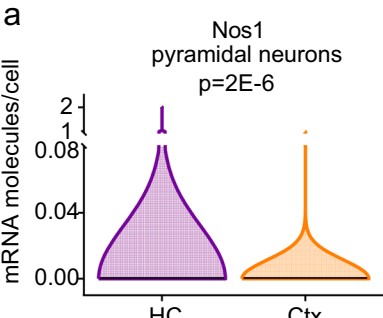
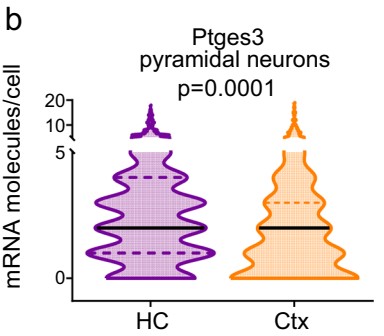
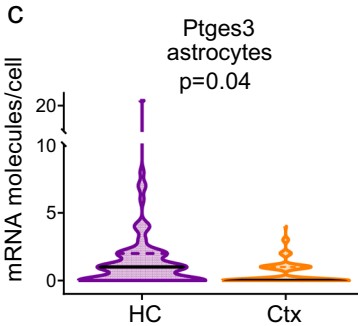

**Fig. 4 Vasodilatory second messenger pathways in HC and cortex.** The expression of **a** Nos1 and **b** Ptges3 were higher in the pyramidal cells of HC (941 cells) than cortex (398 cells). **c** Ptges3 expression was also higher in astrocytes in HC (80 cells) than V1 (143 cells). Horizontal lines on violin plots show median and interquartile range. *P* values represent the result of independent sample *t* tests, with Holm–Bonferroni corrections for the multiple comparisons presented in each of Supplementary Tables 1–3.

and summed regional responses both suggest weaker neurovascular coupling in HC.

We wondered whether the cellular or laminar organisation of HC and V1 could explain the different neurovascular coupling properties, perhaps if second messengers released from different neuronal compartments have differential effects on the vasculature. To this end, we tested whether neurovascular coupling was different in vessels in response to calcium signals in the neuropil (NP) or nearby somas (Supplementary Figure 3). We found that, although vessels in V1 were more likely to respond to calcium signals from the NP, vessel responsiveness in HC did not distinguish between the different neuronal compartments. The $NVC_{indices}$ between the different neuronal compartments were no different within each region, with a significantly greater $NVC_{index}$ in V1 than HC for both NP and soma. We also investigated whether there were laminar differences in neurovascular coupling that could explain our results. We found some differences in responses between layers in both regions (Supplementary Figure 4), however across all layers, the $NVC_{index}$ was significantly greater in V1 than HC.

**HC neurovascular coupling is not lower because of vascular deficits arising from the surgical preparation.** The surgery to create a cranial window over CA1 is more invasive than for V1, because it requires aspiration of some of the overlying cortex and the implantation of a cannula with glass coverslip. We tested for vascular network damage in CA1 in the surgical versus non-surgical hemisphere. We found that the aspiration and cannula implantation did not cause significant compression to dorsal CA1, nor alter the overall vascular density of the region (Supplementary Figure 5). Furthermore, signs of inflammation were limited to tissue <100 μm from the window and did not affect the responses recorded (Supplementary Fig 6).

**Pericytes have a less-contractile morphology in HC than V1.** In the absence of clear differences in neuronal firing properties or expression of neurovascular coupling signalling molecules, we next investigated whether our results could reflect differences in vascular structure or function. The architecture of the vasculature is well established in neocortex, where pial vessels run along the brain's surface before penetrating the tissue and branching into a dense capillary network (e.g., Figure 6a). The hippocampal vascular network is less well characterised, but is known to be inverted compared with that in neocortex, with large arteries and veins emerging in the hippocampal sulcus and sending their arch-like branches up into CA1[15]. Our in vivo recordings confirm this vascular organisation (Fig. 6a–b, Supplementary Figure 7) with the large (>15 μm) diameter perfusing vessels located ~300 μm

below the dorsal surface of layer SO. Our imaging depth was on average 70 μm (range: 1–308 μm), so the vessels we sampled in HC were generally further from their source (HC vessels averaged 227 μm from their source, versus 135 μm in V1), but the distance to the perfusion source did not alter resting RBC velocity or calcium-dependent blood vessel dilations in HC (Supplementary Figure 7). In V1, the distance from the pial arteries did not affect the size of dilations, but RBC velocity was higher in vessels nearer the source arteries (Supplementary Figure 7).

Next, we examined microvascular anatomy for differences that could help explain the functional deficits in NVC in HC. Vascular anatomy in neocortex is often described in reference to branching order from either the pia or the penetrating arteriole, but this is not possible in HC, due both to the less-stereotyped anatomy of the hippocampal vasculature, and the depth at which the largest vessels occurred (sometimes below our imaging plane). Instead, we compared vessels with similar vascular mural cell morphologies (from NG2-DsRed labelling of vascular mural smooth muscle cells (SMCs) and pericytes). The morphology of vascular mural cells is linked to their ability to constrict and dilate blood vessels[18]. Broadly, more contractile cells are shorter in length, with a denser spacing of cells along vessel walls, and more circumferential processes that cover the vessel more completely.

To better understand these morphological differences, we categorised mural cells as being SMCs, or ensheathing, mesh or thin-strand pericytes, based on the morphology of their processes (Fig. 6i)[18]. SMCs have a banded morphology, whereas pericytes have distinct soma and processes, with a morphology that changes along the vascular tree[19]. Ensheathing pericytes express αSMA and are strongly contractile with a shorter length and higher vessel coverage than the less-contractile mesh and thin-strand pericytes, which are sited on smaller vessels[18].

We first studied the number of main arteries and veins, and the length and diameters of arterioles, pre-capillaries and venule branches in imaging stacks recorded in vivo. Arterioles were defined as having a continuous layer of banded SMCs surrounding the vessel, pre-capillaries as being covered in ensheathing pericytes[18], and venules as large vessels proceeding from the mid-capillary bed and lacking continuous SMCs or substantial pericyte coverage. The artery/vein ratio was no different in HC and V1 (0.69 ± 0.25 in V1 and 0.53 ± 0.1 in HC, *t* test, *p* = 0.53, Supplementary Table 7). We measured the length of arteriole and venule branches, and of the pre-capillaries, as well as vessel diameters. The lengths and initial diameters of arterioles, venules and precapillary arterioles were equivalent in HC and V1 (Fig. 6 d, f, g), but at the transition between arteriole and precapillary (the boundary between the last SMC and the first ensheathing pericyte), vessel diameters were significantly smaller in HC than

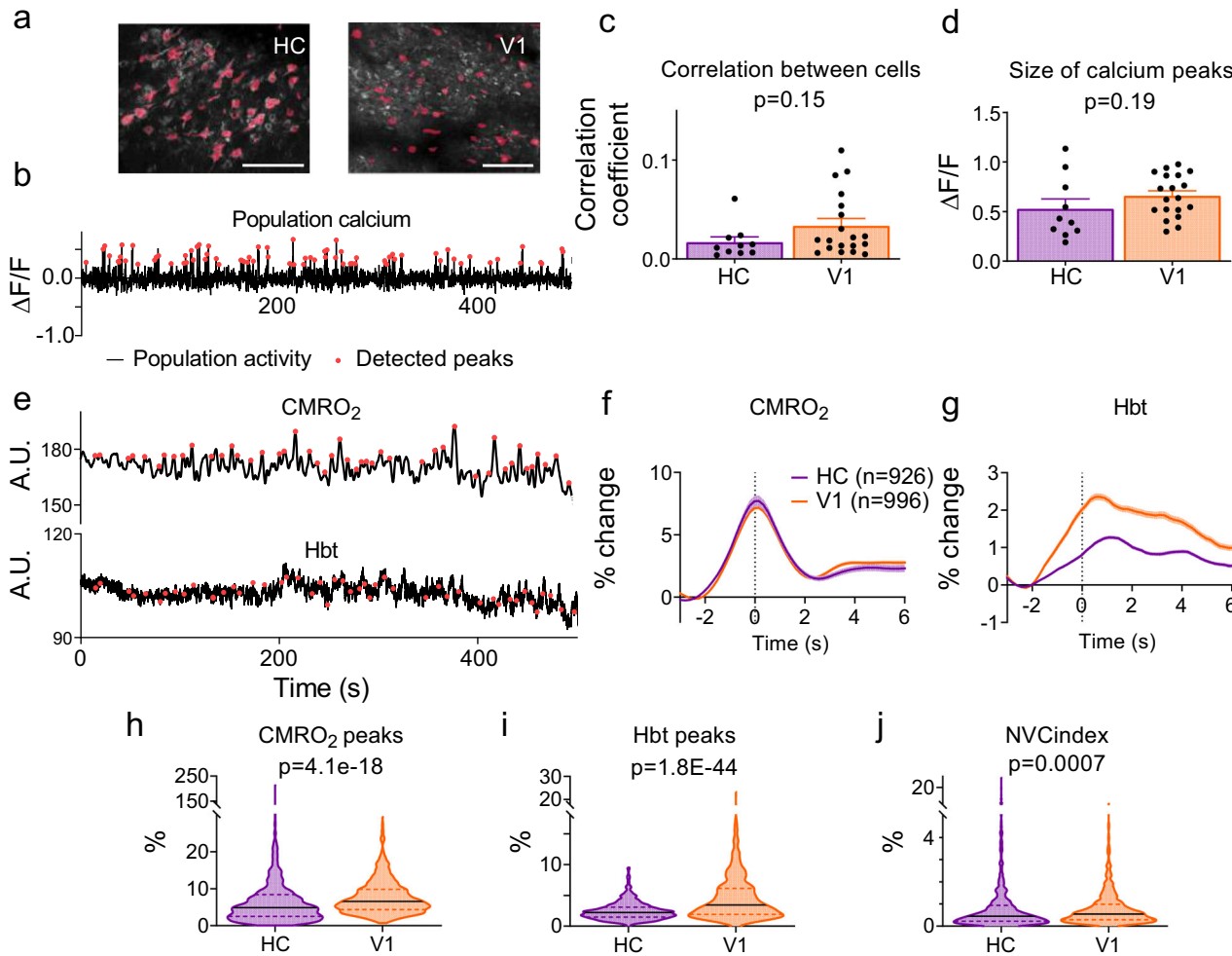

**Fig. 5 Wide-field neuronal activity patterns. a** Representative wide field-of-view recording of calcium signals (white; maximum projection across time) in HC (left, 1 of 10 independent recordings) and V1 (right, 1 of 19 independent recordings). Regions of interest (ROIs) used to measure cell activity are displayed in pink. Scale bar represents 100 μm. **b** Example neuronal calcium trace, averaged across all detected ROIs. Times when net activity peaks (>2 SD above baseline mean) are shown by red dots. Neither **c** the correlation (Pearson's $R$) between individual cells within a field-of-view was different, nor **d** the size of net calcium peaks between the HC (10 recordings from four animals, 407 cells detected in total) and V1 (19 recordings from three animals, 338 cells in total). Dots in **c** and **d** represent separate recording sessions. **e** Example $CMRO_2$ fluctuations over time and corresponding total haemoglobin (Hbt). Detected signal peaks are marked in red. **f** Averaged $CMRO_2$ peaks per region, and **g** the Hbt in response to these $CMRO_2$ peaks (HC data represent 926 peaks taken from nine animals across 37 recordings, V1 data represent 996 peaks taken from 10 animals across 46 recordings). **h** The magnitude of these $CMRO_2$ peaks was larger in V1 than HC, and **i** Hbt increased more in V1 than in HC within 5 s of an increase in $CMRO_2$. **j** The NVCindex was calculated by dividing Hbt/$CMRO_2$, and was significantly lower in HC ($n = 926$ and $n = 996$ data points examined for $CMRO_2$, Hbt and NVCindex statistical comparisons). $P$ values represent the result of Mann–Whitney $U$ tests for all data except **d** which was a result of independent sample $t$ tests (see Statistics Report Table SR3). Bar charts and traces show mean ± SEM, violin plots show median and interquartile range. Source data are provided as a Source Data file.

V1 (Fig. 6e). The spacing of pericytes across pre-capillaries and capillaries (intersoma distance: ISD) was also greater in HC than V1 (Fig. 6h).

We were better able to trace the origin of feeder arterioles deep in the HC using in vivo image stacks than confocal images of brain slices, but our resolution (when imaging whole stacks) using the former was not sufficient to detect small differences in vascular diameters and cell morphologies. We, therefore, further examined the properties of NG2-DsRed-labelled pericytes in the centre of the microvascular network using confocal microscopy of fixed brain slices, with the vascular lumen filled with a fluorescent gelatin[20] (Fig. 6i).

There was no difference in the relative numbers of mural cell types between regions (Fig. 6j). We next determined the length of mural cells in each category and the average diameter of the vessels that they surrounded. Our cell categorisations matched

published descriptions, as vessel diameter decreased and cell length increased from ensheathing to mesh to thin-strand pericytes (Fig. 6k, l). The diameter of vessels at pericyte locations was not significantly different between regions, but blood vessel diameter was smaller in HC than V1 at thin-strand pericytes on the smallest capillaries, whereas pericyte processes were longer for both mesh and thin-strand capillaries in HC than V1. Because pericyte contractility is strongest nearer the cell body[12,21], HC pericytes may be less contractile than their counterparts in V1, which could underlie the weaker neurovascular coupling in HC.

**Functional differences between vascular cells in HC and V1 suggested by different mRNA expression profiles.** In order to test for more general functional differences between vascular cells, we examined the mRNA expression profile of mural ($n =$

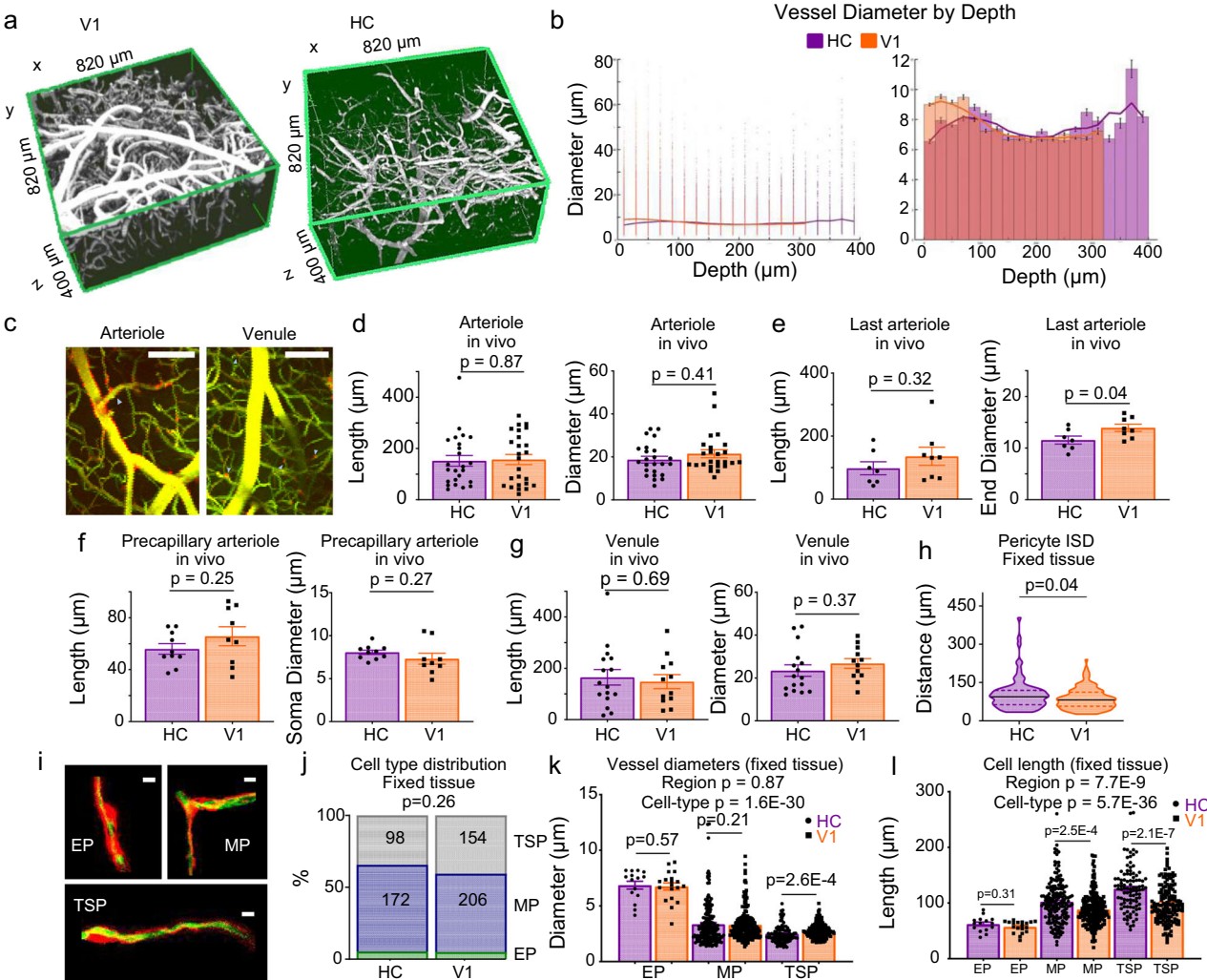

**Fig. 6 Vascular morphology across brain regions. a** Example in vivo Z-stacks of vasculature in V1 (left) and HC (CA1, right, as used in **b**). **b** Left panel: all data points for vessel diameters by depth from the pia in V1 (48458 vessels collapsed from nine stacks/nine animals, orange) or layer SO in HC (21175 vessels collapsed from 8 stacks/6 animals, purple). Right panel: histogram of average ±SEM of these vessel diameters by depth. The overlaying lines are a smoothed trace of the average diameter per bin. **c** Example arteriole (left) and venule (right) from an in vivo Z-stack of CA1 in a NG2-DsRed mouse with FITC dextran-filled vessels (representative image from $N = 3$ stacks from different mice per region). Scale bars represent 100 μm. The termination point of an arteriole was the final smooth muscle cell before an ensheathing pericyte (blue arrow). Venule termination points were the final branch before the capillary bed, identified by the presence of distinct pericytes. Dots represent individual vessels ($N = 3$ stacks from different mice per region). The diameter and length of **d** arteriole branches were not different between HC (purple, $N = 24$) and V1 (orange, $N = 25$). **e** The diameter of the vessel at the final SMC was significantly smaller in HC ($N = 7$) vs V1 ($N = 8$), despite similar vessel lengths. **f** The precapillary arterioles (with ensheathing pericytes, HC $N = 10$, V1 $N = 9$) and **g** venule branches (HC $N = 16$, V1 $N = 12$) showed no differences in length or diameter between regions. **h** Intersoma distances (ISD) between neighbouring pericytes (taken from fixed tissue) were longer in HC than V1 (89 vessels in HC and 127 in V1, from six mice). Confocal stacks of FITC-gel-filled vessels of NG2-DsRed mice were taken in HC and V1 from six mice. **i** Examples of ensheathing (EP), mesh (MP) and thin-strand (TSP) pericytes. Scale bars represent 5 μm. **j** The distribution of cell types was not different across regions (Cochran–Mantel–Haenszel 3D variant of Chi-square test). Numbers inside bars are numbers of vessels. **k** There were regional differences in vessel diameter, although post hoc comparisons revealed that these were specific to TSP locations in HC. **l** Ensheathing pericyte lengths were similar between regions, but both mid-capillary pericyte categories were longer in HC than V1. P values are from one-way ANOVAs with Welch's correction to test for effects of cell type and brain region. P values above the bars are from unpaired t tests, Mann–Whitney U tests or post hoc Mann–Whitney U tests with Holm–Bonferroni correction for multiple comparisons (see Statistics Report Tables SR4a–e). Bar charts are mean ± SEM, and statistical comparisons for **d**–**h** & **k**–**l** compared data from the individual vessels represented by the dots. Horizontal lines on violin plots show median and interquartile range. Source data are provided as a Source Data file.

83, 20 in HC) and endothelial ($n = 137$, 10 in HC) cells using the same single-cell mRNA data set as above[17] across three broad categories associated with neurovascular function: contractile machinery, ion (potassium and calcium) channels and neurovascular signalling pathways (Supplementary Tables 4 and 5). The latter category included recently identified EET and 20-HETE receptors[22,23] and, given the recent finding that endothelial

NMDA receptors can control vascular tone[24], NMDA receptor subunits.

Several genes showed differential expression between V1 and HC, all of which pointed to the vasculature in V1 being more contractile or responsive than in HC (Supplementary Figure 8; Supplementary Tables 4 and 5). Mural cells showed higher expression in V1 of the calcium channel beta subunit Cacnb4,

whereas levels of transcripts for several other ion channel subunits were significantly higher in V1 before correction for multiple comparisons, including stargazin (Cacng2, which can reduce calcium channel activation[25]), a slowly activating potassium channel (Kcnh3), and contractile proteins such as the skeletal muscle actin (Acta1) and regulators of myosin light chain phosphatase (Ppp1r12c) (Supplementary Table 4, Supplementary Figures 8a–e). In endothelial cells, several more transcripts were upregulated in V1 compared with HC (Supplementary Table 5, Supplementary Figures 8f–i), most notably the inwardly rectifying potassium channel Kir2.1 (Kcnj2), which has been shown to mediate the propagation of dilation from the capillary bed to upstream arterioles[26], the NMDA receptor subunit Grin2c, the NO receptor Gucy1a2 and prostaglandin E synthase (Ptges). These transcripts would all be expected to promote dilation of the microvasculature, and their lower expression in HC than V1 may mediate the weaker neurovascular function we observed physiologically. mRNA extraction from HC was not simply lower across all genes, as there was no difference between HC and V1 in the average number of mRNA molecules detected per cell across all genes investigated (Supplementary Table 6).

Thus, pericytes have a less-contractile morphology in HC than V1, whereas HC mural and endothelial cells are less equipped to promote vasodilation via regulation of contractility, glutamate or NO sensing, prostaglandin production or by activation of potassium channels.

**Lower oxygenation in HC may limit function**. To understand how weaker HC neurovascular functioning could affect neuronal oxygen supply, and thus ATP production, we modelled oxygen diffusion and consumption in HC and V1 (Fig. 7a). We first worked out how far brain tissue was, overall, from the nearest blood vessels in vivo (Fig. 7b, c). We then calculated the steady-state oxygen concentration in an average capillary in each region. Because the oxygen level between rather than within RBCs better reflects tissue oxygen levels[27], we used the $pO_2$ between two RBCs as our estimate of capillary $pO_2$ (see Methods). InterRBC $pO_2$ was 15 mmHg (21 μM) in V1 and 10 mmHg (14 μM) in HC. Oxygen diffusion into the tissue was then simulated (Fig. 7d), assuming varying rates of neuronal oxygen consumption corresponding to values reported in rodent tissue[28,29]. Because in some tissue, the oxygen gradient between the interRBC $pO_2$ and the tissue is flat (within measurement error), and equals the interRBC $pO_2$[27], we also ran simulations with a slightly higher capillary $pO_2$ (18 μM in HC, 22 μM in V1), which yielded the predicted interRBC $pO_2$ at the median distance between two capillaries (Supplementary Figure 9). This allows for some $O_2$ delivery directly from RBCs themselves and is still consistent with published tissue $O_2$ gradients.

The combination of the lower capillary oxygen concentration ($[O_2]$) and capillary density meant tissue oxygen levels in HC were lower than in V1 for all conditions simulated (Fig. 7e, f, h, i, k, m). To determine whether oxygen became limiting for ATP production in the tissue, we then calculated the oxygen consumption rate ($VO_2$) as a proportion of the maximum rate of oxygen consumption ($V_{max}$; Fig. 7g, j, l, m). In V1, $VO_2$ (and therefore the rate of ATP generation) occurred at over 90% of the $V_{max}$ even far from a capillary, and at the upper range of $V_{maxes}$ tested, suggesting $[O_2]$ barely limited ATP synthesis. In HC, however, whereas $VO_2$ was sustained at over 90% of the $V_{max}$ in tissue at the median distance from a capillary, in the tissue furthest (95th centile) from a capillary, $[O_2]$ dropped to concentrations that limited $VO_2$ even at low oxygen consumption rates (≥1 mM/min). In our simulations with a larger capillary $pO_2$, the decrease in $O_2$ was slightly smaller, but still limited $VO_2$

far from the capillary to 70% of $V_{max}$ when $V_{max} = 2$ mM/min (Supplementary Figure 9, vs. 60% of $V_{max}$ with capillary $pO_2$ fixed at interRBC $pO_2$, Fig. 7l). Thus, our results suggest that HC ATP production through oxidative phosphorylation is restricted in tissue furthest from a capillary.

The effect of weaker HC neurovascular coupling can be estimated by considering what happens when oxygen consumption increases but capillary oxygenation does not. Typically, neuronal activation increases net $CMRO_2$ by 20–60%[30]. In HC, increasing the $V_{max}$ for oxygen consumption from 2–3 mM/min, caused $[O_2]$ in tissue furthest from vessels to almost halve, reducing $VO_2$ from 59% to 31% of $V_{max}$ (Fig. 7k, l), whereas in V1, $VO_2$ reduced only from 93% to 90%. Thus, the decreased $SO_2$ and increased capillary spacing in HC reduce tissue $[O_2]$, limiting oxidative phosphorylation and ATP synthesis in the tissue furthest from a capillary, and this effect is exacerbated by weaker HC neurovascular coupling.

To estimate the impact of these levels of $O_2$ on tissue respiration (and by extension, function), for a $V_{max}$ of 2 mM/min (in the centre of the range of published values[29]), we estimated how much of the tissue is subject to rate-limiting $[O_2]$ by calculating profiles for $[O_2]$ and $VO_2$ when capillaries were separated by intermediate distances between the median and 95th centile values used above (Fig. 7m). This analysis suggested that low $[O_2]$ inhibits $VO_2$ by at least 10% in 30% of HC tissue, and by at least 20% in 10% of HC tissue.

## Discussion

Our data reveal, for the first time, the properties of neurovascular coupling in mouse HC in vivo and demonstrate that hippocampal vascular function is different from that in neocortex in two major ways. First, despite equivalent resting levels of oxygen consumption in HC compared with the neocortex, resting blood flow in the HC is lower than in neocortex, due both to a lower vascular density and lower RBC velocity and flux in individual capillaries. This lower energy supply leads the HC to have a lower resting blood oxygen saturation than neocortex. A simple model predicts that lower blood oxygenation and vascular density in the HC drive tissue oxygen to levels that readily become limiting for ATP generation. Second, increases in neuronal activity in the HC cause fewer and smaller dilations of local blood vessels and a smaller increase in overall blood volume, suggesting energy supply in the HC is less well-matched to fluctuations in energy demand than in neocortex. Our data suggest that neurovascular coupling is weaker in the HC because of differences in its vasculature rather than neuronal signalling properties, its pericytes having a less-contractile morphology and its vasculature showing a pattern of mRNA expression that may be less able to promote and propagate dilation. Our model predicts that these differences in vascular physiology matter for sustaining neuronal function: the lower oxygenation and vascular density in HC interact with the decreased ability to match increased oxygen use with increases in supply. This limits tissue oxygenation in the furthest regions from blood vessels so much that ATP generation, and therefore neuronal function, are much more readily restricted in HC compared with V1.

**Lower vasodilatory capacity of hippocampal vasculature limits neurovascular coupling**. Our conclusion that differences in microvascular function between HC and V1 underlie the weaker neurovascular coupling observed in HC stems from the observation that neuronal firing and neuronal and astrocytic expression of vasoactive messengers were no different between the brain areas, whereas differences were observed in the anatomy and expression profile of key proteins of the microvasculature.

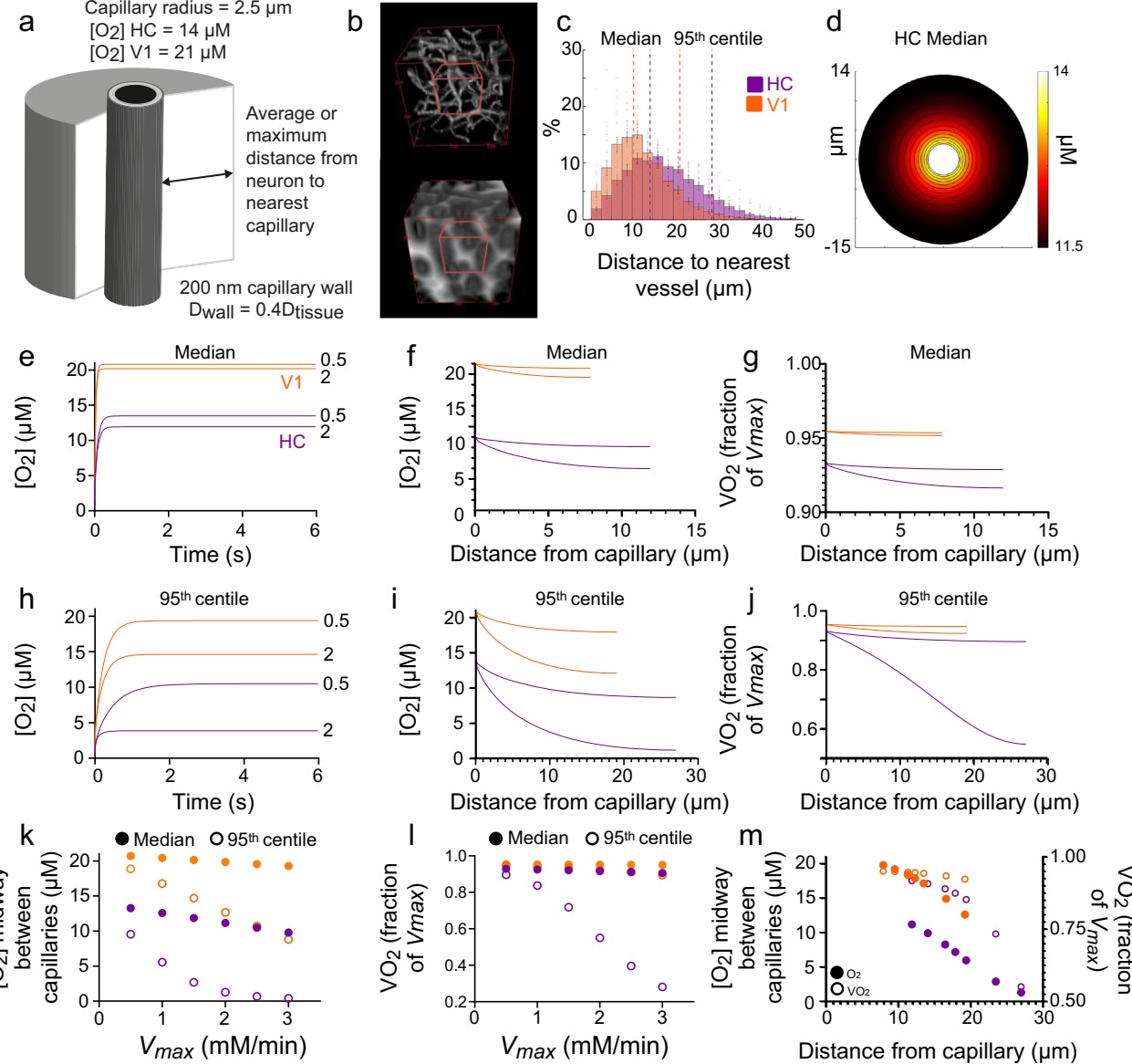

**Fig. 7 Modelling O₂ diffusion in brain tissue. a** Schematic of radial diffusion model. **b** Top: example in vivo Z-stack (250 μm³) of fluorescent dextran-filled vasculature in CA1. Bottom: 3D distance map generated from the z-stack. White pixels denote high values and black low values. Red cube shows example 100 μm³ substack used to generate **c**. **c** Average histogram of the distance of each pixel from a capillary (four mice per region, the value for each mouse representing the average of five non-overlapping substacks sampled from one Z-stack). Dotted lines mark the 50th and 95th percentiles for HC (purple) and V1 (orange), data are presented as mean ± SEM between animals. **d** Example heatmap showing oxygen diffusion from a capillary in HC separated from the next capillary by twice the median tissue distance from a capillary (14.4 μm). Simulated time courses from initial conditions of zero [O₂] for HC (purple) and V1 (orange) for tissue at **e** the median or **h** 95th centile distance from a capillary, showing that steady-state is reached by 6 s (when [O₂] profiles were extracted). Lines represent different values of $V_{max}$ as labelled. [O₂] profiles across tissue for capillary separations of twice the tissue **f** median and **i** 95th centile distance from a capillary. O₂ consumption rate as a fraction of $V_{max}$ (VO₂) for **g** median and **j** 95th centile capillary spacing conditions, calculated from oxygen profiles shown in **e** & **h**. **k** [O₂] and **l** VO₂ reached midway between capillaries for different values of $V_{max}$ in HC (purple) and V1 (orange) at median (solid symbols) or 95th centile (hollow symbols) capillary spacings. **m** [O₂] (solid symbols) and VO₂ (hollow symbols) reached midway between capillaries for capillaries spaced at twice the 50th, 60th, 70th, 80th, 90th and 95th centiles (from left to right for each range of solid or hollow dots) of tissue distance from a vessel in HC (purple) and V1 (orange). Source data are provided as a Source Data file.

First, end-arterioles and mid-capillaries were smaller in diameter in HC than V1, suggesting the microvasculature may be more resistant to flow in HC than V1, contributing to the decrease in net CBF and lower RBC velocity observed. The flow characteristics of the two vascular beds were also different. In V1, RBC velocity was faster nearer the arterial source, consistent with a shorter, lower resistance path being taken by RBCs passing through superficial neocortical layers[31]. HC flow did not show this dependence on distance from the perfusion source,

suggesting the absence of these shorter, lower resistance paths in HC.

Ensheathing pericytes initiate the dilatory response to sensory stimulation in the olfactory bulb[32], and are among the earliest to respond in the neocortex[33]. We found no regional morphological differences in ensheathing pericytes, but vessels they likely covered (>7 μm in diameter) dilated less frequently and when they did respond, dilations were smaller. Capillary (mesh and thin-strand) pericytes were longer in HC than V1, suggesting

functional differences that may include lower contractility, because pericyte contractility is greatest near the soma[21], and because the small capillaries where these mid-capillary pericytes are located dilated less frequently in the HC.

Dilations of these small capillaries are observed in several[12,33,34], but not all studies[35], and whether they are active or passive remains controversial[19,35]. Nevertheless, these dilations seem important. First, at least some of these mid-capillary pericytes express the contractile protein αSMA (though in a form that is less stable than in SMCs and ensheathing pericytes[36]). Second, their level of intracellular calcium decreases in response to neuronal activity (consistent with a relaxation of contractile machinery[32]). Finally, their dilations in olfactory bulb and neocortex seem to mediate a large proportion of the overall increase in blood flow[32,33].

Dilation signals propagate upstream up the vascular network[26,37], such that dilations in upstream vessels presumably reflect summed activity in downstream vessels. Our analyses of published single-cell RNA-Seq data[17] supported vascular functional differences between HC and V1 including a lower hippocampal expression of Kir2.1, which is critical for upstream propagation of dilation signals[26], as well as several other ion channels and neurovascular enzymes that can modulate dilation. Therefore, our data showing smaller and less-frequent dilations in larger capillaries and small arterioles in HC compared with V1 may be in part due to the decreased frequency of mid-capillary dilations (producing fewer signals summating to drive upstream dilation), a failure of upstream propagation of dilation through endothelial cells, and possibly also reduced capability for dilation in the ensheathing pericytes on these larger vessels.

**mRNA data show consistent reduced vascular function in HC compared with V1.** The mRNA data set comprises many more neurons and astrocytes than vascular cells, and more cortical than hippocampal cells. Thus, our power to detect differences at $p = 0.05$ in neuronal or astrocytic expression patterns was high (>88% with an effect size of 0.4), but much lower for endothelial (33%) or mural cells (47%). Our findings should therefore be treated with some caution and should be replicated in a dedicated experiment with larger sample sizes to verify all identified targets. Nevertheless, because *all* the positive results indicated a lack of expression in HC of proteins expected to promote dilation, despite similar levels of mRNA transcripts being detected overall in the two regions, we conclude that there is a difference in physiology between the two vascular beds.

**Regional differences in neurovascular coupling are not caused by the cranial window.** Our use of a chronic cranial window over HC allows us to measure CBF, blood oxygenation and individual vascular and neuronal signals in a region that is normally inaccessible to two-photon imaging. It requires aspiration of a column of neocortex and insertion of a cannula through which dorsal CA1 can be imaged. We have tested whether surgery affected HC function or neurovascular coupling properties (Supplementary Figure 6), or altered the structure and size of CA1 and its vascular network (Supplementary Figure 5). There was a higher expression of inflammatory markers (GFAP and Iba1) near to the cranial window in HC than in V1, but this difference disappeared 100 µm from the window. There was no difference in neurovascular coupling properties in vessels above and below 100 µm, and spatial memory was unimpaired in HC mice. The vascular density was not compromised in the surgical hemisphere, and the overall size of CA1 was also equivalent between surgical and control hemispheres. We, therefore, find no evidence that the more invasive HC cranial windows are driving the regional differences we observe.

**Low tissue oxygenation and weak neurovascular coupling: a perfect storm underlying HC vulnerability to disease.** Because the vascular density in HC is lower, each capillary supplies oxygen to a larger volume of tissue than in V1. This means more oxygen has to be extracted from each capillary to maintain the same rate of tissue oxygen consumption in HC compared with neocortex. This increase in oxygen extraction could come from an increased supply of oxygen to HC (by an increased flux of RBCs in individual HC capillaries), or by extracting more oxygen from each RBC. An increased flux would better maintain oxygen levels in the blood, but our data suggest that in fact oxygen consumption is sustained by increasing oxygen extraction from individual slower-moving RBCs, decreasing blood oxygen saturation in HC.

To our knowledge, this is the first time hippocampal and neocortical oxygenation have been directly compared, and the first time hippocampal oxygenation has been measured in a sealed skull (so the tissue is not exposed to atmospheric oxygen) in awake animals. Our estimates of capillary $[O_2]$ are at the lower end of previously observed values from anaesthetised preparations (14 µM in HC and 21 µM in V1, compared with 16 µM in HC[38] and 56 µM in cortex[39]), which is as expected because oxygen consumption rates are higher and $pO_2$ values are lower in awake animals[40,41], and are within the range reported in the sensory cortex of awake mice by directly measuring $[O_2]$ with a phosphorescent probe[41]. As we are not able to measure oxygen in individual capillaries, our results reflect average blood oxygen levels. In the neocortex, individual capillary oxygen levels, and corresponding tissue supply radii, vary quite widely, such that upstream vessels with higher oxygen levels can supply oxygen to tissue that is physically closer to capillaries with lower blood oxygenation[42]. The impact of this underlying heterogeneity of oxygenation on our results is not clear: our use of an average may be underestimating microvascular $sO_2$[42], but any such error is likely to be smaller in HC than V1, as the lower vascular density will reduce the chance that an upstream vessel's supply radius will overlap with that of a downstream, less oxygenated capillary.

Our model predicts very different tissue $[O_2]$ in V1 and HC. In V1, $[O_2]$ gradients are very shallow, and the oxygen consumption rate by oxidative phosphorylation (i.e., the rate of ATP synthesis) is maintained at >90% of the maximum throughout the tissue. However, in HC the lower vascular density and lower oxygenation mean that $[O_2]$ readily becomes limiting for ATP synthesis for physiological $V_{max}$ values (1–3 mM/min[29]). An inhibition in the rate of oxidative phosphorylation of at least 20% likely occurs in around a tenth of HC volume. Furthermore, when oxygen use outstrips supply, oxygen consumption is barely affected in V1 but is reduced to as little as 30% of the $V_{max}$ in 5% of HC tissue. Physiologically, HC seems adapted to deal with these conditions: it can sustain neuronal function despite oxygen levels that could be considered hypoxic. Decreases in synaptic function occur in vitro after induction of hypoxic conditions using higher $[O_2]$ than we estimate to be normoxic for HC (e.g., 20 mmHg/28 µM[43], compared with <14 µM here) but, in fact, such responses to hypoxia are likely to be caused by the decrease in, rather than the absolute concentration of $O_2$[44]. But while HC may cope with low $[O_2]$ physiologically, its lower oxygenation and weaker ability to match oxygen demand to supply may contribute towards its vulnerability to hypoxia and conditions where cerebral blood flow decreases, such as ischaemia and Alzheimer's disease. Further decreases in oxygen availability caused by these conditions will produce a larger reduction in ATP synthesis over a larger volume of tissue in HC than in neocortex, because it already exists in a state where $O_2$ levels are limiting. However, interventions that boost HC oxygenation may therefore prove particularly beneficial. Indeed, a drug that boosts hippocampal blood flow (nilvedipine[45]) benefitted a subgroup of patients with mild

Alzheimer's disease[46], suggesting that insufficient oxygen delivery to HC might be a key early factor in Alzheimer's disease in some individuals, and that improving oxygen delivery may be therapeutic.

**Interpreting BOLD**. Our results also illuminate why BOLD signals in HC were found to be unreliably coupled to neuronal activity[9,10]. In HC, smaller and less-frequent vascular dilations in response to local changes in neuronal activity will produce smaller positive BOLD signals. BOLD signals are therefore a less-sensitive measure of neuronal activity in HC than in neocortex. Simple experimental designs that compare the degree of activation across the brain could therefore erroneously conclude lower HC activation than cortex even when activity levels are the same. Analyses that test for specific patterns of activity, such as correlations of voxels with a behavioural measure or cognitive model of interest, will be less affected by the different neurovascular coupling properties in HC and V1[47], but the relative insensitivity may nevertheless lead to more failures to detect subtle effects in HC than in neocortex[48].

Our work suggests HC physiological and pathophysiological functioning is shaped (and limited) by its vasculature, an insight that will aid understanding of disease states and human imaging studies. Further work should directly test in which situations hippocampal oxygen availability limits its function and whether boosting oxygen availability by increasing blood flow in these conditions can preserve hippocampal function. This will help guide therapeutic strategies in conditions such as Alzheimer's disease, where boosting hippocampal blood flow in some, but not all people, can be therapeutically effective.

## Methods

**Animal procedures**. All experimental procedures were conducted in accordance with the 1986 Animal (Scientific Procedures) Act, with the approval of the UK Home Office and the University of Sussex animal welfare ethical review board. All experiments used mice with a C57BL/6 J background of either sex, which were either wild types (six in total, four males, two females) or expressed GCaMP6f under the control of the Thy1 promoter (C57BL/6J-Tg(Thy1-GCaMP6f) GP5.5Dkim/J[13]; 30 in total, 15 males, 15 females) and/or DsRed under the control of the NG2 promoter (NG2DsRedBAC[49]; 17 total, nine males, eight females). Mice were housed in a 12 h reverse dark/light cycle environment at a temperature of 22 °C, with food and water freely available.

*Surgery for cranial window placement:* All mice used for in vivo imaging experiments underwent the following surgical procedure under isoflurane anaesthesia (maintained between 0.8 and 2%). The mouse was secured on a stereotaxic frame with a head mount (Kopf). At the beginning of the surgery mice were subcutaneously injected with 2.4 µl/g of dexamethasone (2 mg/ml), 400 µl of saline and 1.6 µl/g of buprenorphine (0.3 mg/ml, diluted 1:10 with saline) to reduce inflammation, dehydration and post-operative pain, respectively. The temperature was maintained at 37 °C throughout using a homeothermic blanket (PhysioSuite, Kent Scientific Corporation). First, the scalp and underlying thin periosteum layer were removed across the entire dorsal skull surface, and a 3 mm circle overlaying the V1 or HC was marked. Scratches were made in the exposed skull using a scalpel to increase the surface area for bonding with the head plate. The skin around the exposed skull was sealed using dripping tissue adhesive (3 M VetBond). The mouse was then tilted on the head mount so that the area marked for the craniotomy laid flat. Black dental cement (Unifast Powder mixed with black ink (1:15 w/w) and Unifast Liquid) was applied to all areas of the exposed skull, except the marked craniotomy region and its immediate surround. A custom-made stainless-steel head plate was placed over the dental cement and left for a few minutes until dry. Next, a craniotomy was performed over the previously marked area using a dental drill (Fine Science Tools, burr size 007), after which the underlying dura was carefully removed. For V1 surgeries, an optical window (made from two 3 mm glass coverslips and a 5 mm glass coverslip (Harvard Apparatus) bonded together with optical adhesive (Norland Products, Inc)) was placed into the craniotomy and secured using dripping tissue adhesive and dental cement[50]. For hippocampal surgeries, ~1.3 mm of cortex was aspirated (New Askir 30, CA-MI Srl) until the striations of the corpus callosum (just above CA1 HC) were visible[11]. A 3 mm round stainless-steel cannula (2.4 mm ID, 3 mm OD, 1.5 mm height, Coopers Needle Works Ltd) with a 3 mm glass coverslip (Harvard Apparatus) attached using optical adhesive was inserted into the craniotomy and secured with dripping tissue adhesive and dental cement (secured at the skull surface, above which ~0.2 mm of cannula protruded into the dental cement). The mouse was then

injected with 5 µl/g of meloxicam (5 mg/ml, solution for injection, diluted 1:10 with saline) subcutaneously to further aid with post-operative pain relief. Finally, for both surgery types, two rubber rings were secured on top of the head plate with dental cement to allow for two-photon imaging using a water-based objective. The mouse was removed from the head mount attached to the isoflurane machine and placed into a heat box (37 °C) to recover, before being singly housed in a recovery cage. Post-surgery meloxicam (200 µl, 1.5 mg/ml) was administered orally in the food for 3 days for additional pain relief during recovery.

*Habituation:* Prior to imaging, and following a post-surgery recovery period of at least one week, mice were habituated to head fixation on a polystyrene cylinder daily (over the course of 5 days). For the first habituation session, mice were handled by the experimenter and allowed to explore the cylinder freely without head fixation. The following sessions consisted of head-fixing the mouse atop of the cylinder for gradually increasing time periods (increasing from 1 minute in session 2–15 min in session 5).

**In vivo imaging**. *Experimental set-up:* Mice were imaged following a suitable recovery period post-surgery (average number of recovery days in HC: 43 ± 6, and V1: 35 ± 4). For imaging, mice were head-fixed atop of a polystyrene cylinder, fitted with a Kuebler rotary encoder (4096 steps/revolution) to measure locomotion, underneath a two-photon microscope (Scientifica) or combined laser doppler flowmetry/haemoglobin spectroscopy probe (VMS-Oxy, Moor Instruments; Fig. 1c). In front of the mice were two computer screens which, when required, displayed a virtual reality maze (custom-designed in ViRMEn[51], MATLAB; for HC surgery mice; Fig. 1e) or drifting gratings (PsychoPy, 315° orientation refreshed at 60 frames per second; for V1 surgery mice; Fig. 1f).

*Combined laser doppler flowmetry/haemoglobin spectroscopy (Oxy-CBF probe):* A combined laser doppler flowmetry/haemoglobin spectroscopy probe (Moor Instruments, VMS-Oxy) allowed for monitoring (at 40 Hz) of the following net haemodynamic measures via VMS4.0 software: blood flow (flux), speed, oxygenated haemoglobin (HbO), deoxygenated haemoglobin (Hbr), total haemoglobin (Hbt, Hbt = HbO + Hbr) and oxygen saturation (SO₂, SO₂ = HbO/ (HbO + Hbr)). Except for SO₂, which is expressed as a percentage of 100% saturation, these measurements are expressed in arbitrary units as we cannot calculate absolute flow rates and haemoglobin concentrations from these data. Experiments typically lasted 0.5–1 h. The CMRO₂ over time (t) was estimated from these variables using Eq. (1)[52]:

$$CMRO_2(t) = CBF(t) \times \frac{Hbr(t)}{Hbt(t)} \qquad (1)$$

Laser doppler flowmetry is commonly used to compare relative not absolute differences in flow, but we reasoned that if we could discount potential artefactual contributions to the LDF signal[53], differences in the signal between regions might be informative about regional differences in CBF. First, we checked whether probe placement across sessions affected signal variability. We plotted the responses recorded on different sessions within the same animal and between animals (Supplementary Figure 10). We then compared the standard deviation for each mouse (with ≥2 recording sessions, N = 6 mice for HC, N = 7 mice for V1) between measurements and across sessions and compared variability between regions. There was no difference in inter-session standard deviation between brain regions for any of the haemodynamic measures (independent sample t tests; p > 0.19). Variability in CBF was greater than that of haemoglobin absorbance recorded using spectroscopy, possibly reflecting differences in probe placement (e.g., over large pial vessels) and spatial variability in perfusion. Nevertheless, there were no significant differences between recordings obtained within a region (V1 or HC), whereas the difference between regions was significant (Fig. 2b).

We also checked that fluorescence signals from the brain did not interfere with the haemodynamic measurements and reached zero in the absence of blood flow following the death of the subject (Supplementary Figure 10). Finally, we considered the possible effects of the large sampling volume of our LDF probe on our results. It has a 500 µm separation between the transmission and detection fibres and a wavelength of 780 nm, so samples deep in the tissue, with a median signal depth of around 0.9 mm and a maximum depth of 2 mm[54]. Thus, some of the signal recorded in V1 comes from underlying corpus callosum and HC, and some of that recorded from HC originates in the thalamus. However, it is highly unlikely that signal from these areas could produce the regional differences we report here, as corpus callosum and HC have low vascular densities so cannot be boosting the apparent V1 CBF signal, whereas the thalamus has a high vascular density[15] so will not be artificially reducing the signal attributed to HC.

*Analysis of oxy-CBF probe data:* The oxy-CBF probe data (flux, speed, SO₂, HbO, Hbr, Hbt or CMRO₂) was categorised as occurring during periods of rest (i.e., no locomotion and no visual screen presentations) or not. The average value of each haemodynamic parameter was calculated per animal (across sessions) during these stationary periods (Fig. 2a–c).

Peaks in the CMRO₂ trace were detected (as specified in the section below: 'Analysis of two-photon microscopy data', Fig. 5f, h), and those peaks which were more than two standard deviations larger than their preceding baseline were extracted, along with their corresponding Hbt traces (Fig. 5g, i).

*Two-photon microscopy:* In order to visualise blood vessels during imaging mice were injected with 2.5% (w/v) Texas Red Dextran dissolved in saline (70 kDa via tail vein or 3 kDa subcutaneously, Fisher Scientific) alone or in combination with

fluorescein Dextran dissolved in saline (70 kDa via tail vein, Fisher Scientific). All blood vessels recorded local to neuronal calcium activity were capillaries, precapillary arterioles or arteriole branches to ensure that similar types of vessels were selected between the regions. High-resolution imaging of vessels and calcium from excitatory neurons was performed with a commercial two-photon microscope (Scientifica), a high numerical aperture water-dipping objective (×20 aperture, XLUMPlanFL N, Olympus or ×16 aperture, LWD, Nikon), and a Chameleon Vision II Ti:Sapphire laser (Coherent). Tissue was excited at 940 nm, and the emitted light was filtered to collect red and green light from Texas Red Dextran (vessel lumen) and Thy1-GCaMP6f (excitatory neurons), respectively. Imaging sessions were recorded in SciScan software (SciScan v1.2.1, Scientifica). Typically, imaging occurred up to a depth of ~500 μm from the surface. Imaging sessions included both wide field-of-view recordings of network neuronal calcium activity (256 × 256 pixels, speed range 6.10–15.26 Hz, speed average 7.75 Hz, pixel size range 1.35–2.56 μm, pixel size average 1.80 μm), and smaller field-of-view recordings of individual blood vessels and local neuronal calcium signalling (256 × 256 pixels, speed range 3.05–7.63 Hz, speed average 6.64 Hz; pixel size range 0.15–0.63 μm, pixel size average 0.23 μm). High speed line scans (speed range 413–2959 Hz, speed average 1092 Hz; pixel size range 0.15–0.46 μm, pixel size average 0.20 μm) were also taken from individual blood vessels to track diameter, RBC velocity, RBC flux and haematocrit from capillaries, and sometimes the line scan path also passed through a nearby neuron (for concurrent calcium readings). Vessels and local calcium were imaged from each of the distinct layers of CA1 (stratum oriens, stratum pyramidale, stratum radiatum, stratum lacunosum–moleculare) and V1 (L1, L2/3, L4). The layers in CA1 were clearly distinguishable by the changing morphology of the pyramidal neurons over increasing depths within this region[55] (Fig. 1b). The layers of V1 were distinguished based on the distance of the imaging site from the pial vessels[56]. Two-photon imaging experiments typically lasted 1–3 h.

*Analysis of two-photon microscopy data*: Excitatory calcium activity was first registered in ImageJ to correct motion artefacts, before being extracted using the CellSort package in MATLAB[57]. CellSort identified regions of interest (ROIs) over multiple active individual cells (both soma and NP). Any ROIs that overlapped the same cell were either merged or removed, as appropriate. For most analyses, net activity in a recording was calculated across ROIs, except when correlations between individual ROIs were calculated (Fig. 5c).

Vessel diameter (XY movies and line scans) and RBC velocity, flux and haematocrit (line scans) data were registered in ImageJ to remove motion artefacts and despeckled. The vascular lumen was bright due to the dye injection, whereas the RBCs within the vessel and background around the vessel were dark. Custom MATLAB scripts were used to extract the diameter along the vessel branch(es), or the RBC velocity, flux or haematocrit. In brief, for each frame of XY movies the vessel was skeletonised, meaning its length was eroded to generate a single-pixel diameter line running through the entire length of the vessel at its centre. The intensity profile of a line perpendicular to this vessel axis was taken at every other pixel along the length of this skeleton, averaged across a running window of five pixels. The diameter was calculated from the full-width half maximum of this intensity profile. For RBC velocity data from line scans, we adapted freely available code[58]. In short, the angle of the shadow cast by the RBCs travelling through the vessel was used to calculate the speed at which they travelled over successive 40 ms (time, $t$) blocks (with an overlap of 10 ms in time (i.e. $t/4$) between blocks). For RBC flux data from line scans, the number of RBCs per second was counted using an automated script in MATLAB. In brief, two-photon recordings were binarized. An average intensity curve was generated for the centre of binarized scan, and groups of dark pixels (RBCs) fell below a minimum threshold, which allowed them to be counted within a 250 ms time window. The accuracy of the automated RBC counter was also verified by checking for consistency with a subset of the data which was manually counted. For calculating haematocrit values within a capillary over time, again line scan recordings were binarized, and the percentage of overall dark space (i.e., containing RBCs) was calculated over successive 40 ms blocks with an overlap of 10 ms in time.

To detect peaks in the neuronal calcium signal local to a blood vessel, the calcium trace was first normalised across the entire recording (e.g., Figure 3b) using Eq. (2), before identifying putative events as peaks that were >10% of the maximum. The data around each calcium peak was then extracted from the raw (unnormalized) trace (5 s before and 10 s after), and these peaks were then normalised to their own baseline (5 s preceding peak) using Eq. (3). Changes in calcium fluorescence from baseline (e.g., Figure 3d, j) are presented as $\Delta F/F$. Delta is conventionally used to denote a change from the initial state, and so delta $F$ over $F$ compares the change of intensity after activation to that at resting baseline.

$$\Delta F/F = \frac{F - F_{\min}}{F_{\max} - F_{\min}} \qquad (2)$$

$$\Delta F/F = \frac{F - F_0}{\mathrm{abs}(F_0)} \qquad (3)$$

$F$ represents the entire fluorescence trace, and $F_0$ represents the baseline fluorescence period.

Events were excluded if their peaks were smaller than two standard deviations above the mean value of the baseline. Vessel traces were extracted across the times of these calcium events, or after bootstrapping by shuffling each of the vessel traces over 100 iterations across time using the MATLAB function randperm, so they were no longer aligned to calcium events. For every calcium event, both the corresponding real vessel diameter trace and shuffled vessel diameter trace were classified as being responsive or not. Responsive diameter traces were those where dilations occurred for >0.5 s, within 5 s of the calcium event, that were >1 standard deviation above baseline.

*Behavioural testing*: Animals that had previously undergone HC or V1 surgery for in vivo imaging underwent a simple hippocampal-dependent object location memory task to test spatial memory[59] (Supplementary Figure 6). The mouse was presented with two objects in a training environment for 10 min, before being removed from the environment. After 5 min in the home cage, the mouse was then returned to the environment, after one of the objects had been moved to a novel location. The time spent exploring both the familiar and novel object locations was assessed by a blinded observer hand-scoring the time per object. Any mice which spent less than total 5 s exploring the test environment were excluded from the analysis.

*Ex vivo imaging. Transcardial perfusion and gel-filling the vasculature*: Adult mice were terminally anaesthetised with pentobarbital. A transcardial perfusion was performed using 4% paraformaldehyde (PFA) in 0.1 M phosphate buffer solution (PBS). In cases where the vasculature was labelled, this was also followed by the perfusion of 5% gelatin containing 0.2% fluorescein isothiocyanate (FITC)-conjugated albumin. Mice were then chilled on ice for at least 30 min, before brains were extracted and fixed in 4% PFA overnight. The brains were subsequently transferred to a 30% sucrose solution in PBS with 0.1% sodium azide for a minimum of 3 days, after which they were sliced (200 μm slices) on a vibratome (Leica), and stored in the fridge (at 4 °C) in PBS with 0.1% sodium azide, before immunohistochemical labelling and/or mounting.

*Immunohistochemical labelling and slice mounting*: Adult NG2-DsRed mice were used for pericyte morphology and vascular network analyses (Figs. 2 and 6), as pericytes and SMCs were transgenically labelled in red. Vessel type could therefore be identified based on the smooth muscle or pericyte cell morphology. Adult wild-type or NG2-DsRed mice, which had previously undergone CA1 surgery for two-photon imaging, were used for vascular network analysis on the surgical versus the control hemisphere, and adult Thy1-GCaMP6f mice, which had also previously undergone CA1 surgery, were used for the analysis of CA1 area on surgical and control hemispheres (i.e., following cannula insertion/possible compression, Supplementary Figure 5).

For analysis of inflammation (Supplementary Figure 6) mice that had previously undergone HC or V1 surgery for in vivo imaging were used, and active astrocytes were labelled for GFAP and microglia labelled for Iba1 using immunohistochemistry. First, brain slices were washed in 1× PBS whilst being shaken for three cycles (10 min per wash). The slices were then blocked in 5% normal goat serum and 0.3% Triton X-100 in 1× PBS for an hour. Slices were incubated in the relevant primary antibodies (chicken anti-GFAP primary antibody; Abcam, ab4674, 1:500 dilution or rabbit anti-Iba1, WAKO, 019-19741, 1:600 dilution) for 36 h at 4 °C. Slices were washed three times in PBS, before being incubated in the relevant secondary antibody for 24 h at 4 °C (Alexa 647 goat anti-chicken, Abcam, 1:500 dilution or Alexa 647 goat anti-rabbit, Abcam, 1:500 dilution). Gel-filled and/or immunohistochemically labelled slices were washed for a final three cycles in PBS, before being mounted onto slides in Vectashield Hardset mounting medium and stored at 4 °C before imaging.

*Confocal imaging*: Imaging was performed on a confocal microscope (Leica SP8, collected using LAS X software) using a ×20 objective (HC PL APO CS2, 20×/0.75, dry). Continuous wave lasers with ~488, ~543 and ~633 nm excitation wavelengths were used for FITC, DsRed and Alexa 647, respectively. Images were collected with dimensions of 1024 by 1024 pixels (with a lateral resolution of 0.57 μm per pixel), averaged 4–6 times per line, and were viewed and analysed in ImageJ software.

*Analysing pericyte and vessel morphology*: For the analysis of individual pericyte cell morphology, composite z-stacks of brain slices with DsRed-labelled pericytes and FITC-albumin labelled blood vessels were used to identify the span along the vessel length of individual pericyte cells, and the vessel diameter measured at the location of the DsRed-positive cell body. The cell was then categorised as being an ensheathing, mesh or thin-strand pericyte, based on its distinct morphology[18] (see Fig. 6i for example images).

To calculate capillary density (Fig. 2d, e, Supplementary Figure 5) and vessel diameter by depth from the imaging surface (Fig. 6b), an in-house ImageJ macro was used to skeletonise 3D stacks of the vasculature, determine all the skeleton coordinates and branch points, and then create distance maps of the vasculature. The capillary density was then calculated by dividing the total length of the vessel skeleton within the region by the tissue volume imaged. The vessel radii were found by determining the grey value from the distance map at each skeleton point. The coordinates of the vessel skeleton were also used to determine the vessel's depth in the tissue.

*Analysing post-surgery inflammation levels*: For each GFAP- or Iba1-labelled brain slice, the z-stack was 2D maximum-projected for both the surgical and non-surgical hemisphere. In each hemisphere, three linear 500 μm ROIs were placed perpendicularly to the cranial window (or equivalent locations in the non-surgical hemisphere). The intensity profiles along these three lines were then averaged and normalised to the average value in the non-surgical hemisphere of the same slice.

*Analysing the area of CA1 in surgical and non-surgical hemispheres*: Wide-field images were taken of the entire brain slice by tile scanning (Supplementary Figure 5). In brief, two positions were marked around the outer edges of the brain slice, one in the upper left corner and one the lower right corner. A rectangle with multiple grids was created between these two positions, and a motorised stage was used to move the sample and create a tiled scan of the whole slice. 2D images were captured at the focal point for each grid, with blending (statistical and linear) between grid images. An ImageJ plugin[60] was then used to stitch the grid images together in post processing. For analysis of the area and aspect ratio of CA1 in surgical and control hemispheres, the freehand tool was used to draw around CA1, and calibrated area and bounding rectangle measurements were generated. The aspect ratio was calculated by dividing the height of the selection by its width, meaning larger values represented a taller/narrower shape, and smaller values a wider/shorter shape.

*Single-cell RNA-Seq data*. The raw single-cell RNA-Seq data were taken from a freely available online database[17] (http://linnarssonlab.org/cortex/). The authors used quantitative single-cell RNA-Seq to perform a molecular census of cells in somatosensory cortex (S1) and HC (CA1) and categorise cells based on their transcriptome. We compared expression of components of neurovascular signalling pathways, ion channels or contractile machinery in vascular cells (pericytes, SMCs, endothelial cells; Supplementary Figure 8) and vasodilatory signalling pathways in pyramidal cells, interneurons, and astrocytes (Fig. 4). The levels of target transcripts in HC and cortex were compared using multiple *t* tests in RStudio, with a Holm–Bonferroni correction for multiple comparisons (see Supplementary Tables 1–4).

*Modelling oxygen concentrations in tissue. Estimating capillary oxygen concentrations*: Our in vivo oxy-CBF probe measurements give us the saturation of oxygen ($SO_2$) in the blood. From $SO_2$, we calculated the partial pressure ($pO_2$) of oxygen in RBCs using the haemoglobin oxygen dissociation curve for C57/BL6 mice (the background of our experimental mice), with a Hill coefficient $h$ of 2.59 and a P50 of 40.2 mmHg[61] (4).

$$sO_2 = \frac{pO_2^h}{pO_2^h + P50^h} \qquad (4)$$

This yielded RBC $pO_2$ values of 30 mmHg (42 μM at 37 °C) in HC and 43 mmHg (60 μM) in V1. Micromolar concentrations of oxygen in tissue at 37 °C were calculated from partial pressures using a Henry's Law constant of $1.3 \times 10^{-5}$ mol/(m$^3$ Pa) at standard temperature and pressure, and a temperature conversion factor of 1500 K[62]. Our calculated RBC oxygen levels were consistent with those measured with a phosphorescent probe in somatosensory cortical and olfactory bulb RBCs in an awake mouse (30–100 mmHg)[27,41], in experiments that also measured $pO_2$ between RBCs (interRBC $pO_2$). We then estimated the interRBC $pO_2$ for our experiments by using those interRBC phosphorescent probe measurements that had the same $pO_2$ as we observed[27,41]. This yielded an estimate of interRBC [$O_2$] in V1 of 15 mmHg/21 μM and in HC of 10 mmHg/14 μM. Because tissue $pO_2$ equilibrates with interRBC $pO_2$ rather than RBC $pO_2$[27], we used these estimates of interRBC [$O_2$] as the capillary [$O_2$] in our model. Indeed if we ran our simulations using RBC $pO_2$ as the capillary [$O_2$], tissue [$O_2$] estimates were impossibly much higher than the expected interRBC [$O_2$] values (reaching 37 μM in HC and 58 μM in V1 with a $V_{max}$ of 3 mM/min), confirming that interRBC rather than intra RBC [$O_2$] levels are likely to be the main drivers of tissue [$O_2$].

*Calculating capillary spacings*: Using ImageJ (Exact Euclidean Distance Transform (3D) plugin) we calculated a 3D distance map from a smoothed and binarized in vivo z-stack of the vascular network in each brain region. We extracted five substacks of $100 \times 100 \times 100$ μm from each distance map. The distribution of the distances of pixels from the nearest vessel was then extracted and averaged across all substacks from a given larger z-stack (giving an average distance per animal). Histograms of these data were averaged across five stacks per brain region (Fig. 7c) and the 50th, 95th and intermediate centile distances from a capillary were calculated (Fig. 7e–l, m). For a given bar in the histogram, contributing pixels are either at a midpoint between two capillaries, or are closer to one capillary than another. Those pixels above the 95th centile must lie more-or-less at the midpoint between vessels, whereas many pixels at the median distance from the vessel will be nearer to one vessel than another. However [$O_2$] at each distance from a vessel would be larger if that point is at the midpoint between two vessels compared with being nearer to one vessel than another (and therefore oxygen from that vessel is having to feed a larger tissue volume). For the diffusion model below, we therefore used the distribution of tissue distances from a capillary as a conservative estimate of capillary separation, meaning that our estimates of levels of oxygen in the tissue are, if anything, slightly overestimated.

*Diffusion model*: We used the Heat Transfer functions in the Partial Differential Equation Toolbox of MATLAB to numerically solve Fick's diffusion equation for radial geometries with Michaelis–Menten consumption of oxygen through oxidative phosphorylation (Eq. (5)):

$$\frac{dC}{dt} = \frac{1}{r}\frac{\partial}{\partial r}\left(rD\frac{\partial C}{\partial r}\right) - \frac{V_{max}C}{C + K_m} \qquad (5)$$

where $r$ is the distance from the centre of a capillary (of radius 2.5 μm; Fig. 2f), $C$ is the concentration of $O_2$, $D$ is its diffusion coefficient in brain at 37 °C ($9.24 \times 10^{-8}$

m$^2$/min[63]), $K_m = 1$ μm is the $EC_{50}$ for $O_2$ activating oxidative phosphorylation[64], and $V_{max}$ is the maximum rate of oxidative phosphorylation at saturating [$O_2$]. The initial conditions were $C(r) = 0$, and the boundary conditions fixed [$O_2$] at the edge of the capillary ($r_{2.5}$) to be the calculated interRBC [$O_2$] for each brain region and at the midway point between two capillaries ($r_{max}$; $dC/dr = 0$). Because the basement membrane of the vessel wall may form a diffusion barrier for oxygen supply to the tissue, reducing the effective diffusion coefficient by up to 40%[65], $D$ was reduced by 40% lower value within a capillary wall of 200 nm (the maximum of the reported range[66], $r = 2.5$–2.7 μm).

The equation was solved over values for $r$ from 2.5 μm up to the average tissue distances from a capillary as calculated above (median, 95th or intermediate centiles) for each brain region.

*Oxygen consumption*: Values for $V_{max}$ (0.5–3 mM/min) were chosen based on the range of brain oxygen consumption rates previously reported[29]. Of this range, values of 1–2 mM/min may be the most physiological, being reported from recent experiments measuring the oxygen gradient using phosphorescent lifetime imaging, e.g., [28] (rather than an invasive electrode) and matching the whole brain averaged oxygen consumption rate[67,68]. However, previous measurements in rodents were done under anaesthesia, so rates may be higher in our awake system.

*Statistical analysis*. The sample size was determined by power analysis using G * Power. All data are presented as mean ± SEM or individual data points unless otherwise stated. Data were tested for equal variance between groups by Kruskal–Wallis or Mann–Whitney test. Two-group comparisons were evaluated by independent sample Student *t* tests, or in cases of unequal variance between groups either Welch's independent sample *t* tests (for mRNA data sets) or Mann–Whitney *U* tests (for two-photon experiments). Paired comparisons were evaluated using the Wilcoxon test (nonparametric data, for comparison between surgical and control hemispheres). Multiple comparisons were assessed using one-way analysis of variance (ANOVA) to compare means, multifactorial ANOVAs to assess for interactions, and in cases where the data violated the assumptions of the ANOVA due to uneven variance between groups, we ran one-way ANOVAs with Welch's correction. Post hoc comparisons were compared with Bonferroni tests when data had equal variances and Games–Howell tests in cases of unequal variance. For the multiple *t* test comparisons, a procedure equivalent to the Holm–Bonferroni correction was applied to calculate an adjusted *p* value. This correction consisted of ranking the *p* values outputted from *t* tests in ascending order and then multiplying the lowest *p* value by the number of comparisons, the second-lowest *p* value by the number of comparisons minus one, and so on, until comparisons were no longer significant at $p = 0.05$. Simple linear regressions were used to compare the slope through a data set to a flat (zero) line, and an analysis of covariance to compare the intercepts and slopes between two fitted regression lines. Chi-square tests ($2 \times 2$ contingency with Fisher's exact significance test, or 3D Cochran–Mantel–Haenszel tests with Pearson's R multiple post hoc comparisons) were used to compute *p* values from a contingency table. The threshold for statistical significance was set at $p \leq 0.05$, and for a trend at $p \leq 0.1$. Statistical analyses were conducted using SPSS, RStudio or Graphpad Prism and figures were created using GraphPad Prism. For further detail, a statistics report is also provided.

**Reporting summary**. Further information on research design is available in the Nature Research Reporting Summary linked to this article.

## Data availability
The mRNA data presented in Fig. 4 and Supplementary Figure 8 were taken from: http://linnarssonlab.org/cortex/. For all main figures, and the supplementary barry figures with unique data sets, the extracted traces from the raw image files are available as .xlxs and .mat files on Figshare[69]. The raw image files are stored in our Dropbox owing to their large size, and are available from the corresponding author upon request. Source data are provided with this paper.

## Code availability
The custom code used for data analysis is available as a Github repository, and has been published via Zenodo[70].

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

## Acknowledgements

This work was supported by an MRC Project Grant (MR/S026495/1), the Royal Society (RGS/R1/191203), an MRC Discovery Award (MC_PC_15071) and an Academy of Medical Sciences/Wellcome Trust Springboard Award to C.N.H., the Alzheimer's Society for O.B.'s studentship and Sussex Neuroscience Ph.D. studentships for D.C. and D.M.G, Sussex University Research Development funding to H.C. and C.N.H. and University of Sussex, School of Psychology funding to C.N.H.

## Author contributions

K.S., L.B., K.B., D.M.G. and D.C. collected data for the studies. K.S, H.C. and C.N.H. designed and analysed the studies. K.S, D.M.G., O.B., D.C. and C.N.H. wrote scripts to analyse the data. K.S. and C.N.H. wrote the manuscript with feedback from all authors.

## Competing interests

The authors declare no competing interests.

There are no competing interests to declare.
