## [Peer Review File · Nature Communications]

Reviewers' comments:

Reviewer #1 (Remarks to the Author):

This study aims to understand why the hippocampus is more vulnerable to degeneration during cerebral hypoperfusion, and why BOLD signals in hippocampus are less reliable compared to neighboring cortex. The methods used to address these questions are state-of-the-art, utilizing in vivo oxy-CBF measurements and in vivo two-photon imaging of neuronal and vascular function in awake, habituated mice, both in visual cortex and hippocampus. The main finding is that the hippocampus is supported by substantially lower vascular density, basal blood flow, and capacity for neurovascular coupling compared to cortex. This difference is suspected to be a result of differences in expression of vasoactive pathways and pericyte/SMC morphology, as opposed to differences in neuronal firing and astrocyte signaling. The various data on oxygen consumption and vascular structure are then cleverly combined with existing data to model oxygen diffusion parameters. Overall, this is a significant and understudied topic, and the use of novel approaches to address the issue is superb. The data are clearly of broad interest to the community.

However, there are a number of issues that if addressed could improve the solidity of the conclusions and interpretation of the data.

Major comments:

(1) Decortication is a very invasive procedure, and I am uncomfortable with the definitive statement (as presented in the discussion) that the hippocampal window itself does not affect basal and evoked vascular function. However, I also appreciate that it is very hard to address this concern without decortication using the present methods. The authors are aware of this concern and have done control studies. However, they are limited. The restricted expression of GFAP and Iba1 to the hippocampal surface may not be sensitive enough to detect broader neurovascular changes. Further, the behavioral tasks are notoriously insensitive, and may only register changes when significant bilateral hippocampal disruption has occurred.

The authors could go further to understand whether surgery affects microvascular structure/function, as this is a key premise for the work. For example, does microvascular structure differ between surgery and non-surgery sides? Also, the open access neuronal, astrocyte, and vascular mRNA data is derived from non-surgerized mice. Could some of these mRNAs, such as those involved in glial vasoactive pathways, be altered by surgery? This seems likely with reactive glial states, and could be examined with RNA hybridization or immunohistochemistry. Perhaps beyond the scope of these studies, but ex vivo studies of hippocampal arteriolar reactivity (surgery vs. no surgery) would really clarify the possibility aberrant vessel function due to surgery.

(2) Related to comment (1), the depth of cortical tissue removed is 1.3 mm, but the implanted cannula is 1.5 mm. Why the discrepancy? Also, a 1.3 mm thickness for cortex (assuming visual cortex) is considered high (typically 0.9-1mm at most). I am concerned about acute compression of the hippocampus in this procedure. This could cause an hypoperfusion like state that impairs neuronal and vascular function from the outset, and then creates an abnormal baseline during in vivo imaging. One way to address the could be

imaging acutely after cortical window and cannula implantation to see how it compares with the basal flow one measures in the chronic state. Another way is to inject probes such as hypoxyprobe or measure neuronal cell death to determine whether there is acute ischemia/hypoxia.

(3) What are the differences in vascular architecture between hippocampus and cortex? Clearly, the arteries perfusing these tissues are distinct (MCA/PCA vs anterior choroidal artery). However, differences at the microvascular level seem still uncharacterized. For example, it is unclear to me how one ensures similar microvascular zones and vessel types are being compared. Are the arterioles imaged in hippocampus more distant from their perfusion source and thus of lower flow? Are vessels being imaged, precapillary arterioles or capillaries? The difference in vascular flow and coupling could be due to sampling of different vessel types compared to cortex. I think the paper would benefit immensely with more details on the vascular anatomy of hippocampus, including details of vascular branch order, distance from perfusion source, SMA expression, proximity to venules etc for vessel regions that would be likely sampled by two-photon. The data on pericytes begins to address this issue, but does not delve deep enough.

(4) Related to comment (3), what is the artery to vein ratio in hippocampus? Considering that the BOLD imaging is more sensitive to oxygen changes in the venous compartment, could artery to vein ratio be a reasons why signals are less reliable?

(5) The authors should heed caution to not overstate the conclusion that reduced resting flow and neurovascular coupling underlies hippocampal vulnerability to ischemia. These are excellent observational and comparative studies. However, the study does not definitely test whether these vascular differences are the cause of vulnerability, as there no manipulations of the vascular reactivity/oxygenation parameters.

Moderate to minor comments:

(6) The precise timing of when imaging is performed after cranial window/cannula implantation is not mentioned. 3-weeks is routine for recovery after cortical window implantation, but how long in this study? Also, should this recovery time be extended for hippocampus to allow inflammation to subside?

(7) In the Introduction, there should be more background on hippocampal vulnerability in the context of vascular dementia and Alzheimer's disease. Only vulnerability to hypoxia is mentioned. Most focus in literature seems to be on BBB integrity, but the premise for focusing on NV coupling could be strengthened.

(8) Using laser doppler to compare resting CBF between brain regions could be an issue, as the approach is good for relative CBF measurements but not absolute flow. Doppler is complemented with RBC velocity measured by two-photon, but does velocity correlate with hematocrit (RBC flux or linear density of RBCs in a line-scan)? Can RBC flux be compared between V1 and HC?

(9) Line 76. "Despite similar energy demands" and Line 92 " neuronal activity, and therefore energy requirements, were the same in each region ...". Is it more accurate to say that neuronal activity/firing is similar, but it feels like a stretch to say that energy demands are the same.

(10) What is considered "resting state" of the animal during imaging? Animals may be immobile but there are certain brain states that can be related to higher CBF (sleep states). These would need electrophysiology to identify.

(11) How was distance between SMCs measured if the cell bodies are not obvious?

(12) Line 477. The meaning of "skeleton pixel" is not clear.

(13) Figure 2e. Capillary length per volume seems odd. ~2000 m of vasculature per cubic mm of tissue is far too high. Perhaps a micrometer symbol is absent. Is it fair to express vascular length per volume if the length is measured in 2D images?

(14) Figure 2g,h. "V1 (39 vessels from 11 mice) than in HC (39 vessels from 9 mice)" This is a fairly small number of capillaries over a large number of mice. Does this mean 4-5 vessels were evenly measured per mouse, or did some mice dominate the data set? Also, what type of vessels were examined? If capillaries only, how does one categorize a capillary in HC? Due to the non-normal distribution of data (skewed to smaller values), non-parametric tests may be needed.

Reviewer #2 (Remarks to the Author):

This is a very interesting paper addressing regional differences with respect to neurovascular coupling. I have a number of serious concerns that need to be addressed.

Major:

- Laser Doppler flowmetry readouts are difficult to quantify. It is okay to compare the measurements over time of the same region, but I am hesitant to believe that a regional comparison holds. Different vascular density and/or vascular geometry can lead to a different result irrespective of the actual CBF differences.
- Line 427. The computation of CMRO₂ based on the LDF and spectroscopy cannot be taken without being more critical. The authors must discuss the possible caveats of this method. It is an oversimplification to suggest that this is a quantitative readout of oxygen consumption. In our own hands, this method is very often showing spurious results and is error prone.
- Along the same line: the only really quantitative imaging readout in this work are the ones from two-photon microscopy, such as diameter and RBC velocity. I would like to see the relative changes of RBC velocity in V1 versus HC in response to activation. This would be really strong and important data for the main conclusion of the paper.
- Line 90-93. It is an oversimplification to state that similar calcium peaks are suggesting the same energy requirements.
- Line 112-115 (and discussion of it). Likewise, it is not straightforward to predict from mRNA levels that neurovascular coupling is the same in HC and V1.
- Line 178. It should be stated that most studies have failed to demonstrate that thin-stranded capillary pericytes are contractile at all.
- Line 270. It is a bit weird to link "contractility" to neurovascular coupling. In fact, it would be more appropriate to talk about the ability to dilation, which would be increasing flow in response to a higher demand.
- Reference 22: There is a more rigid paper by Schmid et al. published in PLoS Comp Biol. This paper needs to be cited as well. It provides a more complete picture of the problem that

that in Gould et al.

- Figure 2f. The scanning trajectory is a problem. The inertia of the scan mirrors will in fact produce a trajectory that is completely different from the one drawn.
- Figure 3e/f: the $\Delta D/D$ values are extremely small. Please explain.

Minor:

- Line 78: lower capillary density does not directly lead to lower CBF and SO_2 .
- Line 95: are dilations really “less frequent” or do they rather occur for a “shorter time period”?
- Figure 5d: Is the size of the calcium peaks affected by the number of detected cells, i.e. is it necessary to normalize the calcium peaks with the number of cells (which are 559 in HC and 338 in V1)?
- Line 253: Is NVC really compromised or just “different”?
- Line 313: Why should lower blood flow allow more oxygen extraction to a larger tissue volume?
- Line 321: Why would oxygen consumption be lower in awake animals? That does not make sense to me.
- Line 595-597: This strongly depends on the oxygen level in the capillary and thus does not generally hold.

Oxygen Modelling:

General:

The oxygen model is very simple, but generally there is nothing wrong with it. In my opinion it is fine to use such a simple model for a rather simple supportive study. However, the limitations/assumptions of such a simple model should be discussed and maybe some additional sensitivity studies should be performed (see comments below).

Major:

- Constants in Equation (4) (lines 604-607): V_{max} and K_m are in the correct range. However, the diffusion coefficient (D) is very weird. First of all, the unit is not correct for a diffusion coefficient, which should be cm^2/s (instead of $1/min$). If this is only a “typo” the chosen value is 6 order of magnitudes different from what is commonly used. I briefly checked the given reference [50]. However, I did not find the stated value there. The diffusion coefficient is a key factor for the plots in Figure 7.
- The interRBC values of $15mmHg/10mmHg$, which were used as boundary conditions for the oxygen model, seem very low compared to the work by the Charpak group. Again, this might affect their general conclusion on the HC being close to hypoxia.
- Points, that should be discussed:
 - o The oxygen saturation in capillaries can vary significantly. This will affect the tissue supply radius.
 - o The role of the vessel wall on oxygen diffusion (smaller diffusion coefficient).
 - o What about hematocrit/RBC flux in the HC? Is it the same as in V1?

Minor:

- Why was an oxygen partial pressure of 0 mmHg used as initial condition?
- Figure 7c: Why is averaging over cases necessary? Simply, combining the entire would data makes more sense.
- Lines 573-576: Please state the used solubility constant used for Henry’s law, instead of pointing to some software.

Response to reviewers

We would like to thank the reviewers for their constructive, broadly very supportive and helpful comments. We have addressed the issues raised with more experiments, analyses and further clarification where relevant, adding 5 new figures to the Supplement and additional panels to 3 further figures. We think the paper is considerably improved, and wish to thank the reviewers for their constructive reviews. Insertions and changes are marked in blue in the main document and the supplement, and are copied below in response to the specific points addressed.

Reviewer #1 (Remarks to the Author):

This study aims to understand why the hippocampus is more vulnerable to degeneration during cerebral hypoperfusion, and why BOLD signals in hippocampus are less reliable compared to neighboring cortex. The methods used to address these questions are state-of-the-art, utilizing in vivo oxy-CBF measurements and in vivo two-photon imaging of neuronal and vascular function in awake, habituated mice, both in visual cortex and hippocampus. The main finding is that the hippocampus is supported by substantially lower vascular density, basal blood flow, and capacity for neurovascular coupling compared to cortex. This difference is suspected to be a result of differences in expression of vasoactive pathways and pericyte/SMC morphology, as opposed to differences in neuronal firing and astrocyte signaling. The various data on oxygen consumption and vascular structure are then cleverly combined with existing data to model oxygen diffusion parameters. Overall, this is a significant and understudied topic, and the use of novel approaches to address the issue is superb. The data are clearly of broad interest to the community.

We thank the reviewer for their positive assessment of our paper.

However, there are a number of issues that if addressed could improve the solidity of the conclusions and interpretation of the data.

Major comments:

(1) Decortication is a very invasive procedure, and I am uncomfortable with the definitive statement (as presented in the discussion) that the hippocampal window itself does not affect basal and evoked vascular function. However, I also appreciate that it is very hard to address this concern without decortication using the present methods. The authors are aware of this concern and have done control studies. However, they are limited. The restricted expression of GFAP and Iba1 to the hippocampal surface may not be sensitive enough to detect broader neurovascular changes. Further, the behavioral tasks are notoriously insensitive, and may only register changes when significant bilateral hippocampal disruption has occurred.

The authors could go further to understand whether surgery affects microvascular structure/function, as this is a key premise for the work. For example, does microvascular structure differ between surgery and non-surgery sides? Also, the open access neuronal, astrocyte, and vascular mRNA data is derived from non-surgerized mice. Could some of these mRNAs, such as those involved in glial vasoactive pathways, be altered by surgery? This seems likely with reactive glial states, and could be examined with RNA hybridization or immunohistochemistry. Perhaps beyond the scope of these studies, but ex vivo studies of hippocampal arteriolar reactivity (surgery vs. no surgery) would really clarify the possibility aberrant vessel function due to surgery.

We agree that it is important to test for any effects of surgery on our results, and the reviewer has suggested some interesting additional approaches which we have addressed where possible with further experiments as follows:

Vascular differences with surgery: As suggested, we have conducted more experiments to assess differences in the vasculature and gross anatomy (in response to the reviewer's later comment) between CA1 in the surgical and non-surgical hemispheres. There were no significant effects of surgery on the aspect ratio, area or vascular density of CA1. These data are now shown in a new figure, Supplementary Figure 9.

mRNA differences with surgery: It would be interesting for future work to study whether the mRNAs involved in neurovascular coupling are affected by surgery. A comprehensive study would be required however, and we do not have the funding for such resources at present. However, in any case, the point we want to emphasize with the mRNA data is that they show that there are differences between these pathways in CA1 and visual cortex in *physiological* conditions. Thus, these mRNA data lend support to our hypothesis that there are vascular differences between these two regions that are not simply due to the effects of surgery.

Ex vivo hippocampal arteriolar reactivity (surgery vs. no surgery): The reviewer suggested this experiment was beyond the scope of this study, but we agree that it could have been helpful, so hoped to do this experiment. Unfortunately, we were prevented by lockdown due to COVID19, and the mice we had prepared for this purpose had to be culled. We have done further analyses of the hippocampal vascular anatomy, however, which shows that the source vessels for those imaged in CA1 originate below CA1 (so are not near the surgical lesion; Figure

6a&b). We therefore do not expect the vessels we are studying to be damaged by the surgery, particularly as we see no differences in responsiveness of vessels nearer and further from the surgery location and any potential inflammatory damage (Supplementary Figure 8).

(2) Related to comment (1), the depth of cortical tissue removed is 1.3 mm, but the implanted cannula is 1.5 mm. Why the discrepancy? Also, a 1.3 mm thickness for cortex (assuming visual cortex) is considered high (typically 0.9-1mm at most). I am concerned about acute compression of the hippocampus in this procedure. This could cause an hypoperfusion like state that impairs neuronal and vascular function from the outset, and then creates an abnormal baseline during in vivo imaging. One way to address the could be imaging acutely after cortical window and cannula implantation to see how it compares with the basal flow one measures in the chronic state. Another way is to inject probes such as hypoxyprobe or measure neuronal cell death to determine whether there is acute ischemia/hypoxia.

Discrepancy between depth of cannular and cortical tissue removed: We apologise for a lack of clarity here. The discrepancy comes because some of the cannula (~0.2mm) lies in line with/above the skull. This makes the cannula more secure for long-term imaging because we can seal it to the skull surface using Vetbond and dental cement. Methods have now been updated to include this information as follows:

“A 3 mm round stainless steel cannula (2.4 mm ID, 3 mm OD, 1.5 mm height, Coopers Needle Works Ltd) with a 3 mm glass coverslip (Harvard Apparatus) attached using optical adhesive was inserted into the craniotomy and secured with dripping tissue adhesive and dental cement (secured at the skull surface, above which ~0.2mm of cannula protruded into the dental cement).”

Compression of hippocampus: As mentioned above, we have investigated this issue further but find no difference in the aspect ratio (height/width) of CA1 in surgical and non-surgical hemispheres, so do not think that significant compression of CA1 is occurring in our mice. For animal welfare reasons, we allow recovery of mice following surgery, and then habituation to the rig before imaging, so cannot image acutely immediately after window implantation, but generally see no deterioration of our measures of brain oxygenation and vascular reactivity over time, even in other experiments with a longer time course (see figures below). Consistent with this, there is no impairment in hippocampal-dependent behaviour over time following surgery (Supplementary Figure 8b).

(3) What are the differences in vascular architecture between hippocampus and cortex? Clearly, the arteries perfusing these tissues are distinct (MCA/PCA vs anterior choroidal artery). However, differences at the microvascular level seem still uncharacterized. For example, it is unclear to me how one ensures similar microvascular zones and vessel types are being compared. Are the arterioles imaged in hippocampus more distant from their perfusion source and thus of lower flow? Are vessels being imaged, precapillary arterioles or capillaries? The difference in vascular flow and coupling could be due to sampling of different vessel types compared to cortex. I think the paper would benefit immensely with more details on the vascular anatomy of hippocampus, including details of vascular branch order, distance from perfusion source, SMA expression, proximity to venules etc for vessel regions that would be likely sampled by two-photon. The data on pericytes begins to address this issue, but does not delve deep enough.

The reviewer here makes several very interesting points around the need to further characterise the differences between the hippocampal and cortical vascular anatomy. We have addressed this in some detail, using new experimental data where we captured the vascular microarchitecture beneath the cranial windows to analyse the position of different vessel types, and the position of the perfusion source (arteries) relative to the typical location of imaged vessels in the two regions. We have added 7 new panels to Figure 6 with the results of these analyses, and an extra Supplementary figure (now Supplementary Figure 5). Our findings relating to the reviewers specific questions are as follows:

What are the differences in vascular architecture between hippocampus and cortex? We show that the perfusing vessels come from the surface in cortex, and from the underlying sulcus in CA1. We find that the length and diameter of the arterioles, precapillaries and venules are broadly equivalent in HC and V1 (Fig. 6), but differences emerge in the diameter of vessels at final smooth muscle cell (before transition to pericyte morphology), and in the capillary bed – the data previously presented, where we find that capillaries are slightly larger in diameter by thin-strand pericytes in V1 compared to HC, and mesh and thin-strand pericytes are significantly shorter in V1 than HC. The relevant text changes are indicated below:

“In the absence of clear differences in neuronal firing properties or expression of neurovascular coupling signalling molecules, we next investigated whether our results could reflect differences in vascular structure or function. The architecture of the vasculature is well established in cortex, where pial vessels run along the brain’s surface before penetrating the neocortex and branching into a dense capillary network (e.g. Figure 6a). The hippocampal vascular network is less well characterised, but is known to be inverted compared to that in neocortex, with large arteries and veins emerging in the hippocampal sulcus and sending their arch-like branches up into CA112. Our in vivo recordings confirm this vascular organisation (Figures 6 a-b, Supplementary Figure 5) with the large (> 15 µm) perfusing vessels located ~300 µm below the dorsal surface of layer SO. As we imaged vessels at 70 µm depth (ranging from 1 to 308 µm), this meant that the vessels we sampled from in HC were generally further from their source (HC vessels averaged 227 µm from their source, versus 135 µm in V1), but the distance to the perfusion source did not alter resting RBC velocity or calcium-dependent blood vessel dilations in HC (Supplementary Figure 5). In V1, the distance from the pial arteries did not affect the size of dilations, but RBC velocity was higher in vessels nearer the source arteries (Supplementary Figure 5).”

“We first studied the length and diameters of arterioles, pre-capillaries and venule branches in in vivo imaging stacks. Arterioles were defined as having a continuous layer of banded SMCs surrounding the vessel, pre-capillaries as being covered in ensheathing pericytes¹, and venules as large vessels proceeding from the mid-capillary bed and lacking continuous SMCs or substantial pericyte coverage. We measured the length of arteriole and venule branches, and of the precapillaries, as well as vessel diameters. The lengths and initial diameters of arterioles, venules and precapillary arterioles were equivalent in HC and V1 (Figures 6 d,f,g), but at the transition between arteriole and precapillary (the boundary between the last SMC and the first ensheathing pericyte), vessel diameters were significantly smaller in HC than V1 (Figure 6e).”

Are the arterioles imaged in the HC more distant from their perfusion source and thus of lower flow?

We have now addressed this point in Supplementary Figure 5. We have shown that the vessels imaged in HC are indeed more distant from their arterial perfusion source vs V1, but that in the HC the distance to this source does not affect resting RBC velocity or calcium-dependent vessel dilations. In V1, however, RBC velocity increases are larger nearer the pial arteries, but diameter changes are not different depending on distance from the perfusion source. We have added the following text to the results section (lines 209-214):

“Our imaging depth was, on average, 70 µm (range: 1 to 308 µm), so the vessels we sampled from in HC were generally further from their source (HC vessels were on average 227 µm from their source, versus 135 µm in V1), but the distance to the perfusion source did not alter resting RBC velocity or calcium-dependent blood vessel dilations in HC (Supplementary Figure 5). In V1, the distance from the pial arteries did not affect the size of dilations, but RBC velocity was higher in vessels nearer the source arteries (Supplementary Figure 5).”

The absence of a decreased RBC velocity in vessels further from the arterial perfusion source is somewhat surprising. In cortex, RBCs that travel through superficial capillaries take a shorter path than those that dive deeper into cortex², whereas the absence of such a pattern in HC suggests that paths through the capillary bed are of equal resistance. The reasons for this would be an interesting matter for future study, using the same sort of analysis as used by Schmid et al in visual cortex². Path lengths might be the same through superficial and deep regions of HC due to convoluted microvascular paths, or capillaries nearer the perfusion source could be smaller, and therefore of higher resistance. We have added a couple of sentences about the different flow patterns to the discussion (lines 384-389):

“The flow characteristics of the two vascular beds were also different. In V1, RBC velocity was faster nearer the arterial source, consistent with a shorter, lower resistance path being taken by RBCs passing through more superficial neocortical layers². HC flow did not show this dependence on distance from the perfusion source, suggesting the absence of these shorter, lower resistance paths in HC.”

It is unclear how one ensures similar microvascular zones and vessel types are being compared? Are vessels being imaged capillaries or precapillary arterioles?

We know broadly that similar vessel types were studied as we sample from the same range of diameters for our high-speed line scans (Figure 2) and our xy recordings of vessel diameter (Figure 3) across both brain regions.

The line scan data presented in Figure 2 and Supplementary Figure 1 (a new figure which measures RBC velocity and diameter changes in response to local calcium changes) is all taken from capillaries (diameters < 7 μm , HC average: 4.9 μm , V1 average: 5.2 μm), as it is harder to get a good signal in larger vessels.

The xy recordings of vessel diameter over time presented in Figure 3 include arteriole branches, pre- or postcapillaries and capillaries, but not pial or penetrating arterioles, over a diameter range of 3.6-19.5 μm (HC average: 9.3 μm , V1 average: 8.3 μm). The average diameter of the vessels studied in V1 and HC was the same ($p=0.28$, Supplementary Figure 2, statistics report table SR2b), suggesting they are from the same population. We were not able to trace vessel branch order in the less stereotypically organised hippocampus, where it was not easily possible to find the feeding vessel deep in the tissue, so we could not compare branching orders for the two regions. Also, because we did not have pericytes labelled with DsRed for many of our datasets (mice being only singly transgenic for Thy1-GCaMP6f), we were not able to determine exactly which vessel types we were imaging from within this diameter range. However, to further investigate the contribution of the different vessel types, we added a new figure (Supplementary Figure 2) in which we split the xy recordings of diameter by capillary sizes equivalent to those presented in Supplementary Figure 1 (i.e. $\leq 7 \mu\text{m}$ and $> 7 \mu\text{m}$). In the vessels larger than 7 μm , in V1 we see that dilations are both larger and more frequent in response to calcium events compared to in HC. However, when we look at the smallest vessels only (i.e. $\leq 7 \mu\text{m}$), we again see a higher frequency of responsive vessel dilations to calcium events in V1 vs HC, but these dilations were equivalently sized between regions.

These results were consistent with our data presented in the new Supplementary Figure 1, which we added to address the reviewer's question about stimulation-induced changes in RBCV. In this additional experiment, we used line scans to measure diameter, RBC velocity and neuronal calcium concurrently, in only capillaries (3.0-7.2 μm , HC average: 4.7 μm ; V1 average: 5.1 μm). Diameters were not significantly different: $p = 0.32$, unpaired t-test). In response to calcium events, in agreement the data from xy recordings presented in Supplementary Figure 2, the dilations were more frequent in V1 vs HC, but were equivalently sized. When we measured RBC velocity in response to calcium events however, whilst the frequency of RBCV increases were equivalent in HC and V1, the velocity changes were larger in V1.

For both the xy and line scan data, equivalently smaller HC dilations in capillaries would probably not be detectable in HC as we are likely at the limit of detection with the dilations we report. In vessels > 7 μm , HC dilations are around 50% of those in V1. In vessels < 7 μm this sort of reduction would lead to dilations <60nm or only 0.5 SD above baseline). Thus altered HC microvascular function in these small vessels, consistent with the changes in pericyte morphology we report, would be expected to be detected in our experiments as a reduction of response frequency rather than, necessarily of response size.

Given that dilations and signals to dilate are known to propagate upstream from the capillary bed, and the decrease in capillary responsiveness in HC, the less frequent and smaller dilations we see in larger HC vessels could either reflect the lower sum of downstream dilation events and/or a deficit in pericytes or endothelial cells in these larger vessels.

These results regarding the contribution of different vessel types have been added as Supplementary Figures (1 & 2), and have been described in the text (lines 119-132).

"These smaller dilations we observed across all vessels corresponded to smaller increases in RBC velocity in HC capillaries following local calcium activation, as assessed using fast line scanning of capillaries (< 7 μm) and nearby neuronal soma (Supplementary Figure 1), though RBC velocity was equally likely to increase in the two regions. Whilst these same HC capillaries captured by fast line scanning were less likely to dilate than V1 capillaries, if they did dilate, their responses were the same size in both regions. When we split our movie data in Figure 3 based on vessel size, we found the same result: vessels (both those $\leq 7 \mu\text{m}$ and $> 7 \mu\text{m}$) were less likely to dilate in HC than V1. Those larger than 7 μm had smaller dilations in HC, whereas those smaller than 7 μm showed the same sized dilations between regions (Supplementary Figure 2). Dilations of HC vessels > 7 μm were 51% of those in V1, however, so similarly-scaled responses in smaller vessels would be undetectable (and classed as non-responders) as they would be <60 nm, or only 0.5 SDs above baseline. "

What is the artery to vein ratio in the hippocampus? Considering that the BOLD imaging is more sensitive to oxygen changes in the venous compartment, could artery to vein ratio be a reasons why

signals are less reliable?

We have investigated this using low magnification *in vivo* z-stack recordings, in mice with DsRed-labelled pericytes and smooth muscle cells. In the tissue column, under our HC cranial window (5 stacks from 3 animals) we see an average of 2.2 veins (vein range 2-3) and 1.2 arteries (artery range 1-2, artery/vein ratio: 0.53), and under our V1 cranial window (8 stacks from 8 animals) we see an average of 2 veins (vein range 1-3) and 1.25 arteries (artery range 1-2, artery/vein ratio: 0.69). The artery/vein ratio measured under our cranial window was not significantly different between regions (unpaired t-test, $p=0.53$). We have added the following text (lines 238-244):

*“We first studied the number of main arteries and veins, and the length and diameters of arterioles, pre-capillaries and venule branches in *in vivo* imaging stacks. Arterioles were defined as having a continuous layer of banded SMCs surrounding the vessel, pre-capillaries as being covered in ensheathing pericytes¹, and venules as large vessels proceeding from the mid-capillary bed and lacking continuous SMCs or substantial pericyte coverage. The artery/vein ratio was no different in HC and V1 (0.69 ± 0.25 in V1 and 0.53 ± 0.1 in HC, t test, $p = 0.53$, Supplementary Table 7).“*

We have also added more information about the composition of the vascular network, and specifically the morphology of the perfusing arteries and veins in both regions (Figure 6, lines 201-216). We do not observe any differences between arteriole and venule length.

*“The architecture of the vasculature is well established in cortex, where pial vessels run along the brain’s surface before penetrating the neocortex and branching into a dense capillary network (e.g. Figure 6a). The hippocampal vascular network is less well characterised, but is known to be inverted compared to that in neocortex, with large arteries and veins emerging in the hippocampal sulcus and sending their arch-like branches up into CA1¹². Our *in vivo* recordings confirm this vascular organisation (Figures 6 a-b, Supplementary Figure 5) with the large ($> 15 \mu\text{m}$) perfusing vessels located $\sim 300 \mu\text{m}$ below the dorsal surface of layer SO. As we imaged vessels at $70 \mu\text{m}$ depth (ranging from 1 to $308 \mu\text{m}$), this meant that the vessels we sampled from in HC were generally further from their source (HC vessels averaged $227 \mu\text{m}$ from their source, versus $135 \mu\text{m}$ in V1), but the distance to the perfusion source did not alter resting RBC velocity or calcium-dependent blood vessel dilations in HC (Supplementary Figure 5). In V1, the distance from the pial arteries did not affect the size of dilations, but RBC velocity was higher in vessels nearer the source arteries (Supplementary Figure 5).“*

(5) The authors should heed caution to not overstate the conclusion that reduced resting flow and neurovascular coupling underlies hippocampal vulnerability to ischemia. These are excellent observational and comparative studies. However, the study does not definitely test whether these vascular differences are the cause of vulnerability, as there no manipulations of the vascular reactivity/oxygenation parameters.

We thank the reviewer for the point, which is well-taken. We intend only to offer this as a possible contributory factor, the importance of which should be examined experimentally (we are currently setting up experiments to do just this). We have revised the discussion. Originally, this read *“But while HC may cope with low [O₂] physiologically, its lower oxygenation and weaker ability to match oxygen demand to supply can help explain its vulnerability to hypoxia and conditions where cerebral blood flow decreases, such as ischaemia and Alzheimer’s disease.”*

We have changed the language used to be less conclusive, *“can help explain its vulnerability to”* now reads *“may contribute towards its vulnerability to”*

Moderate to minor comments:

(6) The precise timing of when imaging is performed after cranial window/cannula implantation is not mentioned. 3-weeks is routine for recovery after cortical window implantation, but how long in this study? Also, should this recovery time be extended for hippocampus to allow inflammation to subside?

We image at least 1 month after surgery (habituation to the rig begins after > 1 week), and on average the recovery period is longer for HC surgeries beginning 43 ± 6 days in HC (N=22) and 35 ± 4 days in V1 (N=28). The methods section is now clearer with regards to the recovery period, *“recovery period of at least one week”* has been updated to say *“recovery period (average number of recovery days in HC: 43 ± 6 , and V1: 35 ± 4)”*. In these conditions, responses are stable from the start of imaging, with no changes being observed in either calcium or vessel diameters with increased time from surgery (as shown in the figures above).

(7) In the Introduction, there should be more background on hippocampal vulnerability in the context of vascular dementia and Alzheimer’s disease. Only vulnerability to hypoxia is mentioned. Most focus in literature seems to be on BBB integrity, but the premise for focusing on NV coupling could be strengthened.

We agree with the reviewer that our work has huge relevance for the fields of vascular dementia and Alzheimer's disease where early vascular changes are likely to be key drivers of subsequent pathology. We have added text to the first paragraph of the introduction (lines 20-23) to highlight hippocampal vulnerability to vascular dementia and Alzheimer's disease as follows:

"In Alzheimer's disease, blood brain barrier (BBB) dysfunction and inadequate cerebral perfusion promote A β accumulation and neurofibrillary tangles^{3,4}, and patients with mild-cognitive injury show lower local blood volume in the hippocampus⁵."

We have also added a sentence towards the end of the introduction (lines 51-56) to explain the premise for focussing on NVC:

"We studied both resting haemodynamics and neurovascular coupling in hippocampus and cortex, as both factors are likely to be important for maintaining neuronal and cognitive health. As discussed above, decreases in overall perfusion have been associated with brain damage⁷, while neuronal activity-driven vessel dilations correlated with reduced oxygenation and preceded cell death in cortex¹²."

(8) Using laser doppler to compare resting CBF between brain regions could be an issue, as the approach is good for relative CBF measurements but not absolute flow. Doppler is complemented with RBC velocity measured by two-photon, but does velocity correlate with hematocrit (RBC flux or linear density of RBCs in a line-scan)? Can RBC flux be compared between V1 and HC?

Use of LDF: This was an interesting point from the reviewer. Relative changes are mostly reported using LDF, and indeed, it is not possible to calibrate the readout to get an actual value for flux so our reported effects are qualitative not quantitative. However, we think it holds value for comparing between regions, or for use on baseline readings, as long as the factors that could be affecting the signal are taken into account. The main relevant potential confounding factors in using LDF that could create problems when comparing regions are (see Ref ⁸):

- 1) Movement artefacts: our probe is fixed and the mouse is head-fixed so these are minimised.
- 2) Biological zero: Our probe is calibrated so that it reads zero at biological zero, something we verified in Supplementary Figure 10d by recording haemodynamic parameters while culling the mouse.
- 3) Spatial heterogeneity of perfusion within subjects, and probe placement: We plotted the variability between data recorded on different sessions within the same animal and between animals in Supplementary Figure 10a. Variability in values of CBF was greater than that of the haemoglobin absorbance values recorded using spectroscopy, possibly reflecting differences in probe placement (e.g. over large pial vessels) and spatial variability in perfusion. Nevertheless, there were no significant differences between recordings obtained within a region (V1 or HC) while the difference between regions was significant (as reported in Fig. 2b)
- 4) Sampling depth: Our LDF probe has an excitation-detection fibre separation of 500 μm and a excitation wavelength of 780 nm so therefore, unlike other fibre designs, has a deep sampling depth, with a median depth sampled of around 0.9 mm and vessels as deep as 2 mm can contribute to the signal ⁹. Thus, the sampling volume for both regions extends much deeper than the depth of vessels imaged using 2-photon microscopy (400 μm): In mouse the cortical thickness is about 1 mm, while the thickness of CA1 and the sulcus containing the feeding vessels is about 0.9 mm and the whole hippocampus (CA1, plus the deep vessels in the sulcus and the dentate gyrus) is about 1.4 mm ¹⁰. From Fabricius et al ⁹, this means that approximately 70 % of the HC LDF signal actually derives from the hippocampus (including the dentate gyrus; 55% originating from CA1 plus the feeding vessels), the rest being from thalamus, and 60 % of the V1 LDF signal derives from V1, the rest being from the corpus callosum and posterior hippocampus. Because the corpus callosum has a low vascular density, so cannot be artificially boosting the V1 LDF signal, and the thalamus has a high vascular density¹¹ so cannot be artificially reducing the HC LDF signal, the LDF differences we observe in V1 compared to HC are highly likely to originate in perfusion differences between these two regions.

One potential interacting factor is the position of the high flow feeding vessels in the two regions. Importantly, LDF signals in both regions include these larger arteries that feed the microvascular networks in the respective regions, but in V1 the more superficial pial vessels will contribute more strongly to the net signal than the deeper feeding vessels in the hippocampus. However the total pial contribution is likely less than 10% of the total signal (as 10% of the signal originates in the first 100 μm of the tissue⁹, which is larger than the pial depth ¹²), so the superficial location of V1 feeding vessels is unlikely to be the cause of the 33 % difference in LDF signal between the two regions and rather the

majority of this signal is likely to be due to the different vascular density, and possibly flow in individual vessels.

In summary, after considering the different potential confounding factors that could generate artefactually different LDF signals in V1 and HC, we think it is highly likely that the observed regional difference in LDF signal between these two regions does, indeed, reflect differences in CBF, particularly as the vascular anatomy and single vessel flow also reflects these results. To reflect some of these points, we have added the following to the methods:

“Laser doppler flowmetry is commonly used to compare relative not absolute differences in flow, but we reasoned that if we could discount potential artefactual contributions to the LDF signal⁸, differences in the signal between regions might be informative about regional differences in CBF. Firstly, we checked whether probe placement across sessions affected signal variability. We plotted the variability between data recorded on different sessions within the same animal and between animals (Supplementary Figure 10). Variability in values of CBF was greater than that of the haemoglobin absorbance values recorded using spectroscopy, possibly reflecting differences in probe placement (e.g. over large pial vessels) and spatial variability in perfusion. Nevertheless, there were no significant differences between recordings obtained within a region (V1 or HC) while the difference between regions was significant (Fig. 2b)

We also checked that fluorescence signals from the brain did not interfere with the haemodynamic measurements and reached zero in the absence of blood flow following the death of the subject (Supplementary Figure 10). Finally, we considered the possible effects of the large sampling volume of our LDF probe on our results. It has a 500 μm separation between the transmission and detection fibres and a wavelength of 780 nm, so samples deep in the tissue, with a median signal depth of around 0.9 mm and a maximum depth of 2 mm⁹. Thus, some of the signal recorded in V1 comes from underlying corpus callosum and hippocampus, and some of that recorded from HC originates in the thalamus. However, it is highly unlikely that signal from these areas could produce the regional differences we report here, as corpus callosum and hippocampus have low vascular densities so cannot be boosting the apparent V1 CBF signal, while the thalamus has a high vascular density¹¹ so will not be artificially boosting the signal attributed to HC.”

RBC Flux: This was another good point from the reviewer - we have now extracted RBC flux and haematocrit from our line scan recordings and incorporated these data into Figure 2. Similarly to the significant regional differences we observed in RBC velocity, we also saw increased RBC flux and haematocrit in the individual capillaries of V1 compared to HC. This provides more of a basis for understanding the decrease in CBF in HC. We have added extra panels with these data to Figure 2, amended the methods to explain extraction of these data and have amended the results text to read:

“However, when we measured red blood cell (RBC) flux, haematocrit and RBC velocity in individual blood vessels, after loading them with fluorescent dextran (Figure 2g), we found that, despite the sampled vessels themselves being of equal size (Figures 2f), RBC velocity, flux and haematocrit were significantly lower in the HC than V1 (Figures 2h-j). This combination of a lower capillary density and RBC velocity and flux in the HC explains both the observed lower net flow and, because fractionally more oxygen is extracted from capillaries with lower flow rates¹³, the lower blood oxygenation we measured in HC.”

(9) Line 76. “Despite similar energy demands” and Line 92 “neuronal activity, and therefore energy requirements, were the same in each region ...” Is it more accurate to say that neuronal activity/firing is similar, but it feels like a stretch to say that energy demands are the same.

We agree with the reviewer and have removed this comment. In fact we reran this analysis using Mann-Whitney U tests for non-normally distributed data, which indicated that calcium events were significantly *larger* in HC than V1. The relevant paragraph now reads (lines 103-111):

“Because fMRI studies suggest neuronal activity might be less well matched to blood flow in HC than cortex, we investigated the capacity of blood vessels to respond to local excitatory neuronal calcium events in vivo by capturing movies of blood vessels and neurons in HC and V1 using 2 photon microscopy (Figure 3). In both regions, vessels dilated shortly after neuronal calcium events (Figures 3d, e, g, h), however, the frequency and size of dilations were significantly greater in V1 than in HC (Figures 3c, e, h), while the average size of calcium peaks was larger in HC (Figures 3d, g, j, l). This suggests hippocampal vessels were less able to match increased activity with increased vasodilation and therefore blood flow.”

(10) What is considered "resting state" of the animal during imaging? Animals may be immobile but there are

certain brain states that can be related to higher CBF (sleep states). These would need electrophysiology to identify.

We apologise if this was unclear. We do not use the term “resting state”, as indeed we cannot make conclusions about the underlying state of the animal. We simply wish to refer to the behavioural state of the animal – i.e. it is resting, or immobile, in the absence of external stimuli. In the first paragraph of the results section we have now added a sentence to clarify what we mean when we use the term “resting” (lines 76-79):

“Mice could run on a running cylinder or remain stationary, while visual stimuli (drifting gratings or a virtual reality environment) or a black screen were presented (Online Methods, Figures 1a-d). Throughout, resting baseline measurements refer to those taken when the animal is immobile and in the dark.”

(11) How was distance between SMCs measured if the cell bodies are not obvious?

SMCs can be identified from their banded morphology, whereas pericytes have a distinct soma and processes. The distance between SMCs can therefore be measured as the distance between adjacent bands, in cells without a clear protruding soma. However, we have now removed the data presenting the distance between SMCs from Figure 6, because given our further analysis of arterioles and venules we have now split our analysis of vascular anatomy into two broad sections. First (Figs 6a-g) is now an analysis of in vivo image stacks, which allows us better to trace the connectivity of larger vessels but do not give us sufficient resolution to accurately measure the diameters of small vessels. Second (Fig 6h-k), we present the analysis of the capillary bed in fixed tissue (pericyte vascular territories). In confocal images of this preparation, resolution is high enough to allow for accurate identification of different pericyte types (where fine processes are not possible to see in lower resolution stacks) and measurement of small vessel diameters, but slicing of the brain prevents us from tracing large vessel connectivity.

(12) Line 477. The meaning of “skeleton pixel” is not clear.

We apologise for not having explained this well enough. We have now updated the methods section to add more detail about vessel skeletonization.

“Briefly, for each frame of XY diameter movies the vessel was skeletonised, meaning its length was eroded to generate a single-pixel diameter line running through the entire length of the vessel at its centre. The intensity profile of a line perpendicular to this vessel axis was taken at every other pixel along the length of this skeleton, averaged across a running window of 5 pixels. The diameter was calculated from the full width half maximum of this intensity profile.”

(13) Figure 2e. Capillary length per volume seems odd. ~2000 m of vasculature per cubic mm of tissue is far too high. Perhaps a micrometer symbol is absent. Is it fair to express vascular length per volume if the length is measured in 2D images?

The reviewer is absolutely correct. This was a typo left over when we converted the axis scale from mm to m. We have now updated the axis so that the value is also in metres (i.e. 2m/mm³ rather than 2000(m)/mm³). The vascular length was calculated in 3D image stacks representing a volume, so should indeed be expressed in length/volume.

(14) Figure 2g,h. “V1 (39 vessels from 11 mice) than in HC (39 vessels from 9 mice)” This is a fairly small number of capillaries over a large number of mice. Does this mean 4-5 vessels were evenly measured per mouse, or did some mice dominate the data set? Also, what type of vessels were examined? If capillaries only, how does one categorize a capillary in HC? Due to the non-normal distribution of data (skewed to smaller values), non-parametric tests may be needed.

We have added more line scan data since the reviewers comments so the relevant sentence now reads “V1 (54 vessels from 14 mice) than in HC (55 vessels from 11 mice)”

In Figure 2, all the vessels sampled were capillaries (as they were taken by a line scan which cannot take clear velocity readings from larger vessels, and were less than 7 µm in diameter: HC 4.89µm, V1 5.24µm). At least 7 animals per brain region contributed at least 4 vessels to the dataset (average number in HC is 5, average in V1 is 4). In a new experiment we recorded line scans from an additional 15-16 capillaries, from 5 mice per region, again from vessels less than 7 µm in diameter Supplementary Figure 1).

In Figure 3 (HC: 46 vessels from 6 animals, V1 41 vessels from 7 animals) the vessels were small arterioles or

capillaries (not pial or penetrating arterioles), and at least 5 animals per region contributed at least 4 vessels (average number in HC is 8, average in V1 is 6). We have now added an additional supplementary Figure (Supplementary Figure 2) where vessels were categorised based on size, as we did not have branching order information available in HC (though there was no significant difference in the range of vessel sizes sampled from in HC and V1, $p=0.28$, Range: 3.6-19.5 μm , HC average: 9.3 μm , V1 average: 8.3 μm).

In order to test whether some vessel recordings do “dominate the dataset” we have now run a linear mixed model (LMM) with diameter/calcium events, brain region and responsive index as fixed factors, and vessel ID inputted as a random factor, allowing us to test for any interaction between vessel ID and responses. LMM for vessel diameter peaks still showed a significant interaction between brain region and vessel responsiveness ($p=9.2\text{E-}6$), whereas our LMM for calcium peaks showed no significant differences between brain regions ($p=0.75$). However we haven’t used the results from the LMM in the revised manuscript because the reviewer was correct in pointing out that nonparametric tests would be more appropriate, so we have used these analyses instead.

Generally speaking, our two-photon data was not typically normally distributed but was skewed to lower values (Figure 2 line scan resting RBCV/flux/haematocrit, Figure 3, Figure 5 calcium imaging), so for our pairwise comparisons we switched from using t-tests to Mann-Whitney tests (for unpaired data) and Wilcoxon tests (for paired data). For our ANOVAs we switched from a traditional ANOVA to a Welch’s ANOVA (e.g. when the assumptions of equal variance between groups were violated; Figure 6 confocal diameters and cell lengths, Supplementary Figure 3 vessel diameter and $\text{NVC}_{\text{index}}$, Supplementary Figure 4 diameter, Supplementary Figure 5 diameters). A Welch’s ANOVA is a one-way ANOVA, informing us only of overall differences between groups, so we used the non-parametric Games-Howell post hoc comparisons to see where our differences occurred (as specified in both the figure legend and in the supplementary statistics report).

Reviewer #2 (Remarks to the Author):

This is a very interesting paper addressing regional differences with respect to neurovascular coupling. I have a number of serious concerns that need to be addressed.

Major:

- Laser Doppler flowmetry readouts are difficult to quantify. It is okay to compare the measurements over time of the same region, but I am hesitant to believe that a regional comparison holds. Different vascular density and/or vascular geometry can lead to a different result irrespective of the actual CBF differences.

We have addressed this point in detail above in response to the first reviewer’s similar point, and explain why we think that the LDF signal does indeed reflect regional CBF, and the textual changes we have introduced to further discuss this important point. We deal with differences in vascular geometry in the above discussion (there is probably a very minor contribution to the detected difference due to the different location of the large feeding vessels). Regarding vascular density, we absolutely agree that our CBF measurements will be impacted by vascular density and think that this is a major reason for the lower CBF in HC vs. V1, as we discuss in the manuscript:

e.g. Results; **Blood flow and oxygenation are lower at rest in hippocampus than V1:** *“This combination of a lower capillary density and RBC velocity and flux in the HC explains both the observed lower net flow and, because fractionally more oxygen is extracted from capillaries with lower flow rates¹³, the lower blood oxygenation we measured in HC.”*

- Line 427. The computation of CMRO2 based on the LDF and spectroscopy cannot be taken without being more critical. The authors must discuss the possible caveats of this method. It is an oversimplification to suggest that this is a quantitative readout of oxygen consumption. In our own hands, this method is very often showing spurious results and is error prone.

We agree with the reviewer that our measurements of CMRO2 are not fully quantitative – they rely on the use of arbitrary units of deoxy and total haemoglobin and CBF, so do not inform about the actual number of oxygen molecules consumed. By measuring the relative, rather than absolute, levels of oxy- and deoxyhaemoglobin, however, a number of the limitations of the spectroscopy method for measuring haemoglobin levels, and thus CMRO2 are overcome (e.g. light scattering in tissue¹⁴). Indeed the combined haemoglobin and CBF probe we use has been verified in different tissue types from skin to brain^{14,15} so we can be confident in the relative levels of oxy- and deoxyhaemoglobin we measure across the hippocampus and cortex. The other factor that feeds into the CMRO2 calculation is cerebral blood flow, as measured with LDF. As discussed above, LDF is not commonly used for baseline recordings, but we discuss at length above the reasons why we think this is valid in our

experiments. The same arguments, and limitations, apply for the CMRO₂ estimates that depend on these values. We concluded above, that different organisation of large vessels in hippocampus versus cortex (deep vs. superficial, respectively) was the only factor that could reasonably affect our measured CBF differences in the two regions, but it was very unlikely to account for our measured effect (estimated effect of vascular organisation <10 %, measured effect = 33%). Specifically considering CMRO₂ measurements, we find the same estimated CMRO₂ in both regions despite different CBF and sO₂ (calculated from oxyHb/(oxyHb + deoxyHb) measurements. That the two different methods employed by the probe (LDF and spectrometry) both give consistent results - i.e. sO₂ and CBF both are lower in the hippocampus, gives support to the idea that these results are based on physiological differences between the two regions, and give added confidence to our estimate of CMRO₂.

As others have found¹⁶, we do find that the variability in our CMRO₂ measurements is quite large between imaging sessions, largely driven by the variation in CBF (Supp Fig 9a and b). Nevertheless, we would expect to be able to detect a difference in the size of CMRO₂ fluctuations between the two regions of a similar scale as the difference in HbT that we observed (see Figure 5g and i). Indeed CMRO₂ measurements measured this way have been shown to scale with electrical activity¹⁷, suggesting they are a good representation of tissue's metabolic activity.

In addition to the text changes discussed in reference to LDF measures, which are also pertinent to this question, we have amended the text to emphasise that the measurements of CBF and CMRO₂ are not fully quantitative, being based on arbitrary rather than absolute units with the following sentence inserted into the methods:

“Except for sO₂, which is expressed as a percentage of 100% saturation, these measurements are expressed in arbitrary units as we cannot calculate absolute flow rates and haemoglobin concentrations from these data.”

In addition, we have referred to CMRO₂ as being “estimated” instead of “calculated” to further emphasise the semi-quantitative nature of this measure.

- Along the same line: the only really quantitative imaging readout in this work are the ones from two-photon microscopy, such as diameter and RBC velocity. I would like to see the relative changes of RBC velocity in V1 versus HC in response to activation. This would be really strong and important data for the main conclusion of the paper.

We agree that with the reviewer that the most direct and quantitative information about what is going on comes from the single vessel recordings (though we do think that the net measures add some useful information). We had not previously included the experiment suggested by the reviewer as it is somewhat more challenging than the association of calcium events with diameter changes, as it requires using line scan recordings through capillaries and nearby individual neurons. Thus the chance of seeing a calcium event is lower than when more cells can be recorded simultaneously, and the recordings are more sensitive to interference from the mouse's movement. We agree that it is useful to compare the association of RBC velocity and neuronal activity between the two regions, however, so have now performed this experiment. The data are now presented in Supplementary Figure 1: In agreement with the dilation data in Figure 3, we again see smaller changes in RBC velocity in response to local calcium events in HC vs V1.

- Line 90-93. It is an oversimplification to state that similar calcium peaks are suggesting the same energy requirements.

This is a fair point and we have removed this comment. In fact, the data presented in this figure was not normally distributed, so a Mann-Whitney test was run instead of a t-test. The Mann-Whitney test showed calcium peaks to be actually significantly higher in HC, so incorporating the suggested language change and the new statistical analysis the text now reads (now lines 103-111):

“Because fMRI studies suggest neuronal activity might be less well matched to blood flow in HC than cortex, we investigated the capacity of blood vessels to respond to local excitatory neuronal calcium events in vivo by capturing movies of blood vessels and neurons in HC and V1 using 2 photon microscopy (Figure 3). In both regions, vessels dilated shortly after neuronal calcium events (Figures 3d, e, g, h), however, the frequency and size of dilations were significantly greater in V1 than in HC (Figures 3c, e, h), while the average size of calcium peaks was larger in HC (Figures 3d, g, j, l). This suggests hippocampal vessels were less able to match increased activity with increased vasodilation and therefore blood flow.”

- Line 112-115 (and discussion of it). Likewise, it is not straightforward to predict from mRNA levels that neurovascular coupling is the same in HC and V1.

We apologise that our meaning is unclear. We don't mean to imply that we can test, using mRNA levels, whether or not neurovascular coupling is the same in HC and V1. Rather, we want to investigate any underlying differences that could explain the mechanism for the differences we observe using 2 photon imaging of neurons and vessels. We have made our point more specific, altering the text from:

“There were no differences in levels of mRNA in individual pyramidal cells, interneurons and astrocytes of the synthetic enzymes for prostaglandins, EETs, and nitric oxide that could explain neurovascular coupling being weaker in HC than V1.”

The section in bold now reads (now lines 154-155): *“suggesting that cells in HC are as capable of producing these vasoactive molecules as those in cortex.”*

- Line 178. It should be stated that most studies have failed to demonstrate that thin-stranded capillary pericytes are contractile at all.

We would dispute that *most* studies have not found thin-stranded pericytes to dilate, though agree that this is a controversial issue, and tackle this in the discussion. We argue that the evidence does suggest (in studies with sufficient imaging resolution to detect small diameter changes) that the diameter of capillaries covered in thin-strand pericytes is modulated but, as we mention in the discussion, “Whether these dilations are active or passive remains controversial”. We have substantially rewritten this part of the discussion to discuss the new findings about different results in different sized vessels, and have included text to further clarify that this topic is controversial, but justifying our position that mid-capillary pericytes are important (lines 399-407):

“Dilations of these small capillaries are observed in several 12,33,34, but not all studies³⁵ and whether they are active or passive remains controversial 19,35. Nevertheless, these dilations seem important. Firstly, at least some of these mid-capillary pericytes express the contractile protein α SMA (though in a form that is less stable than in smooth muscle cells and ensheathing pericytes 36). Secondly, their level of intracellular calcium decreases in response to neuronal activity (consistent with a relaxation of contractile machinery 32). Finally, their dilations in olfactory bulb and neocortex seem to mediate a large proportion of the overall increase in blood flow 32,33.”

- Line 270. It is a bit weird to link “contractility” to neurovascular coupling. In fact, it would be more appropriate to talk about the ability to dilation, which would be increasing flow in response to a higher demand.

We have amended the text from “Lower contractility of hippocampal vasculature limits neurovascular coupling.” to read “Lower vasodilatory capacity of hippocampal vasculature limits neurovascular coupling” (lines 375-376).

- Reference 22: There is a more rigid paper by Schmid et al. published in PLoS Comp Biol. This paper needs to be cited as well. It provides a more complete picture of the problem than that in Gould et al.

We think probably the reviewer is referring to Schmid et al (2017)²: “Depth-dependent flow and pressure characteristics in cortical microvascular networks”. Like the Gould paper, this elegant paper presents a detailed analysis of the pressure drops in the microvascular network but concludes that, for deep tissue layers, the largest drop in pressure is not in the capillary bed, as suggested by the Gould paper, but in the arterioles and precapillaries. The Gould paper is not being cited by us, however, because of the analysis of pressure across the vascular bed, but because it (unlike the Schmid paper) also investigates oxygen extraction from the blood and, to do so, provides a summary of experimentally-determined CMRO₂ values across the field. This is the range of values we simulated in Fig. 6, hence the citation. However, the Schmid paper is very relevant for our new findings that RBC velocity is highest near to the perfusion source in V1 but not in HC (Supplementary Figure 5), agreeing with experimental work they cite and their simulations. Specifically, they find RBC flow is higher in capillaries nearer to the pia than in deeper layers, in which the pressure gradient is smaller across the capillary bed. In HC, however, we do not find a significant relationship between distance from the feeding vessel and the RBCV (Supplementary Figure 5) and RBCV is lower. This suggests that similar short path length, low resistance routes through the capillary bed are not present in HC. We have added the following text to the discussion about these new results (lines 384 to 389):

“The flow characteristics of the two vascular beds were also different. In V1, RBC velocity was faster nearer the arterial source, consistent with a longer, higher resistance path being taken by RBCs passing through deeper cortical layers². HC flow did not show this dependence on distance from the perfusion source, suggesting the absence of these shorter, lower resistance paths in HC.”

- Figure 2f. The scanning trajectory is a problem. The inertia of the scan mirrors will in fact produce a trajectory that is completely different from the one drawn.

This is true. We have drawn the scan trajectory we requested in the software and not that actually taken by the mirrors. We are, nevertheless, able to identify the relevant parts of the scan trajectory required from analysis, i.e. the vessel axis (for RBC flux/velocity and haematocrit readings), vessel width (for diameter), and now (for Supp Fig 1), the neuronal soma from the scan pattern. For analysis purposes we select only the centre of the linescan, where the RBC "streaks" are linear for velocity and flux calculation, and the centre of the neuronal soma, so that any deviations in the trajectory due to mirror inertia will be minimised. For the diameter measurements, the trajectory deviates from the normal line across the vessel well away from the edge of the vessel, so deviations in the trajectory should not affect diameter measurements. We have added an explanatory statement in the legends for Figure 2f (now Figure 2g) and Supplementary Figure 1 to clarify this point, as follows:

Figure 2g: "Example line scan trajectory from one *in vivo* two-photon recording (top; as inputted into the acquisition software – the actual trajectory will differ from that shown due to mirror inertia)."

Supplementary Figure 1:

"The line scan path (as inputted into the acquisition software – the actual trajectory will differ from that shown due to mirror inertia) was directed through the centre of a FITC dextran-filled vessel to extract RBC velocity and a nearby GCaMP6f-positive cell for calcium activations."

- Figure 3e/f: the delta D/D values are extremely small. Please explain.

These delta D/D values are small because the traces contain all the data, which also includes those (majority of) traces where the vessel did not respond to the preceding calcium event. In the traces where only responding data is presented, the delta D/D values show dilations of 1-3%, which is consistent with other results from the capillary bed (e.g. Kisler et al, 2017').

Minor:

- Line 78: lower capillary density does not directly lead to lower CBF and SO₂.

We think the relevant line is below:

"the resting CBF and SO₂ were significantly lower in HC (Figures 2b, c). In part, these differences in net blood flow and oxygenation arise from a lower capillary density in HC than cortex (Figures 2d, e⁹)."

We disagree here and do think that the decreased capillary density is leading to lower CBF and sO₂. For the overall flow of RBCs through the network to be the same in two regions with different vascular densities, then the flux of RBCs through capillaries in the region with the lower density would have to be correspondingly higher. Instead, our new analyses find that the flux is lower in HC than in V1 in individual vessels, as was already suggested by the lower RBC velocities we had reported in the original version of the manuscript. A lower vascular density could also contribute to decreased sO₂: For the same arterial sO₂, and oxygen consumption rate by the tissue, a lower vascular density increases the diffusion path for oxygen from the vessel to the tissue and will decrease tissue oxygen levels, increasing the concentration gradient from the capillary to the tissue and therefore the extraction of oxygen from the blood. Increased oxygen extraction from the blood is also promoted by the slower flow rates of RBCs in capillaries. We summarise this view in the conclusion of the paragraph containing the line the reviewer cites (lines 97-100):

"This combination of a lower capillary density and RBC velocity and flux in the HC explains both the observed lower net flow and, because fractionally more oxygen is extracted from capillaries with lower flow rates¹³, the lower blood oxygenation we measured in HC."

We have also added an extra paragraph to the discussion in response the reviewer's later question about oxygen extraction which also speaks to this point (see below).

- Line 95: are dilations really "less frequent" or do they rather occur for a "shorter time period"?

Dilations are less frequent. We measure whether a dilation (lasting at least 0.5 seconds and of > 1SD of the baseline) occurs in 5 seconds after a calcium event. Therefore dilation duration does not influence our metric of vessel responsiveness.

- Figure 5d: Is the size of the calcium peaks affected by the number of detected cells, i.e. is it necessary to normalize the calcium peaks with the number of cells (which are 559 in HC and 338 in V1)?

The calcium peaks are measured by taking the average trace across all detected cells, not the summed response, or the average across the field of view, irrespective of cell number, so our data should not be affected by the number of detected cells. However we were interested in the point the reviewer raised and conducted further analysis to explicitly test this:

For each recording we randomly selected a subset of the total cells 10 times using the randperm function in MATLAB (i.e. N=15 cells across 10 subsets per each recording). We then averaged across each of these subset traces (now with fewer detected cells) to compare to the overall average trace across all cells. The average peak size was calculated for each of these subset traces, and compared to the mean peak sizes for the averaged net calcium traces (i.e. the data shown in Figure 5d). We did this for HC, V1 and both regions combined, and found no significant differences between the calcium peak size in the averaged traces versus the subset traces with fewer detected cells. This check was most important for the hippocampal data, where the average number of cells detected per recording was 73 (vs in V1, where the average was lower, 23).

- Line 253: Is NVC really compromised or just “different”?

This is a very valid point. We suspect that the weaker NVC might underlie compromised function in disease states but that does not mean that this weaker NVC is compromised in the base state. We have changed the statement (line 354): “demonstrate that hippocampal vascular function is compromised compared to that in neocortex in two major ways”, replacing ‘compromised’ with ‘different’.

- Line 313: Why should lower blood flow allow more oxygen extraction to a larger tissue volume?

We apologise – this sentence is not clear. A “better” way to increase the oxygen supplied by an individual capillary in HC (which supplies a larger tissue volume than a capillary in V1) might be to increase the flux of RBCs. Instead this does not happen and more oxygen has to be extracted from slower moving RBCs. On further consideration, though the slower RBC velocity may help this process, because increasing RBC flux in each vessel would seem to be a more effective way to deal with a lower vascular density, we have removed the sentence about this characteristic being “compensatory” and have added more explanation as follows (lines 449-458):

“Because the vascular density in HC is lower, each capillary supplies oxygen to a larger volume of tissue than in V1. This means more oxygen has to be extracted from each capillary to maintain the same rate of tissue oxygen consumption in HC compared to neocortex. This increase in oxygen extraction could come from an increased supply of oxygen to HC (by an increased flux of RBCs in individual HC capillaries), or by extracting more oxygen from each RBC. An increased flux would sustain oxygen levels in the blood, but our data suggest that in fact oxygen consumption is sustained by increasing oxygen extraction from individual slower-moving RBCs, decreasing blood oxygen saturation in HC.”

- Line 321: Why would oxygen consumption be lower in awake animals? That does not make sense to me.

We apologise – there is a typo in this sentence, which should have read:

*“which is as expected because oxygen consumption rates **are higher** and pO₂ values are lower in awake animals^{31,32}”.*

- Line 595-597: This strongly depends on the oxygen level in the capillary and thus does not generally hold.

We are a little confused as to the reviewer’s meaning here. We think they refer to how we calculated pO₂ in RBCs from sO₂ based on a published Hill curve. We appreciate that the pO₂ will vary with the sO₂ in each individual capillary but our model only calculates an average value based on the average sO₂ in the capillary

bed, as measured with our probe. Thus using the average sO₂ in the Hill equation, with Hill coefficient and P₅₀ calculated from mice of the same background seems to us to be sensible.

Oxygen Modelling:

General:

The oxygen model is very simple, but generally there is nothing wrong with it. In my opinion it is fine to use such a simple model for a rather simple supportive study. However, the limitations/assumptions of such a simple model should be discussed and maybe some additional sensitivity studies should be performed (see comments below).

We thank the reviewer for their general approval. We appreciate it is a very simple model designed to test some simple implications of our experimental results. We would be very interested in seeing further investigation by groups with expertise in modelling oxygen and flow in the whole vascular network to probe the implications of the vascular differences we report here on oxygen supply (as per Schmid et al, 2017², for example).

Major:

- Constants in Equation (4) (lines 604-607): V_{max} and K_m are in the correct range. However, the diffusion coefficient (D) is very weird. First of all, the unit is not correct for a diffusion coefficient, which should be cm²/s (instead of 1/min). If this is only a "typo" the chosen value is 6 order of magnitudes different from what is commonly used. I briefly checked the given reference [50]. However, I did not find the stated value there. The diffusion coefficient is a key factor for the plots in Figure 7.

We are really sorry for the typos in the diffusion coefficient. The diffusion coefficient we ran the model with was correct, being 9.24×10^{-4} cm²/min, but we had both written the wrong value in m²/min for the text and also had then missed out "m²" from the units. It previously read 9.24×10^{-9} /min but now reads " 9.24×10^{-8} m² /min⁵¹". The reference cited gives the diffusion coefficient as 1.54×10^{-5} cm²/sec, equivalent to 9.24×10^{-4} cm²/min.

- The interRBC values of 15mmHg/10mmHg, which were used as boundary conditions for the oxygen model, seem very low compared to the work by the Charpak group. Again, this might affect their general conclusion on the HC being close to hypoxia.

Our predicted values are within the range published by the Charpak group and indeed originate from their work. To try to estimate the value for the interRBC oxygen concentration from our calculated pO₂ values in the RBC we had to use the Charpak group's work to see the general relationship between interRBC and RBC pO₂ values. We looked in their data for RBC pO₂ values that best matched our data and assumed the corresponding interRBC pO₂ values (assuming the same haematocrit as in the Charpak work). Our calculated RBC pO₂ values were 30 mmHg in HC and 43 mmHg in V1, which are both lower than the average RBC pO₂ found in the Charpak work (ca. 65 mmHg in cortex and 55 mmHg in the olfactory bulb; Lyons et al, 2016²⁴ and Parpaleix et al, 2013²⁵, respectively). Nevertheless our values are within the range of values reported across these two brain regions and papers, though the hippocampal values are towards the lower end of this range. From Parpaleix et al (2013)²⁵, Figure 1b, and Lyons et al (2016)²⁴, Figure 4b, data for individual cells RBC and interRBC pO₂ values are given (copied below). We looked at both these figures to estimate the interRBC for both V1 and HC, but erred towards the data from olfactory bulb, which suggested higher estimates of interRBC than could have been estimated from solely considering the cortical data (which could have resulted in estimates of 10 mmHg for V1 and even lower for HC).

Parpaleix et al (2013) Nat Med 19 (2) 241-6

Lyons et al (2016) eLife 5, 1-16

- Points, that should be discussed:

- o The oxygen saturation in capillaries can vary significantly. This will affect the tissue supply radius.

We agree sO₂ and PO₂ are highly variable between capillary branches^{26–28}. Whilst we acknowledge that capillary oxygen saturation is heterogeneous, our methods can only estimate the average steady-state oxygen concentration, and we therefore model the oxygen level in an average capillary. We absolutely acknowledge that this means we are not capturing the heterogeneity in the oxygen levels in and around capillaries in V1 and HC and have added a discussion, as suggested, on this point as follows (lines 467-476):

“As we are not able to measure oxygen in individual capillaries, our results reflect average blood oxygen levels. In the neocortex, individual capillary oxygen levels, and corresponding tissue supply radii, vary quite widely, such that up-stream vessels with higher oxygen levels can supply oxygen to tissue that is physically closer to capillaries with lower blood oxygenation²⁶. The impact of this underlying heterogeneity of oxygenation on our results is not clear: our use of an average may be underestimating microvascular sO₂²⁶, but any such error is likely to be smaller in HC than V1, as the lower vascular density will reduce the chance that an upstream vessel’s supply radius will overlap with that of a downstream, less oxygenated capillary.”

- o The role of the vessel wall on oxygen diffusion (smaller diffusion coefficient).

In response to the reviewer’s comment, we ran the model with lower oxygen diffusion in the vessel wall. There is only a minor effect of incorporating slower diffusion in the vessel wall ([O₂] midway between 50% centile

capillaries changes from 20.3 to 20.2 μM in V1 and 12.1 to 11.9 μM in HC). We have added this analysis into a new supplementary figure (Supplementary Figure 7) which compares our original oxygen diffusion model (from figure 7) with this alternative model with the lower diffusion in the vessel wall for 50th and 95th percentile capillary spacings. We have also added the following text to the results section (lines 347-350):

“Finally, because the vessel wall may act as a diffusion barrier to O₂ delivery to the tissue, we repeated these simulations with a 40% lower diffusion constant in the vessel wall²⁹. Slower diffusion in the vessel wall had only a very small effect on predicted [O₂] and VO₂ profiles in both HC and V1 (Supplementary Figure 7).”

And a description of the model to the methods section (lines 881-886):

“Reduced oxygen diffusion in vessel wall: The basement membrane of the vessel wall may form a diffusion barrier for oxygen supply to the tissue, reducing the effective diffusion coefficient by up to 40%³⁰. To test whether this slowed diffusion significantly affected oxygen delivery to the tissue, we reran the simulations above with a 40% lower value of D with a capillary wall 200 nm thick (the maximum of the reported range⁶⁹; Supplementary Figure 7).”

o What about hematocrit/RBC flux in the HC? Is it the same as in V1?

This was a valuable question, which we have answered with new analyses. Consistent with the higher velocity seen in individual capillaries of V1, the haematocrit and flux are also higher in V1 than HC. We have now added figures showing RBC flux, velocity and haematocrit for each region to Figure 2 (h-j) in the main manuscript, and have updated our methods section to include details on how we calculated these additional parameters.

Minor:

- Why was an oxygen partial pressure of 0 mmHg used as initial condition?

As we are studying steady-state conditions and not interrogating the time-course taken to reach steady-state, the initial starting position for the model does not affect the steady-state solution that we are interested in. There was no particular rationale for using 0 mmHg except that it is a simple value to use across different conditions (rather than, for example, the concentration which changes between regions), though the results would have been the same whatever value was chosen.

- Figure 7c: Why is averaging over cases necessary? Simply, combining the entire would data makes more sense.

We wanted to average the distributions across animals (method 1 in the data given below) so that our values reflected the variability across animals and were not dominated by data from any one animal, if we happened to have more vessels from that animal. Calculating the distribution of capillary spacings by simply combining all the data, as suggested by the reviewer, produced very similar results (see below for V_{max} of 2mM/min for 50th and 95th centiles):

Method 1 is displayed using a solid line and represents the data averaged across animals

Method 2 is displayed using a dotted line and represents the results when combining the whole dataset (i.e. averaging across pixels)

50th percentile

HC: 14.4 (method 1) vs 13.9 μm (method 2)

V1: 10.3 (method 1) vs 10.0 μm (method 2)

95th percentile

HC: 29.5 (method 1) vs 31.6 μm (method 2)

V1: 21.6 (method 1) vs 23.5 μm (method 2)

Running the model for the two different methods using the 50th and 95th centile capillary spacings, with a V_{max} of 2 mM/min showed a very minor difference in the results using either method of calculating 50th and 95th percentile spacings (see below). As we think averaging across animals better reflects the variation observed, we have kept our original method in the manuscript.

- Lines 573-576: Please state the used solubility constant used for Henry's law, instead of pointing to some software.

We have now performed our own calculations for the concentration of oxygen in solution, using a Henry's Law constant and temperature conversion factor as per Sander (2005), and cite this new reference. (Sander, R. Compilation of Henry's law constants (version 4.0) for water as solvent. *Atmos. Chem. Phys.* 15, 4399–4981 (2015)). We get the same values of $[O_2]$ as we did before when using the software.

References in cited text or responses to questions:

1. Grant, R. I. *et al.* Organizational hierarchy and structural diversity of microvascular pericytes in adult mouse cortex. *J. Cereb. Blood Flow Metab.* **39**, 411–425 (2019).
2. Schmid, F., Tsai, P. S., Kleinfeld, D., Jenny, P. & Weber, B. Depth-dependent flow and pressure characteristics in cortical microvascular networks. *PLoS Comput. Biol.* **13**, 1–22 (2017).
3. Montagne, A. *et al.* Blood-brain barrier breakdown in the aging human hippocampus. *Neuron* **85**, 296–302 (2015).
4. Montagne, A. *et al.* Brain imaging of neurovascular dysfunction in Alzheimer's disease. *Acta Neuropathol.* **131**, 687–707 (2016).
5. Wang, H., Golob, E. J. & Su, M. Y. Vascular volume and blood-brain barrier permeability measured by dynamic contrast enhanced MRI in hippocampus and cerebellum of patients with MCI and normal controls. *J. Magn. Reson. Imaging* **24**, 695–700 (2006).
6. Perosa, V. *et al.* Hippocampal vascular reserve associated with cognitive performance and hippocampal volume. *Brain* **143**, 622–634 (2020).
7. Kisler, K. *et al.* Pericyte degeneration leads to neurovascular uncoupling and limits oxygen supply to brain. *Nat. Neurosci.* **20**, 406–416 (2017).
8. Fagrell, B. & Nilsson, G. Advantages and limitations of one-point laser Doppler perfusion monitoring in clinical practice. *Vasc. Med. Rev.* **6**, 97–101 (1995).
9. Fabricius, M., Akgören, N., Dirnagl, U. & Lauritzen, M. Laminar analysis of cerebral blood flow in cortex of rats by Laser-Doppler flowmetry: A pilot study. *J. Cereb. Blood Flow Metab.* **17**, 1326–1336 (1997).
10. Paxinos, G. & Franklin, K. *The Mouse Brain in Stereotaxic Coordinates.* (Elsevier, 2001).
11. Zhang, X. *et al.* High-resolution mapping of brain vasculature and its impairment in the hippocampus of Alzheimer's disease mice. *Natl. Sci. Rev.* **6**, 1223–1238 (2019).
12. Steinman, J., Koletar, M. M., Stefanovic, B. & Sled, J. G. 3D morphological analysis of the mouse cerebral vasculature: Comparison of in vivo and ex vivo methods. *PLoS One* **12**, 1–17 (2017).
13. Buxton, R. B. & Frank, L. R. A model for the coupling between cerebral blood flow and oxygen metabolism during neural stimulation. *J. Cereb. Blood Flow Metab.* **17**, 64–72 (1997).
14. Kohl-Bareis, M. *et al.* System for the measurement of blood flow and oxygenation in tissue applied to

- neurovascular coupling in brain. *Opt. InfoBase Conf. Pap.* **5859**, 1–7 (2005).
15. Liu, H., Kohl-Bareis, M. & Huang, X. Design of a tissue oxygenation monitor and verification on human skin. *Opt. InfoBase Conf. Pap.* **8087**, 1–10 (2011).
 16. Chong, S. P., Merkle, C. W., Leahy, C. & Srinivasan, V. J. Cerebral metabolic rate of oxygen (CMRO 2) assessed by combined Doppler and spectroscopic OCT. **6**, 803–812 (2015).
 17. Huppert, T. J. *et al.* Sensitivity of neural-hemodynamic coupling to alterations in cerebral blood flow during hypercapnia. *J. Biomed. Opt.* **14**, 044038 (2009).
 18. Hall, C. N. *et al.* Capillary pericytes regulate cerebral blood flow in health and disease. *Nature* **508**, 55–60 (2014).
 19. Fernández-Klett, F., Offenhauser, N., Dirnagl, U., Priller, J. & Lindauer, U. Pericytes in capillaries are contractile in vivo, but arterioles mediate functional hyperemia in the mouse brain. *Proc. Natl. Acad. Sci. U. S. A.* **107**, 22290–5 (2010).
 20. Hill, R. A. *et al.* Regional Blood Flow in the Normal and Ischemic Brain Is Controlled by Arteriolar Smooth Muscle Cell Contractility and Not by Capillary Pericytes. *Neuron* **87**, 95–110 (2015).
 21. Atwell, D., Mishra, A., Hall, C. N., O'Farrell, F. M. & Dalkara, T. What is a pericyte? *J. Cereb. Blood Flow Metab.* **36**, 451–455 (2016).
 22. Alarcon-Martinez, L. *et al.* Capillary pericytes express α -smooth muscle actin, which requires prevention of filamentous-actin depolymerization for detection. *Elife* **7**, 1–17 (2018).
 23. Rungta, R. L., Chaigneau, E., Osmanski, B.-F. & Charpak, S. Vascular Compartmentalization of Functional Hyperemia from the Synapse to the Pia. *Neuron* **99**, 362-375.e4 (2018).
 24. Lyons, D. G., Parpaleix, A., Roche, M. & Charpak, S. Mapping oxygen concentration in the awake mouse brain. *Elife* **5**, 1–16 (2016).
 25. Parpaleix, A., Goulam Houssen, Y. & Charpak, S. Imaging local neuronal activity by monitoring PO₂ transients in capillaries. *Nat. Med.* **19**, 241–6 (2013).
 26. Sakadžić, S. *et al.* Large arteriolar component of oxygen delivery implies a safe margin of oxygen supply to cerebral tissue. *Nat. Commun.* **5**, 5734 (2014).
 27. Gould, I. G., Tsai, P., Kleinfeld, D. & Linninger, A. The capillary bed offers the largest hemodynamic resistance to the cortical blood supply. *J. Cereb. Blood Flow Metab.* **37**, 52–68 (2017).
 28. Schmid, F., Barrett, M. J. P., Jenny, P. & Weber, B. Vascular density and distribution in neocortex. *Neuroimage* 1–14 (2017) doi:10.1016/j.neuroimage.2017.06.046.
 29. Colom, A., Galgoczy, R., Almendros, I., Xaubet, A. & Farr, R. Oxygen diffusion and consumption in extracellular matrix gels : Implications for designing three-dimensional cultures. doi:10.1002/jbm.a.34946.

Reviewers' comments:

Reviewer #1 (Remarks to the Author):

The authors have been very thorough in responding to my comments I remain highly enthusiastic about the manuscript. It is a very unique study that finally puts some strong experimental evidence to why hippocampal tissues exhibit variable BOLD signals, and might be more vulnerable to ischemia/hypoperfusion. It is an important paper that will be relevant to many aspects of cerebrovascular biology and neuroimaging. My remaining minor comments are meant to help improve clarity for readers.

- Line 105. Please clarify what types of blood vessels you are studying around this point. I.e. that the majority of the work focuses on capillary or pre-capillary zones.

- Line 108. Please clarify that you are looking at spontaneously occurring neuronal activity, i.e. not triggered by a timed stimulus.

- Line 125. The following two sentences sound at odds with each other. Please clarify. "When we split our movie data in Figure 3 based on vessel size, we found the same result: vessels (both those smaller and larger than 7 μm) were less likely to dilate in HC than V1. Those larger than 7 μm had smaller dilations in HC, whereas those smaller than 7 μm showed the same sized dilations between regions (Supplementary Figure 2)."

- Line 191. It is impressive that different HC layers can be studied in vivo. But I don't think that the different layers and their importance in NV coupling are adequately introduced. Some background on HC anatomy would be helpful. Also, clarify whether imaging GCaMP at different layers is measuring from different subcellular elements of neurons.

- Line 293. Please state how many pericytes and endothelial cells were in the scRNAseq data set.

- Line 395. This sentence is a bit speculative, as a modest difference in MP or TSP length between hippocampus and V1 may not mean a difference in contractile function. The statement is based on the fact that EPs are shorter, but EPs are a very different type of cell in terms of acta2 expression and contractility in vivo. "Capillary (mesh and thin-strand) pericytes were longer in HC than V1, suggesting functional differences that are likely to include a lower contractility, because shorter mural cells are more contractile and pericyte contractility is greatest near the soma 18,21, and because the small capillaries where these mid-capillary pericytes are located dilated less frequently in the HC.

- Line 433. The control studies that hippocampal differences are not due to cranial window implantation should be moved to the results, not the Discussion, as it is an important control.

- This is one of the first studies to image hippocampal vasculature in vivo, and study pericytes histologically. It would be nice to have more comparative pictures to look at, perhaps as extras in the supplement. For example, if the pre-capillary-capillary transition is different, it would be nice to have an accompanying picture to show this if possible.

Reviewer #2 (Remarks to the Author):

See uploaded document. (following page)

Major Concerns:

I appreciate the extensive work the authors have put into the revised version of the manuscript. However, there are serious concerns remaining.

Comment 1: Laser Doppler flowmetry:

I am not convinced by the provided arguments. I think there is still no clear proof that the observed differences in CBF are not caused by the different vascular densities (not sure if this also the case for the spectroscopy data). The RBC velocity data adds to the study. However, as RBC velocities in V1 are recorded closer to the arteriole (see Supplementary Figure 5) this might be the actual cause for the higher RBC velocities in V1. I think these problems need to be discussed more clearly.

Detailed Comments:

- The authors themselves state that the lower vascular density is a “major reason for the lower CBF”. This needs to be described more clearly in the manuscript.
 - o E.g. Line 92-92: The word “arise” is misleading, because in my opinion it suggests the capillary density causes a lower CBF.
 - o E.g. Line 358-359: “blood flow is lower due to a lower vascular density”.
- Impact of superficial layer:
 - o Generally, a relevant and plausible point. But the provided information on the exact methodology used is not sufficient. The contribution from different depths is complex and depends of course on the wavelengths used for spectroscopy (Ref. 9, Table 1).
- Supplementary Figure 10:
 - o Why is the variability within animals and across animals relevant here?
 - o CBF and HbT/HbR values could be plotted together for individual mice/sessions. Together with the available data for capillary density. To gain some insight on the interplay of these quantities.
 - o Error bar for some cases missing or extremely small?
 - o How was the variability of CBF and haemoglobin compared/computed?
 - o For V1 9 mice, in Fig 2 only 8 („outlier mouse“ with CBF ~600 A.U. not in Fig2)

Comment 2: CMRO2:

The argumentation regarding the relative levels of oxy- and deoxyhaemoglobin seems plausible. However, the CBF issue persists. Moreover, CMRO2 is not an independently measured quantity but computed from CBF and HbR and HbT. Formulations such as “CMRO2 is the same despite different CBF and SO2” (line 89-92) are misleading because CMRO2 is computed from these quantities.

Comment 3: RBC velocity in response to activation.

Supplementary Figure 3 adds more insight on this question. Possible points to consider/discuss: 1. The distance between neurons and vessels probably differs in HC and V1, which might explain some differences. 2. If the frequency of RBC velocity increase is the same in HC and V1, this might suggest, that flow regulation occurs at a different location in HC.

Comment 4: Calcium peaks and energy requirement.

This is fine as it is just a hypothesis. Nonetheless, I still have doubts if higher calcium peaks in HC allow the conclusion that the energy requirement is higher and that vessels are “less able to match the demand”. Maybe HC vessels dilate less, because that is the sufficient increase in HC? Same holds for the NVC_index interpretations. (Line 111-112, Line 139-140, Line 364).

Comment 5: mRNA levels and NVC.

OK.

Comment 6: Dilation/constriction of thin-stranded capillary pericytes.

One of the problems here is the exact definition of a thin-stranded pericyte. I am missing the work of Andy Shih and co-workers in the present paper. Ref 32 (Rungta et al.) does not state that capillary dilation mediates a large proportion of overall blood flow increase.

Comment 7: Contractility and NVC.

OK

Comment 8: Reference to Schmid et al.

OK

Comment 9: Scanning trajectory.

OK

Comment 10: Extremely small diameter changes.

Even in the case of a 1% diameter change (i.e. $\Delta D = 0.04$ microns) the authors should keep in mind that this is at the detection limit of 2p microscopy.

Minor Concerns:

Comment 1: Lower capillary density vs CBF and SO₂.

OK

Comment 2: Dilations less frequent.

OK

Comment 3: Size of calcium peaks:

OK

Comment 4: NVC compromised or “different”?

This point should still be interpreted more carefully in the manuscript in general. The NVC rather looks different than compromised. Because of these differences it might be more sensitive.

Comment 5: Blood flow and oxygen extraction.

OK

Comment 6: Oxygen extraction in awake animals.

OK

Comment 7: Oxygen level in the capillary.

Still not 100% clear what the authors mean with the sentence (now lines 853-856).

Oxygen modelling:

General:

The discussion of the model is still not discussed significantly more than before, aspects like potential differences in V_{max} in V1 in HC should be discussed in my opinion or at least tested in the model.

Comment 1: Diffusion coefficient.

OK

Comment 2: interRBC values.

Generally, I agree with the approach of estimating the interRBC PO₂ from available *in vivo* data. However, the observed values are clearly at the lower end of what has been observed *in vivo*. Additionally, only looking at the interRBC PO₂ as oxygen source neglects the significant direct efflux from the RBCs (e.g. see below from Eggleton et al. 2000, Fig. 5, [https://doi.org/10.1016/S0025-5564\(00\)00038-9](https://doi.org/10.1016/S0025-5564(00)00038-9)). Both these factors cause that the modelling studies are a very conservative estimate of how much oxygen is actually available. As suggested previously the sensitivity of these observations should be tested for higher interRBC values to allow robust conclusions on the oxygen availability in HC (e.g. the conclusions in line 361-362, line 370-374).

Fig. 5. Axial profiles of the radial flux density for reference case at the middle of the capillary. Radial flux density (j) values at different radial distances (IC, inner capillary wall; OC, outer capillary wall; OI, outer interstitial space; OT, outer tissue edge) in the model domain as a function of normalized axial position z/r_p , where z ranges from $-L_{tot}/2$ to $+L_{tot}/2$. The vertical dotted lines indicate the trailing and leading edges of the erythrocyte.

Comment 3: Oxygen saturation and tissue supply radius.

OK

Comment 4: Vessel wall

The study is a nice addition to the manuscript. I would actually recommend to only use the results of this steady instead of the old version without capillary wall. However, related to comment 2 the effect of the capillary wall becomes more pronounced for large intravascular oxygen values (e.g. see below from Eggleton et al. 2000, Fig. 3, [https://doi.org/10.1016/S0025-5564\(00\)00038-9](https://doi.org/10.1016/S0025-5564(00)00038-9)). This should be discussed.

Fig. 3. Radial PO_2 profiles for reference case at the middle of the capillary. PO_2 values at different cross sections (C, erythrocyte centerline; LE, leading edge erythrocyte cap; TE, trailing edge erythrocyte cap; LE gap, leading edge of the domain boundary in the plasma gap; TE gap, trailing edge of the domain boundary in the plasma gap); in the model domain as a function of normalized radial position r/r_p , where r ranges from 0 to r_c . The vertical dotted lines from left to right indicate the positions of the erythrocyte lateral surface, inner capillary wall, outer capillary wall, outer interstitial space, and outer tissue edge, respectively.

Comment 5: Hematocrit and RBC flux.

OK

Comment 6: Oxygen partial pressure of 0mmHg as initial condition.

OK. However, in that case, I don't really understand why the transient figures (Fig 7e and h) are added to the manuscript.

Comment 7: Averaging over cases.

OK

Comment 8: Solubility constant:

OK

Comments on newly added text:

- Line 97: Difference in velocity might be because of the differences in the distance to arterioles.
- Lines 243-244: → "no difference could be detected in the artery vein ratio in HC and V1". Stacks with only 1 artery are likely rather small, those concluding that there are no differences might be a little bit strong.
- Figure 3: l,m: p-values are missing (were available in first submission).

Typos:

- Line 692: "where" → were
- Line 853: "be being" → "being"

RESPONSE TO SECOND REVIEW

We would like to thank the reviewers for their time and comments on our revised manuscript and address these all individually below. In addition, since our resubmission we also noticed that we had missed updating statistics on one panel (CMRO₂ fluctuations, Fig. 5h) from parametric to non-parametric statistics. When we did this we found that CMRO₂ fluctuations were significantly larger in V1 than HC. We also therefore calculated an NVC_{index} (CMRO₂/HbT; new panel Fig 5j) to measure the amount of vascular response for each region relative to the changes in energy use of that region. As for our individual vessel 2-photon data, we found that the NVC_{index} was significantly lower in HC than V1. We have therefore updated the results and discussion accordingly as follows:

Results:

“The sizes of the peaks in the CMRO₂ signal (Figures 5f, h) and the associated cerebral blood volume changes (Figures 5g, j) were both smaller in HC than V1, but in HC, these blood volume increases were smaller relative to the change in CMRO₂ (Figure 5j; $NVC_{index} = CMRO_2/Hbt$). Thus, measurements of single vessels and summed regional responses both suggest weaker neurovascular coupling in HC.”

Discussion:

“Firstly, despite equivalent **resting** levels of oxygen consumption in HC compared to the neocortex...”

And

“Secondly, increases in neuronal activity in the HC cause fewer and smaller dilations of local blood vessels and a smaller increase in overall blood volume,...”

Reviewer #1 (Remarks to the Author):

The authors have been very thorough in responding to my comments I remain highly enthusiastic about the manuscript. It is a very unique study that finally puts some strong experimental evidence to why hippocampal tissues exhibit variable BOLD signals, and might be more vulnerable to ischemia/hypoperfusion. It is an important paper that will be relevant to many aspects of cerebrovascular biology and neuroimaging. My remaining minor comments are meant to help improve clarity for readers.

We thank the reviewer and appreciate their comments, which we address below.

- Line 105. Please clarify what types of blood vessels you are studying around this point. I.e. that the majority of the work focuses on capillary or pre-capillary zones.

We have clarified this and the text now reads: “we investigated the capacity of blood vessels to respond to local excitatory neuronal calcium events *in vivo* by capturing movies of neurons and nearby blood vessels (including arterioles, precapillary arterioles and capillaries) in HC and V1 using 2 photon microscopy (Figure 3).”

In the following paragraph, where we go into more detail about separating our data by vessel size (i.e. to specifically look at capillaries <7µm), we have now added the following introductory sentence for clarity:

“We next looked at vessels with diameters < 7 µm to explore the effect of neuronal calcium activity specifically on capillaries.”

- Line 108. Please clarify that you are looking at spontaneously occurring neuronal activity, i.e. not triggered by a timed stimulus.

We are looking at neuronal activity local to a vessel regardless of external condition. For clarity we have added the line: "Calcium events occurred across environmental conditions (i.e. in the dark or when stimuli were presented on the screens, and when the mouse was at rest or running)."

- Line 125. The following two sentences sound at odds with each other. Please clarify. "When we split our movie data in Figure 3 based on vessel size, we found the same result: vessels (both those smaller and larger than 7 μm) were less likely to dilate in HC than V1. Those larger than 7 μm had smaller dilations in HC, whereas those smaller than 7 μm showed the same sized dilations between regions (Supplementary Figure 2)."

This reference to the "same result" is supposed to refer to the fact that the linescan data and the xy movie data for capillaries $<7 \mu\text{m}$ both show that HC vessels are less likely to dilate than V1 vessels. We have rewritten this paragraph, which we hope is now clearer.

"When we split our xy movie data in Figure 3 into vessels smaller and larger than 7 μm , we confirmed that both groups of vessels were less likely to dilate in HC than V1. When they did dilate, vessels larger than 7 μm also had smaller dilations in HC, whilst dilations in smaller vessels were the same size in both regions (Supplementary Figure 2)."

- Line 191. It is impressive that different HC layers can be studied in vivo. But I don't think that the different layers and their importance in NV coupling are adequately introduced. Some background on HC anatomy would be helpful. Also, clarify whether imaging GCaMP at different layers is measuring from different subcellular elements of neurons.

We have moved Supplementary Figure 8 a&b, showing the anatomical organisation of V1 and HC to Figure 1 and have added a couple of sentences to the introduction to give more background on the HC anatomy and the different subcellular elements in these layers:

"This cranial window allowed us to record from different layers across both regions, which in visual cortex comprised of layers I-IV (Figure 1a), and in CA1 of stratum oriens, stratum pyramidale, stratum radiatum, and stratum lacunosum-moleculare (Figure 1b). In V1 neuronal cell bodies are more dispersed throughout the layers, although slightly more concentrated in layer IV, whereas in CA1 the cell bodies are densely packed in stratum pyramidale and send their long, apical dendrites into stratum radiatum."

The data in Supplementary Figs 3 and 4 could suggest increased NVC in dendritic compartments, but are not unequivocal. We have therefore added some discussion on this topic in the supplementary material after Supplementary Figure 4:

"The subcellular elements of neurons are differently represented in the different layers, with a higher density of soma in SP of HC and L2/3 of V1 compared to the other layers. Our results across Supplementary Figs 3 and 4 show some indication that dendritic areas might show stronger neurovascular coupling: In HC, regions with more soma show smaller dilations (Fig S3f), and the soma-dense pyramidal layer also shows the smallest dilations (Fig. S4f). Furthermore, in V1, L1 showed larger diameter changes and $\text{NVC}_{\text{indices}}$ than L2/3 (Fig S4h-j). However, the evidence for stronger NVC in neuropil-dominated regions is equivocal. When the different sizes of local calcium responses are accounted for, by using the $\text{NVC}_{\text{index}}$, there are no longer any differences in HC

between layers (Fig S4g), or between neuropil- and soma-dominated regions (Fig S3g). Additionally, in V1, the significant laminar changes between L1 and L2/3 (Fig S4h-j) cannot be attributed to different levels of soma vs neuropil, as when we explicitly tested this, there were no significant differences between soma or neuropil-dominated regions (Fig S3h-j). Therefore, the presence or absence of laminar differences do not seem to be simply due to preferential vascular responding to signals from neuropil over soma.”

- Line 293. Please state how many pericytes and endothelial cells were in the scRNAseq data set.

We have now included the numbers of mural and endothelial cells in the results section (“Functional differences between vascular cells in HC and V1 suggested by different mRNA expression profiles” on lines 299-300):

“we examined the mRNA expression profile of mural (n=83; 20 in HC) and endothelial (n=137; 10 in HC) cells”.

- Line 395. This sentence is a bit speculative, as a modest difference in MP or TSP length between hippocampus and V1 may not mean a difference in contractile function. The statement is based on the fact that EPs are shorter, but EPs are a very different type of cell in terms of acta2 expression and contractility in vivo. “Capillary (mesh and thin-strand) pericytes were longer in HC than V1, suggesting functional differences that are likely to include a lower contractility, because shorter mural cells are more contractile and pericyte contractility is greatest near the soma^{18,21}, and because the small capillaries where these mid-capillary pericytes are located dilated less frequently in the HC.

We agree that the suggestion that shorter cells may be less contractile is speculative. It is not entirely based on the fact that EPs are shorter than MP or TSPs (and SMCs are shorter still), but also on our own observations that response frequency tends to decrease at high branch orders beyond the switch from EP to MP/TSP. We agree that this speculation is a bit out of place, however, so have deleted the reference to shorter mural cells, so the sentence in the results now reads:

“Because pericyte contractility is strongest nearer the cell body^{12,21}, HC pericytes may be less contractile than their counterparts in V1, which could underlie the weaker neurovascular coupling in HC.”

And that in the discussion:

“Capillary (mesh and thin-strand) pericytes were longer in HC than V1, suggesting functional differences that may include lower contractility, because pericyte contractility is greatest near the soma^{1,2}, and because the small capillaries where these mid-capillary pericytes are located dilated less frequently in the HC. “

- Line 433. The control studies that hippocampal differences are not due to cranial window implantation should be moved to the results, not the Discussion, as it is an important control.

We have now added a discussion about the effects of the cranial window surgery on hippocampus to the main results section, and added the bolded text (the relevant supplementary figures are now S5 and S6):

“HC neurovascular coupling is not lower because of vascular deficits arising from the surgical preparation

“The surgery to create a cranial window over CA1 is more invasive than for V1, because it requires aspiration of some of the overlying cortex and the implantation of a cannula with glass coverslip. We tested for vascular network damage in CA1 in the surgical versus non-surgical hemisphere. We found that the aspiration and cannula implantation did not cause significant compression to dorsal CA1, nor alter the overall vascular density of the region (Supplementary Figure 5). **Furthermore, signs of inflammation were limited to tissue <100 µm from the window and did not affect the responses recorded (Supplementary Fig 6).”**

- This is one of the first studies to image hippocampal vasculature in vivo, and study pericytes histologically. It would be nice to have more comparative pictures to look at, perhaps as extras in the supplement. For example, if the pre-capillary-capillary transition is different, it would be nice to have an accompanying picture to show this if possible.

The confocal images taken of the vasculature with DsRed labelled SMCs and pericytes were imaged 2 years ago, and so are unfortunately no longer fluorescent. Whilst the images used in the manuscript were taken in enough detail for us to classify the pericyte type and measure the pericyte length/vessel diameter (e.g. for Figure 6), we did not take high resolution images of the transition point in the vascular networks of CA1 and cortex. Sadly, we also no longer have any NG2-dsRed mice available to FITC-perfuse and take more detailed confocal pictures of the vasculature.

In order to show the reviewer, we have imaged some slices from NG2-DsRed human APOE3 targeted replacement mice which have been FITC gel-filled. We have included some images below of the transition point between arterioles and the capillary bed (vessels in green, NG2-DsRed labelled pericytes in red). Whilst we have not formally characterised this here, we do not observe any striking differences between the neocortex and CA1, as the hippocampus also appears to show a pattern of increasingly reduced vessel diameter and mural cell coverage as arterioles transition into capillaries. Given this, and the fact that the APOE3 genotype is different from the murine APOE used for our other data (which may, or may not affect pericyte coverage, Halliday et al. (2016); doi: 10.1038/jcbfm.2015.44, Koizumi et al. (2020); doi: 10.1038/s41467-018-06301-2) we do not think it a sufficiently reliable addition to the manuscript.

V1

Below are images of just the NG2-dsRed labelled pericytes around transition points in HC (top row) and V1 (bottom row):

Reviewer #2 (Remarks to the Author):

Major Concerns:

I appreciate the extensive work the authors have put into the revised version of the manuscript. However, there are serious concerns remaining.

Comment 1: Laser Doppler flowmetry:

I am not convinced by the provided arguments. I think there is still no clear proof that the observed differences in CBF are not caused by the different vascular densities (not sure if this also the case for the spectroscopy data). The RBC velocity data adds to the study. However, as RBC velocities in V1 are recorded closer to the arteriole (see Supplementary Figure 5) this might be the actual cause for the higher RBC velocities in V1. I think these problems need to be discussed more clearly.

We agree that the lower vascular density is likely to be a contributing factor (but not the whole explanation) of the lower CBF in HC as we explain further below. Regarding the RBC velocities, we wondered if the reviewer's point might be the reason for our observed differences, but our analysis

suggests that, unlike in V1, there is no increase in HC RBC velocity nearer to the source (Supplementary Figure 7c), suggesting that distance from source cannot simply explain the different in RBC velocity observed. Rather, the differences come because in V1 some vessels close to the source have higher RBCV than those at an equivalent distance from the source in HC. We provide further address to the specific points below.

Detailed Comments:

- The authors themselves state that the lower vascular density is a “major reason for the lower CBF”. This needs to be described more clearly in the manuscript.

We absolutely agree that a major contributing factor to the different CBF values for the two regions is the difference in vascular density. The net CBF in the region will be a function of both the vascular density and the flux in each of these vessels. Our data show that there is a decrease in CBF in HC, and that this is likely caused by a reduction in both the vascular density, and a decrease in the flux of red blood cells in each vessel. We do indeed state this in the manuscript, saying (L98-99) “In part, these differences in net blood flow and oxygenation arise from a lower capillary density in HC than cortex (Figures 2d, e^{3,4}.” And at the end of the same paragraph (L104): “This combination of a lower capillary density and RBC velocity and flux in the HC explains both the observed lower net flow and, because fractionally more oxygen is extracted from capillaries with lower flow rates⁵, the lower blood oxygenation we measured in HC.” To try and further clarify that we think that the vascular density and the lower RBCV and flux measures create this lower CBF, we have amended L104 to emphasise that the RBCV/flux differences are additional to the vascular density differences.

“we found that, despite the sampled vessels themselves being of equal size (Figures 2f), RBC velocity, flux and haematocrit were **also** significantly lower in the HC than V1 (Figures 2h-j).”

o E.g. Line 92-92: The word “arise” is misleading, because in my opinion it suggests the capillary density causes a lower CBF.

This is our intention. As discussed above, the amount of blood flow through a region is a function of the number/density of vessels and the flow in each vessel. Therefore, a lower capillary density does indeed cause a lower CBF.

o E.g. Line 358-359: “blood flow is lower due to a lower vascular density”.

- Impact of superficial layer:

o Generally, a relevant and plausible point. But the provided information on the exact methodology used is not sufficient. The contribution from different depths is complex and depends of course on the wavelengths used for spectroscopy (Ref. 9, Table 1).

Here the reviewer is referring to our previous response to reviews not the main text (point 4 of our response to their point 8). Our previous response gave the wavelength used and fibre separation (780nm and 500 μ m; copied below) but we realise now that we did not make it sufficiently clear that these parameters are the same as those used to experimentally determine the contribution of different wavelengths by Fabricius et al (PMID: 9397032). Thus, we can accurately use their results (as in our earlier response) to estimate the contribution of different layers to our measured response. We do give these parameters and cite Fabricius in our methods section, as well as their measured depth range for these parameters (median = 0.9mm, max = 2mm), as well as our conclusion that the contaminating regions are unlikely to be generating the regional differences we report. We are not clear what additional methodological information the reviewer would like us to report.

For reference we copy the relevant text below, highlighted to show these methodological details:

“Sampling depth: Our LDF probe has an excitation-detection fibre separation of **500 μm and an excitation wavelength of 780 nm** so therefore, unlike other fibre designs, has a deep sampling depth, with a median depth sampled of around 0.9 mm and vessels as deep as 2 mm can contribute to the signal ⁶. Thus, the sampling volume for both regions extends much deeper than the depth of vessels imaged using 2-photon microscopy (400 μm): In mouse the cortical thickness is about 1 mm, while the thickness of CA1 and the sulcus containing the feeding vessels is about 0.9 mm and the whole hippocampus (CA1, plus the deep vessels in the sulcus and the dentate gyrus) is about 1.4 mm ⁷. From Fabricius et al ⁶, this means that approximately 70 % of the HC LDF signal actually derives from the hippocampus (including the dentate gyrus; 55% originating from CA1 plus the feeding vessels), the rest being from thalamus, and 60 % of the V1 LDF signal derives from V1, the rest being from the corpus callosum and posterior hippocampus. Because the corpus callosum has a low vascular density, so cannot be artificially boosting the V1 LDF signal, and the thalamus has a high vascular density⁴ so cannot be artificially reducing the HC LDF signal, the LDF differences we observe in V1 compared to HC are highly likely to originate in perfusion differences between these two regions.”

- Supplementary Figure 10:

o Why is the variability within animals and across animals relevant here?

It shows that the probe placement does not wildly affect the results (e.g. due to placement over large feeding vessels for example) – suggesting that this method is providing consistent information over repeated sessions and animals.

o CBF and HbT/HbR values could be plotted together for individual mice/sessions. Together with the available data for capillary density. To gain some insight on the interplay of these quantities.

As suggested we have plotted Hbt and Hbr against CBF (see below), averaged for each mouse (top row) or as individual sessions (bottom row). These plots suggest that there is no interaction between CBF and Hbt or Hbr. i.e. blood flow is not differently related to blood volume or deoxyHb in the two regions. We do not have capillary density data from the same mice as the haemodynamic measures so are unable to directly correlate capillary density measurements with probe recordings from the same animals. Because of the lack of interactions we have not included these figures in the manuscript.

o Error bar for some cases missing or extremely small?

Each symbol represents a mouse, while error bars reflected repeated sessions in the same animal (+/-SEM). Some mice were only imaged for one session (or had very small error bars between sessions), we have now added the range of session numbers per mouse in the legend of Figure 2 (“1-5 sessions per mouse”).

o How was the variability of CBF and haemoglobin compared/computed?

We realise that this section was not sufficiently clear or accurate, as we had not previously formally compared variability. We have now done this, by calculating intersession standard deviation for each animal and comparing these values across regions. The text has been corrected as follows.

We have changed. “We plotted the variability between data recorded on different sessions within the same animal and between animals (Supplementary Figure 10).”

To...

“We plotted the responses recorded on different sessions within the same animal and between animals (Supplementary Figure 10). We then compared the standard deviation for each mouse (with ≥ 2 recording sessions, N=6 mice for HC, N=7 mice for V1) between measurements and across regions and compared variability between regions. There was no difference in inter-session standard deviation between brain regions for any of the haemodynamic measures (independent sample t-tests; $p > 0.19$).

o For V1 9 mice, in Fig 2 only 8 (outlier mouse“ with CBF ~600 A.U. not in Fig2)

The reviewer is correct, that an “outlier” mouse was initially removed from the oxyprobe data in Figure 2, but not in Supplementary Figure 10. This mouse had a V1 surgery, and was removed due to

the very high flux values (flux is >50% larger than the average, as for this mouse it was 624, and the average across the other animals is 377). Including data from this mouse in Fig 2 increases the size of the regional difference in CBF (see below). The removal of this mouse from the dataset did not change the results (and in fact only further emphasizes the higher values in V1 vs HC).

Includes 'outlier' mouse:

Excludes 'outlier' mouse:

The outlier mouse is shown in our Supp Fig. 10 (copied below) as the 4th data point in the V1 data for each haemodynamic measure. Inspecting these plots reveals that the mouse was only an outlier in CBF (and CMRO2 that is calculated from CBF) and not in the haemoglobin spectrometry data. We have therefore concluded that there is not sufficient reason to remove it and now have included data from this mouse in Figure 2.

Comment 2: CMRO2:

The argumentation regarding the relative levels of oxy- and deoxyhaemoglobin seems plausible. However, the CBF issue persists. Moreover, CMRO2 is not an independently measured quantity but computed from CBF and HbR and HbT. Formulations such as “CMRO2 is the same despite different CBF and SO2” (line 89-92) are misleading because CMRO2 is computed from these quantities.

We wonder here if the reviewer means that if CBF readings are lower because of a lower vascular density then this could invalidate the calculation of CMRO₂ in HC? We again would like to agree that CBF, as a measure of flow of blood through a tissue volume, is going to be affected by the vascular density, but this is the appropriate measure to use for the calculations of tissue oxygen consumption: the difference in arterial vs venous blood oxygen, estimated from the fraction of

deoxyhaemoglobin vs total haemoglobin, and the amount of blood flowing through an area (which is indeed dependent on the vascular density) are both needed for $CMRO_2$ calculation. Thus, a low flow through a region (because of a low vascular density, for example) would need to have a large oxygen extraction fraction (reflected in a large fraction of deoxyhaemoglobin vs total haemoglobin) in order to sustain a high level of oxygen consumption.

We agree, of course, that $CMRO_2$ is not independent of CBF and Hbr and is calculated from these measures. That does not mean, however, that it is not noteworthy that resting $CMRO_2$ does not differ between regions, when CBF and sO_2 do - they represent different, but connected features of the system and by considering all three we can best understand what is going on. If $CMRO_2$ was much lower in HC than V1, for example, CBF could be lower in HC than V1, but sO_2 higher. By calculating $CMRO_2$ from CBF and sO_2 we therefore learn whether the changes in CBF and sO_2 reflect altered underlying oxygen use.

Comment 3: RBC velocity in response to activation.

Supplementary Figure 3 adds more insight on this question. Possible points to consider/discuss:

1. The distance between neurons and vessels probably differs in HC and V1, which might explain some differences.
2. If the frequency of RBC velocity increase is the same in HC and V1, this might suggest, that flow regulation occurs at a different location in HC.

In answer to point 1, we are not sure whether the reviewer means that there might be a difference in the distance of the closest neuronal soma to a vessel between the two regions, or whether an average neuron is further or closer to the vessel in the two regions. If the former, the question becomes whether neuronal soma or neuropil being nearest the vessel could more strongly affect NVC (as parenchymal space can be considered as containing either neuropil or neuronal soma). We consider this question in Supplementary Fig 3, finding that in HC, but not V1, NVC was stronger in regions with more neuropil (and therefore on average likely further from a neuronal soma), though as discussed above in response to the other reviewer, the results are somewhat equivocal on the question of whether responses are generally stronger in neuropil. If the reviewer meant that the average distance of a neuron (be it soma or neuropil) from a vessel is different in the two regions, then indeed this is the case: the lower vascular density in HC means that, on average, neurons are further from a capillary than in V1, and we model the effect this has on O_2 levels. It is not clear to us how this difference would affect RBCV values, however.

For the reviewer's second point, it is indeed possible that flow regulation occurs at a different location in HC and V1, but I do not think it is necessarily the case. Vasodilatory signals usually propagate upstream to initiate vasodilation^{8,9} and increase in RBCV, but reduced dilations in any vascular compartment will reduce the size of these RBCV responses⁸. Because RBCV reflects the pressure state of the wider network, in our experiments as long as any upstream vessel dilates, there is likely to be an increase in RBCV, but this response will be smaller if the local or upstream dilations are smaller or less frequent. In this scenario, the location of flow regulation might be less consistent than in V1, but not necessarily in a different location or vascular compartment. This is a very interesting question that deserves further study but our thoughts on this are currently somewhat too speculative an answer for the current Discussion, given space constraints.

Comment 4: Calcium peaks and energy requirement.

This is fine as it is just a hypothesis. Nonetheless, I still have doubts if higher calcium peaks in HC allow the conclusion that the energy requirement is higher and that vessels are "less able to match

the demand". Maybe HC vessels dilate less, because that is the sufficient increase in HC? Same holds for the NVC_index interpretations. (Line 111-112, Line 139-140, Line 364).

We take the reviewer's point and have removed phrases suggesting a lack of ability with those pointing out a difference, as follows:

L120: "This suggests hippocampal vessels were less able to match increased activity with increased vasodilation and therefore blood flow." Becomes "This suggests that, relative to the amount of neuronal activity, hippocampal vessels dilate less than those in V1, supplying less oxygen".

L150: "suggesting a decreased ability of the hippocampal vasculature to match increased blood flow to increased oxygen use compared to cortex. This reduced ability to match blood supply..." becomes "suggesting that the hippocampal vasculature increases blood flow less in response to increased oxygen use compared to cortex. The reduced matching of blood supply...".

We think the reviewer's mention of L365 is in reference to our term "weak neurovascular coupling" which we have retained as it means not a lack of ability, but a less strongly coupled response (as we observe).

Comment 6: Dilation/constriction of thin-stranded capillary pericytes.

One of the problems here is the exact definition of a thin-stranded pericyte. I am missing the work of Andy Shih and co-workers in the present paper.

Our analyses of thin-strand, mesh and ensheathing pericytes are directly based on Andy Shih's elegant work describing these pericyte variants and we cite his 2019 JCBFM paper extensively (Ref 18, first author R.I. Grant).

Ref 32 (Rungta et al.) does not state that capillary dilation mediates a large proportion of overall blood flow increase.

Their figure 6 makes this conclusion clear (partially copied below – the predicted impact on CBF of removing dilation by each vessel type is shown in the third column) and the authors state in the relevant section of their results "Strikingly, blockage of the passive capillary dilation dramatically affected the hemodynamic response in the other compartments as well as the blood flow change".

Comment 10: Extremely small diameter changes.

Even in the case of a 1% diameter change (i.e. $\Delta D = 0.04$ microns) the authors should keep in mind that this is at the detection limit of 2p microscopy.

This is a very good point. Indeed, the size of some of the dilations we are reporting are very small (e.g. a 2% dilation of a 6 μm vessel [Supplementary Figure 2] is only 120nm). Our responses are classified by being >1 SD from baseline and were not observed when vessel data was shuffled with respect to calcium peaks – so we are confident we can detect small changes. Based on the average baseline SD, we estimate our average minimum resolvable dilation as being around 120 nm (the average baseline SD of our recordings). Therefore, the smallest dilations we observe are at our detection limit, and we can only detect these smallest responses we report when the baseline is particularly smooth. This means that we would be unable to detect even smaller changes, should they occur, a point we consider when noting that we would not be able to detect hippocampal dilations in the smallest vessels if they were equivalently reduced compared to V1, as they would be below our detection limit, saying:

“When they did dilate, vessels larger than 7 μm also had smaller dilations in HC, whilst dilations in smaller vessels were the same size in both regions (Supplementary Figure 2). Dilations of HC vessels larger than 7 μm were 51% of those in V1, however, so similarly-scaled responses in smaller vessels would be undetectable (and classed as non-responders) as they would be <60 nm, or only 0.5 SDs above baseline.”

Minor Concerns:

Comment 4: NVC compromised or “different”?

This point should still be interpreted more carefully in the manuscript in general. The NVC rather looks different than compromised. Because of these differences it might be more sensitive.

We have tempered these conclusions generally as suggested by the reviewer and as detailed above.

Comment 7: Oxygen level in the capillary.

Still not 100% clear what the authors mean with the sentence (now lines 853-856).

We have now added the Hill equation for dissociation of oxygen from haemoglobin to make clear what we are referring to here, as well as slightly rephrasing the sentence as follows:

“From SO₂, we calculated the partial pressure (pO₂) of oxygen in RBCs using the haemoglobin oxygen dissociation curve for C57/BL6 mice (the background of our experimental mice), with a Hill coefficient of 2.59 and a P50 of 40.2 mmHg¹⁰ (4).”

$$pO_2 = \frac{sO_2^{2.59}}{sO_2 + 40.2} \quad (4)''$$

Oxygen modelling:

General:

The discussion of the model is still not discussed significantly more than before, aspects like potential differences in V_{max} in V1 in HC should be discussed in my opinion or at least tested in the model.

We did not explicitly look at different V_{maxes} in the two regions, because our empirical measurements suggest similar rates of CMRO₂ in both regions, suggesting similar V_{maxes}. However we simulated O₂ levels across the likely physiological range of V_{max} for brain so comparisons between regions with different V_{maxes} can be achieved by examining Fig 7k-m. Fig 7l suggests that the tissue furthest from a capillary in HC will have a more restricted V_{O₂} than V1 even if the V_{max} in HC is substantially lower than that in V1 (e.g. HC V_{max} = 1mM/min, V1 V_{max} = 3mM/min). We have made this more explicit in the text amending the text as follows:

“To determine if oxygen became limiting for ATP production in the tissue, we then calculated the oxygen consumption rate (VO₂) as a proportion of the maximum rate of oxygen consumption (V_{max}; Figures 7g,j,l,m). In V1, VO₂ (and therefore the rate of ATP generation) occurred at over 90% of the V_{max} even far from a capillary, **and at the upper range of V_{maxes} tested**, suggesting [O₂] barely limited ATP synthesis. In HC, however, while VO₂ was sustained at over 90% of the V_{max} in tissue at the median distance from a capillary, in the tissue furthest (95th centile) from a capillary, [O₂] dropped to concentrations that limited VO₂ **even at low oxygen consumption rates (≥1 mM/min).**”

Comment 2: interRBC values.

Generally, I agree with the approach of estimating the interRBC PO₂ from available *in vivo* data. However, the observed values are clearly at the lower end of what has been observed *in vivo*. Additionally, only looking at the interRBC PO₂ as oxygen source neglects the significant direct efflux from the RBCs (e.g. see below from Eggleton et al. 2000, Fig. 5, [https://doi.org/10.1016/S0025-5564\(00\)00038-9](https://doi.org/10.1016/S0025-5564(00)00038-9)). Both these factors cause that the modelling studies are a very conservative estimate of how much oxygen is actually available. As suggested previously the sensitivity of these observations should be tested for higher interRBCs values to allow robust conclusions on the oxygen availability in HC (e.g. the conclusions in line 361-362, line 370-374).

Fig. 5. Axial profiles of the radial flux density for reference case at the middle of the capillary. Radial flux density (j) values at different radial distances (IC, inner capillary wall; OC, outer capillary wall; OI, outer interstitial space; OT, outer tissue edge) in the model domain as a function of normalized axial position z/r_p , where z ranges from $-L_{tot}/2$ to $+L_{tot}/2$. The vertical dotted lines indicate the trailing and leading edges of the erythrocyte.

We are not convinced that using artificially raised estimates of interRBC pO_2 would allow stronger conclusions about the different oxygen levels. We used interRBC pO_2 as this has been shown empirically to equilibrate with and reflect tissue pO_2 , in olfactory bulb where capillaries are spaced close together as in V1 (see Parpaleix et al, 2013, Fig 2, copied below).

This lack of a detectable O_2 gradient across the tissue is similar to our predicted curves for V1 (our Fig 7f) where the small drop in O_2 away from the capillary could be hidden by noise in the experimental phosphorescent detection of pO_2 (see their fig 2c, above). Thus, the interRBC pO_2 is the best estimate of the capillary pO_2 that is driving tissue pO_2 . However to verify this, we have done two new analyses. Firstly we simulated O_2 diffusion into the tissue using the RBC pO_2 values (see below; V_{max} values given next to each labelled line). As expected, these simulations yielded much higher tissue pO_2 values, which were much above the interRBC values estimated from RBC pO_2 – and must therefore be unphysiologically high, as the tissue pO_2 is expected to be no higher than the predicted interRBC pO_2 (14 and 21 μM for HC and V1, respectively).

Secondly, we tried to account for some additional O₂ delivery from RBCs in a physiologically constrained way, by considering that the interRBC value instead reflected the median mid-capillary pO₂, and this (rather than a decrease in O₂ from the interRBC value) produced the flat gradient observed by Parpaleix et al. To study this we varied the capillary pO₂ to generate the same pO₂ at the median mid-capillary position as the estimated interRBC pO₂ (see below) for each region (allowing for some O₂ delivery directly from RBCs) for a V_{max} of 2. This generated relatively flat tissue pO₂ gradients that, particularly in V1 were shallow enough to be consistent with the flat inter-capillary gradient observed by Parpaleix et al. However, using this same higher capillary pO₂ to model tissue at the 95th centile furthest from a capillary, O₂ levels now fall to a level where respiration becomes limited by O₂, dropping to around 70% of V_{max} (compared with falling to 60% in our original simulation).

This simulation is more realistic than that using RBC pO₂ at the capillary, as the median condition mimics experimental data in regions with a high capillary density. We have therefore included this simulation as Supplementary Figure 9 and added the following text.

“Because the oxygen level between rather than within RBCs better reflects tissue oxygen levels²⁷, we used the pO₂ between two RBCs as our estimate of capillary pO₂ (see Online Methods). InterRBC pO₂ was 15 mmHg (21 µM) in V1 and 10 mmHg (14 µM) in HC. Oxygen diffusion into the tissue was then simulated (Figure 7d), assuming varying rates of neuronal oxygen consumption corresponding

to values reported in rodent tissue^{28,29}. Because in some tissue, the oxygen gradient between the interRBC pO₂ and the tissue is flat (within measurement error), and equals the interRBC pO₂²⁷, we also ran simulations with a slightly higher capillary pO₂ (18 μM in HC, 22 μM in V1), which yielded the predicted interRBC pO₂ at the median distance between two capillaries (Supplementary Figure 9). This allows for some O₂ delivery directly from RBCs themselves and is still consistent with published tissue O₂ gradients.”

Then later report the results as follows:

“In our simulations with a larger capillary pO₂, the decrease in O₂ was slightly smaller, but still limited VO₂ far from the capillary to 70% of V_{max} when V_{max} = 2 mM/min (Supplementary Figure 9, vs. 60% of V_{max} with capillary pO₂ fixed at interRBC pO₂, Figure 7I). Thus our results suggest that HC ATP production through oxidative phosphorylation is restricted in tissue furthest from a capillary.”

And the following in the methods:

“Indeed, if we ran our simulations using RBC pO₂ as the capillary [O₂], tissue [O₂] estimates were impossibly much higher than the expected interRBC [O₂] values (not shown), confirming that interRBC rather than intraRBC [O₂] levels are likely to be the main drivers of tissue [O₂].”

Comment 4: Vessel wall

The study is a nice addition to the manuscript. I would actually recommend to only use the results of this steady instead of the old version without capillary wall. However, related to comment 2 the effect of the capillary wall becomes more pronounced for large intravascular oxygen values (e.g. see below from Eggleton et al. 2000, Fig. 3, [https://doi.org/10.1016/S0025-5564\(00\)00038-9](https://doi.org/10.1016/S0025-5564(00)00038-9)). This should be discussed.

Fig. 3. Radial PO₂ profiles for reference case at the middle of the capillary. PO₂ values at different cross sections (C, erythrocyte centerline; LE, leading edge erythrocyte cap; TE, trailing edge erythrocyte cap; LE gap, leading edge of the domain boundary in the plasma gap; TE gap, trailing edge of the domain boundary in the plasma gap); in the model domain as a function of normalized radial position r/r_p , where r ranges from 0 to r_s . The vertical dotted lines from left to right indicate the positions of the erythrocyte lateral surface, inner capillary wall, outer capillary wall, outer interstitial space, and outer tissue edge, respectively.

This is a very good point. We have now updated the main oxygen diffusion figure (Figure 7) to only include the results of the model with a capillary wall (see below), and have amended the methods appropriately. (The new simulations (now supp fig 9) above also use the lower D in the capillary wall). Given that we now use this for all conditions simulated we have not included any additional discussion about the relative effect of including different diffusion in the vessel wall at different oxygen concentrations. We are also not sure that the effects of increasing haematocrit (and therefore capillary pO₂), as simulated in the reference cited by the reviewer, would be directly

comparable to the effects of simply assuming a higher pO₂ – as the contribution of RBC pO₂ will likely be more dominant in the former situation.

Figure 7

Comment 6: Oxygen partial pressure of 0mmHg as initial condition.

OK. However, in that case, I don't really understand why the transient figures (Fig 7e and h) are added to the manuscript.

This was to demonstrate that the model has reached steady-state at the time point we then use to extract the concentration profiles across the tissue. This has now been clarified in the legend for figure 7 as follows: "Simulated time courses from initial conditions of zero [O₂] for HC (purple) and V1 (orange) for tissue at (e) the median or (h) 95th centile distance from a capillary, showing that steady-state is reached by 6s (when [O₂] profiles were extracted)."

Comments on newly added text:

- Line 97: Difference in velocity might be because of the differences in the distance to arterioles.

As mentioned above, we also had wondered if an increased distance from a feeding arteriole might explain the slower RBCV in HC. However, our Supplementary Figure 7c shows that while RBCV decreases with increased distance from the feeding arteriole in V1, this is not the case in HC, where RBCV is low even close to the arterioles. The difference between V1 and HC in the distance between measured vessels and arterioles cannot therefore simply account for the difference in RBCV. We discuss in the results:

“...the vessels we sampled in HC were generally further from their source (HC vessels averaged 227 μm from their source, versus 135 μm in V1), but the distance to the perfusion source did not alter resting RBC velocity or calcium-dependent blood vessel dilations in HC (Supplementary Figure 7). In V1, the distance from the pial arteries did not affect the size of dilations, but RBC velocity was higher in vessels nearer the source arteries (Supplementary Figure 7).”

and in the Discussion:

“In V1, RBC velocity was faster nearer the arterial source, consistent with a shorter, lower resistance path being taken by RBCs passing through superficial neocortical layers³¹. HC flow did not show this dependence on distance from the perfusion source, suggesting the absence of these shorter, lower resistance paths in HC.”

- Lines 243-244: à “no difference could be detected in the artery vein ratio in HC and V1”. Stacks with only 1 artery are likely rather small, those concluding that there are no differences might be a little bit strong.

We think this is actually a carefully worded statement already – we purposefully do not say that there are definitely no differences, just that we could not detect a difference.

- Figure 3: l,m: p-values are missing (were available in first submission).

We had accidentally removed these during figure editing (they were still in the text), but have now put them back as suggested.

Typos:

- Line 692: “where” → were
- Line 853: “be being” → “being”

Thank you – these typos have now been corrected.

Reviewers' comments:

Reviewer #1 (Remarks to the Author):

The authors have addressed all of my concerns. I congratulate them on a timely and innovative study. It is a foundational step toward understanding how hippocampal blood flow dynamics are regulated in health, and eventually in disease.

Reviewer #2 (Remarks to the Author):

The authors have adequately addressed all my concerns in this revision.